# Microvascular endothelial scavenger receptor class B type I protects against heart failure with preserved ejection fraction by inhibiting T-cell cardiotropism

Yufei Wu[1,19], Xiaomei Yang[2,3,19], Yu Bai[4], Chenze Li[5], Peng Wang[6], Qing Xu[7], Hui Li [ID][8], Xiaoli Rao[9], Yangkai Xu[1], Jie Chen[1], Huanhuan Cao[1], Qi Zhang[1], Mingming Zhao[8], Rui Zhan[10], Xue Fan[11], Yuedong Hou[2], Jie Liu[12], Hong S Lu [ID][13], Tianyun Wang [ID][9], Wei Dong Gao[14], Linzhang Huang [ID][15], Han Xiao [ID][8], Lingyun Zu[8,16], Alan Daugherty [ID][13], Mingguo Xu [ID][17✉] & Lemin Zheng [ID][1,18✉]

## Abstract

Cardiac microvascular endothelial cells (CMECs) dysfunction is a well-recognized mediator of heart failure with preserved ejection fraction (HFpEF), but the underlying mechanism remains unclear. Here we find that scavenger receptor class B type I (SR-B1) is predominantly expressed in CMECs and decreased significantly in HFpEF. Endothelial-specific SR-B1 deficiency exacerbates cardiac pathological remodeling and diastolic dysfunction in HFpEF, which can be prevented by endothelial SR-B1 reconstitution through adeno-associated virus serotype 1 (AAV1)-mediated delivery in endothelial-specific SR-B1-deficient mice. Single-cardiac-endothelial-cell transcriptomics and lineage-tracing system reveal that inflammatory CMECs subcluster activation is responsible for the deteriorating HFpEF progression induced by endothelial SR-B1 loss, rather than endothelial-to-mesenchymal transition. Mechanistically, SR-B1 loss drives increased CXCL10 secretion, which orchestrates CMECs activation and CXCR3-positive T-cell cardiotropism to promote diastolic dysfunction—a process associated with endothelial IRF1 activation. Most importantly, the SR-B1-CXCL10-CXCR3 axis is activated in human HFpEF cardiac tissue, and the elevated CXCL10 level in plasma is independently associated with a higher HFpEF prevalence. This study uncovers that activation of the SR-B1–CXCL10–CXCR3 axis in CMECs aggravates HFpEF pathogenesis through the accumulation of CXCR3-positive T-cells in hearts.

**Keywords** Heart Failure with Preserved Ejection Fraction; Cardiac Microvascular Endothelial Dysfunction; Scavenger Receptor Class B Type I (SR-B1); Chemokine C-X-C Motif Chemokine Ligand 10 (CXCL10)
**Subject Categories** Cardiovascular System; Immunology

## Introduction

Heart failure with preserved ejection fraction (HFpEF) is the most common type of heart failure, and its incidence is increasing (Dunlay et al, 2017; Groenewegen et al, 2020). In the clinic, HFpEF is associated with comorbidities such as obesity, hypertension, diabetes mellitus (DM), systematic inflammation, cardiac arrhythmias, and impaired oxygen utilization (Gao, 2024). Yet, treatments available for HFpEF are not effective, unlike in the heart failure with reduced ejection fraction (HFrEF) population (Borlaug, 2020). Recently, sodium/glucose cotransporter 2 inhibitors have been demonstrated to be beneficial to patients with HFpEF, albeit with limited therapeutic effect (Abraham et al, 2021; Butler et al, 2022). Such findings highlight the urgent need for a better

[1]The Institute of Cardiovascular Sciences and Institute of Systems Biomedicine, School of Basic Medical Sciences, State Key Laboratory of Vascular Homeostasis and Remodeling, NHC Key Laboratory of Cardiovascular Molecular Biology and Regulatory Peptides, Beijing Key Laboratory of Cardiovascular Receptors Research, Health Science Center, Peking University, Beijing, China. [2]Department of Anesthesiology, Qilu Hospital of Shandong University, Jinan, China. [3]Department of Cardiology, Johns Hopkins University School of medicine, Baltimore, MD, USA. [4]State Key Laboratory for Innovation and Transformation of Luobing Theory, Key Laboratory of Cardiovascular Remodeling and Function Research of MOE, NHC, CAMS and Shandong Province, Department of Cardiology, Qilu Hospital of Shandong University, Jinan, China. [5]Department of Cardiology, Zhongnan Hospital of Wuhan University, Institute of Myocardial Injury and Repair, Wuhan University, Wuhan, China. [6]Cardiac Department, Aerospace Center Hospital, Peking University Aerospace School of Clinical Medicine, Beijing, China. [7]Core Facilities Centre, Capital Medical University, Beijing, China. [8]Department of Cardiology and Institute of Vascular Medicine, Peking University Third Hospital, Beijing, China. [9]Department of Medical Genetics, Center for Medical Genetics, School of Basic Medical Sciences, Peking University Health Science Center, Beijing, China. [10]Research Center for Cardiopulmonary Rehabilitation, University of Health and Rehabilitation Sciences Qingdao Hospital (Qingdao Municipal Hospital), School of Health and Life Sciences, University of Health and Rehabilitation Sciences, Qingdao, China. [11]Clinical Research Institute, Shanghai General Hospital, Shanghai Jiao Tong University School of Medicine, Shanghai, China. [12]State Key Laboratory of Cardiovascular Diseases and Medical Innovation Center, Shanghai Heart Failure Research Center, Department of Cardiovascular Surgery, Shanghai East Hospital, Tongji University School of Medicine, Shanghai, China. [13]Department of Physiology, Saha Cardiovascular Research Center and Saha Aortic Center, University of Kentucky, Lexington, KY, USA. [14]Department of Anesthesiology and Critical Care Medicine, The Johns Hopkins University School of Medicine, Baltimore, MD, USA. [15]Shanghai Key Laboratory of Metabolic Remodeling and Health, Institute of Metabolism and Integrative Biology, Fudan University, Shanghai, China. [16]Institute of Tibetan Plateau, Peking University, Beijing, China. [17]Department of Pediatrics, The Third People's Hospital of Longgang, Clinical Institute of Shantou University Medical College, Shenzhen, China. [18]Beijing Tiantan Hospital, China National Clinical Research Center for Neurological Diseases, Advanced Innovation Center for Human Brain Protection, Beijing Institute of Brain Disorders, The Capital Medical University, Beijing, China. [19]These authors contributed equally: Yufei Wu, Xiaomei Yang. ✉E-mail: docjxzhu@stu.edu.cn; zhengl@bjmu.edu.cn

understanding of the underlying mechanisms of HFpEF and effective HFpEF therapeutics (Shah et al, 2020).

Recent studies have shown that HFpEF pathogenesis is complex, involving not only cardiomyocyte dysfunction (Deng et al, 2021; Tong et al, 2021) but also non-cardiomyocytes dysregulation, including cardiac endothelial cells (Li et al, 2024; Paulus, 2020) and immune T cells (Ovchinnikov et al, 2023; Smolgovsky et al, 2023). Among non-cardiomyocytes, endothelial cells constitute the predominant cell type in the heart (Pinto et al, 2016). In patients with HFpEF, the severe dysfunction of cardiac microvascular endothelial cells (CMECs), mainly manifesting as inflammatory activation and impaired endothelial nitric oxide synthase function, has been demonstrated (Franssen et al, 2016; Kolijn et al, 2021). Furthermore, the infiltration of activated CD4 + T cells into myocardial tissue has been identified as a critical contributor to cardiac hypertrophy and diastolic dysfunction in HFpEF (Smolgovsky et al, 2023). CXCR3, a G protein-coupled receptor predominantly expressed on CD4 + Th1 cells, has been shown to facilitate CD4 + T cell recruitment into the heart, thereby inducing cardiac remodeling in HFrEF (Ngwenyama et al, 2019). CXCL10, a specific ligand for CXCR3, has been closely associated with the progression of heart failure (Altara et al, 2016). Notably, in a deoxycorticosterone acetate (DOCA)-induced HFpEF model with diastolic dysfunction, a significant increase in CXCR3 + T cell populations was observed (Smart et al, 2023). Collectively, these findings strongly suggest that CXCR3 + T cells likely play a pivotal role in the development of cardiac dysfunction in HFpEF. In summary, inflammatory activation of CMECs and immune T cell infiltration represent crucial pathological features of HFpEF; however, the regulatory mechanisms underlying their interaction remain poorly understood.

Scavenger receptor class B type I (SR-B1) is widely expressed in numerous tissues and cells, including the liver, adrenal glands, adipose cells, ECs, and monocytes (Shen et al, 2018). SR-B1 has a major role in facilitating cholesterol uptake and transportation in the liver and steroidogenic tissues (Shen et al, 2018). In addition, SR-B1 regulates peripheral EC function *via* activation of several kinases, such as Src kinases, phosphatidylinositol 3-kinase (PI3K), and protein kinase B/mitogen-activated protein kinases after cholesterol (HDL) binding. SR-B1 plays a complex role in the peripheral endothelium (Mineo and Shaul, 2007; Seetharam et al, 2006; Zhu et al, 2008). Previously, our group has shown that SR-B1 is an essential mediator of carotid endothelial repair in vivo (He et al, 2018). Global deletion of SR-B1 leads to the early onset of severe cardiac dysfunction, mainly due to occlusive atherosclerotic coronary artery (Braun et al, 2002). Nevertheless, SR-B1 potentially exerts a protective role in cardiac endothelial cells, especially during heart failure, which remains to be investigated.

Here, we demonstrate that cardiac SR-B1 is located predominantly in CMECs, and its deficiency deteriorates pathological remodeling and cardiac diastolic dysfunction severity in HFpEF. Furthermore, we demonstrate that endothelial SR-B1 deficiency promotes CMECs' inflammatory activation and enhances CXCR3 + T cell infiltration in the heart through increased CXCL10 secretion, thereby driving cardiac dysfunction in HFpEF.

# Results

## Cardiac microvascular endothelial SR-B1 contributes to HFpEF pathogenesis

Immunofluorescence analysis revealed that SR-B1 was predominantly present in mouse and human ventricular CMECs (Fig. 1A,B; Appendix Fig. S1A,B). We validated that SR-B1 was highly abundant in CD31-positive ECs isolated from mouse ventricles (Fig. 1C,D; Appendix Fig. S1C).

Single-nucleotide polymorphism rs10846742 (Rasooly et al, 2023) in *SCARB1* (SR-B1) is significantly associated with heart failure pathogenesis (Appendix Fig. S1D), indicating its potential involvement in HFpEF and HFrEF in vivo. Subsequently, endothelial SR-B1 expression was assessed during HFpEF or HFrEF pathogenesis. Firstly, SR-B1 expression decreased significantly in primary mouse cardiac ECs treated with a HFpEF-like stimulation in vitro (Appendix Fig. S1E). Consistently and importantly, SR-B1 protein abundance was decreased significantly in cardiac ECs isolated from "two-hit" HFpEF mice compared with ECs from controls (Fig. 1E). However, SR-B1 transcription in cardiac ECs of HFrEF mice induced by TAC remained unaltered (Appendix Fig. S1F). Importantly, *Scarb1* mRNA expression level in mouse cardiac ECs was sequentially decreased in parallel with the worsening of HFpEF-related cardiac dysfunction (Appendix Fig. S1G). Therefore, SR-B1 in cardiac CMECs could be involved in HFpEF pathogenesis.

## Endothelial-specific SR-B1 deficiency aggravates diastolic dysfunction in HFpEF

To directly investigate the role of endothelial SR-B1 in HFpEF pathogenesis, endothelial-specific SR-B1 knockout mice ($S^{fl/fl}$; *Cdh5-Cre$^{ER}$*: $S^{\Delta EC}$) were designed based on a tamoxifen-inducible Cre-loxP system (Fig. 2A). Briefly, $SR-B1^{flox/+}$ ($S^{fl/+}$) mice were constructed by inserting loxP sequences into the flanking introns of exon 2 (158 bp) of the mouse *Scarb1* (SR-B1) gene using CRISPR-Cas9 gene-editing technology. Subsequently, tamoxifen-inducible endothelial-specific cadherin 5 (Cdh5)-Cre$^{ER}$ mice were crossed with $S^{fl/+}$ mice to generate $S^{fl/+}$ *Cdh5-Cre$^{ER}$* mice. Then, $S^{\Delta EC}$ and littermate control $S^{fl/fl}$ mice were obtained from the cross of $S^{fl/+}$ *Cdh5-Cre$^{ER}$* mice and $S^{fl/+}$ mice, which were confirmed by DNA genotyping (Appendix Fig. S2A). In contrast to a previous report (Huby et al, 2006), this loxP-inserting strategy did not disrupt SR-B1 endogenous expression (Appendix Fig. S2B). Finally, SR-B1 abundance and knockout specificity were evaluated, demonstrating that endothelial SR-B1 was specifically deleted, with ~70% reduction in cardiac endothelial cells observed 2 weeks after tamoxifen injection (Fig. 2B,C; Appendix Fig. S2C,D).

Subsequently, we investigated the role of endothelial SR-B1 in HFpEF mice (i.e., $S^{fl/fl}$ HFpEF and $S^{\Delta EC}$ HFpEF mice). After 10 weeks of HFD plus L-NAME regimens, both $S^{fl/fl}$ HFpEF and $S^{\Delta EC}$ HFpEF mice exhibited significant left ventricular hypertrophy and diastolic dysfunction with normal fractional shortening (FS) compared with the control $S^{fl/fl}$ and $S^{\Delta EC}$ mice, confirming the successful establishment of our HFpEF model (Appendix Fig. S3A–G). Notably, compared to $S^{fl/fl}$ HFpEF mice, $S^{\Delta EC}$ HFpEF mice exhibited a consistent trend toward more severe cardiac hypertrophy (as

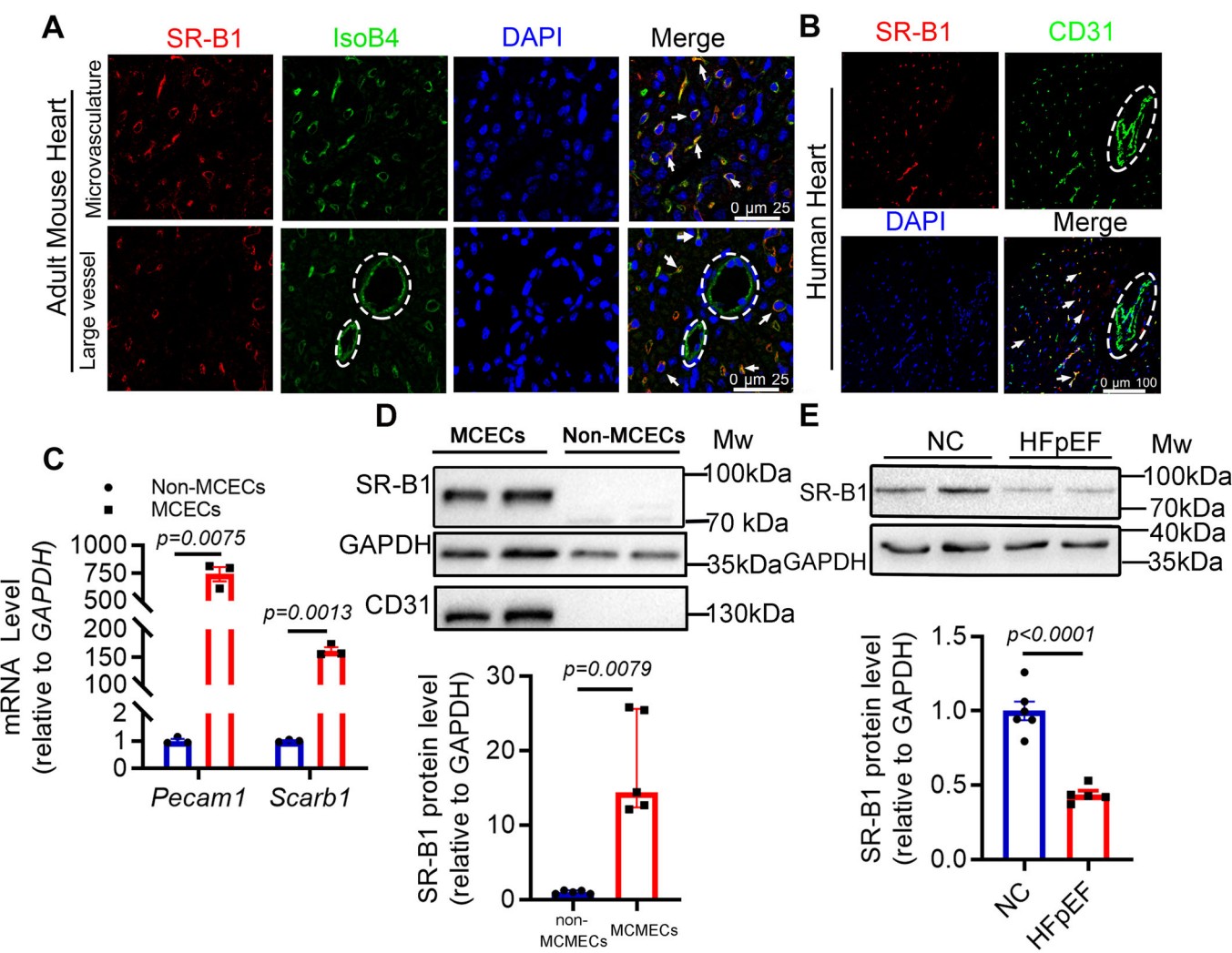

**Figure 1. SR-B1 is predominantly present in cardiac microvascular endothelial cells and significantly decreased in the HFpEF heart.**

(A) Immunofluorescence staining of SR-B1 in the mouse cardiac left ventricle co-stained with fluorescence-conjugated isolectin B4 (IsoB4) to label endothelial cells and DAPI to label nuclei. Scale bar, 25 µm. Arrow, SR-B1/IsoB4 co-localization. Dashed ellipse: vascular lumen. (B) Double-staining of SR-B1 and CD31 in the cardiac left ventricle of a healthy human. Scale bar, 100 µm. Arrow, SR-B1/CD31 co-localization. Dashed ellipse: vascular lumen. (C) *Scarb1* (SR-B1) and *Pecam1* (CD31) mRNA abundance was determined in adult mouse cardiac endothelial cells (MCECs) and non-MCECs by RT-qPCR; *Pecam1* transcript level served as quality control of purified MCECs (n = 3). Statistical significance was evaluated by Welch's *t* test. (D) Upper panel, SR-B1 protein abundance was determined in adult mouse cardiac endothelial cells (MCECs) and non-MCECs by immunoblotting; CD31 abundance served as quality control of purified MCECs; GAPDH as a loading control. Lower panel, Quantification of SR-B1 protein abundance in MCECs and non-MCECs (n = 5). Statistical significance was evaluated by Mann–Whitney test. (E) Relative SR-B1 protein abundance in MCECs isolated from 20 weeks of two-hit HFpEF mice and control mice was measured by immunoblotting (upper panel) and was statistically analyzed *via* Student's *t* test (lower panel, n = 6, 5 independent experiments using pooled samples, with each data point representing 2–3 mouse hearts). All numerical data were presented as mean ± SEM. Source data are available online for this figure.

evidenced by LV mass and left ventricular remodeling index (LVRI) parameters) and worse diastolic dysfunction (indicated by myocardial performance index (MPI) and E/e' ratio), but these differences did not reach statistical significance.

Remarkably, upon extending the HFD and $_L$-NAME regimen to 20 weeks, $S^{\Delta EC}$ HFpEF mice exhibited significantly exacerbated cardiac hypertrophy and diastolic function than $S^{fl/fl}$ HFpEF mice, suggesting faster deleterious disease progression in the absence of endothelial SR-B1. Specifically, echocardiography showed significantly increased left ventricular diastolic posterior wall thickness (LVPWd), LV mass, LVRI, MPI, and E/e' ratio with normal FS (Fig. 2D–M). In addition, $S^{\Delta EC}$ HFpEF mice ran significantly short

distances during stress exercise testing (Fig. 2N). The data show that endothelial SR-B1 loss accelerated the adverse HFpEF remodeling and progression.

## Endothelial-specific SR-B1 deletion worsens cardiac pathological remodeling in HFpEF, whereas SR-B1 restoration alleviates dysfunction and prevents adverse remodeling in SR-B1-deficient mice

Current research demonstrates that global SR-B1 deficiency robustly increases plasma HDL cholesterol (HDL-C) and total cholesterol (TC) concentration and causes cardiac dysfunction

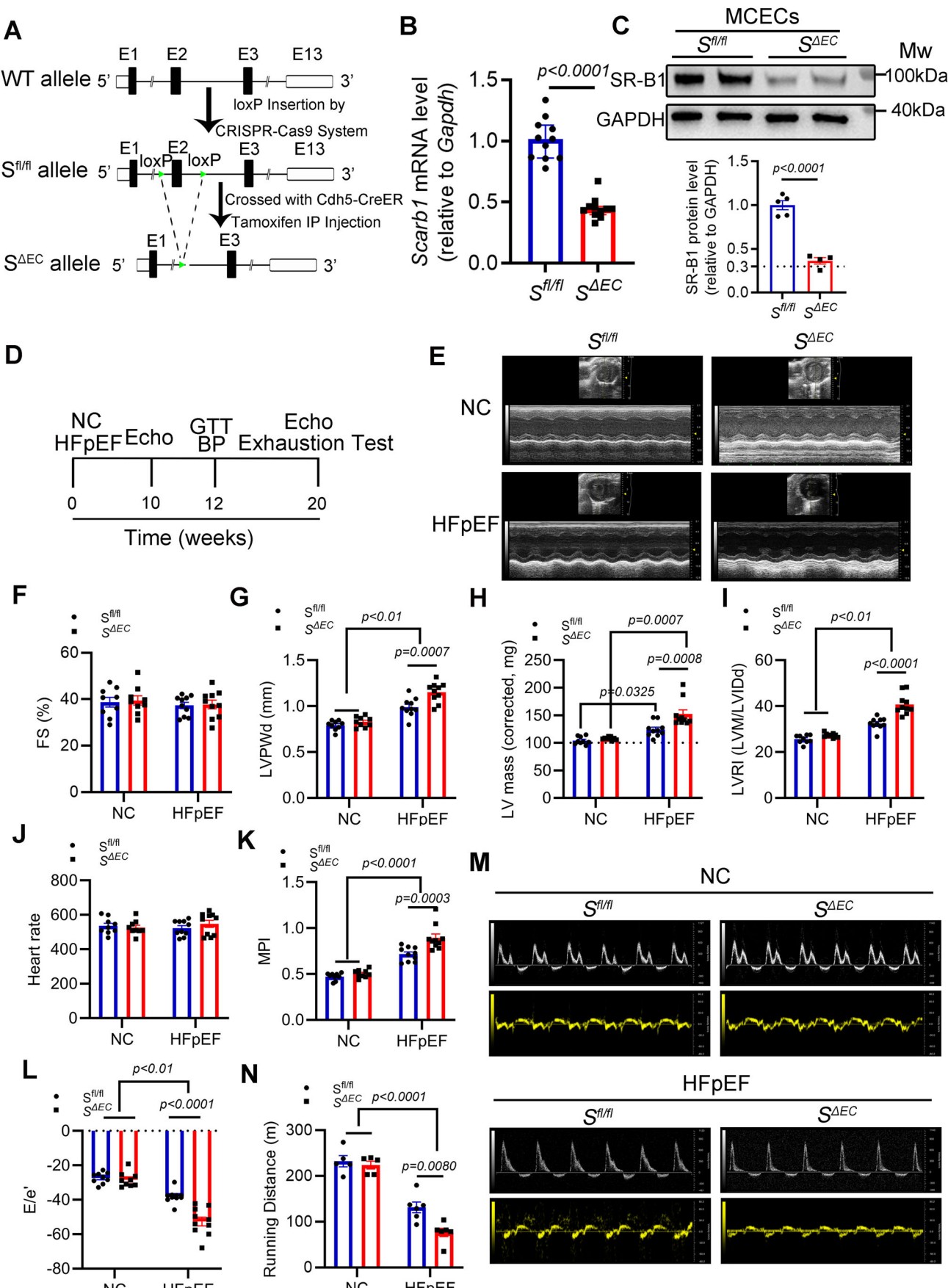

Figure 2. Endothelial-restricted SR-B1 deletion drives cardiac diastolic dysfunction in HFpEF.

(A) A schematic overview of the CRISPR-based genome-editing strategy used to generate the SR-B1 deficiency mice. (B) *Scarb1* (SR-B1) mRNA abundance was assessed in the cardiac ventricle of $S^{fl/fl}$ and $S^{\Delta EC}$ mice 2 weeks post-tamoxifen administration ($n = 11$, Mann–Whitney test). (C) SR-B1 protein abundance was evaluated in mouse cardiac endothelial cells (MCECs) isolated from $S^{fl/fl}$ mice and $S^{\Delta EC}$ mice 2 weeks after tamoxifen administration by immunoblotting and statistically evaluated *via* Student's *t* test ($n = 5, 4$). (D–L) $S^{fl/fl}$ mice and $S^{\Delta EC}$ mice were fed with a normal diet or HFD plus ʟ-NAME regimens for 20 weeks to investigate the effect of SR-B1 deletion on cardiac function of HFpEF mice ($n = 9, 9, 10, 10$; statistical differences were evaluated by two-way ANOVA followed by Bonferroni post-hoc test). (D) Workflow of in vivo experiments. (E) Representative M-mode images in the parasternal short-axis view. (F) FS, left ventricular fractional shortening. (G) LVPWd, left ventricular diastolic posterior wall thickness. (H) LV mass, left ventricular mass. (I) LVRI, left ventricular remodeling index. (J) Heart rate in parasternal short-axis view. (K) MPI, myocardium performance index, also denoted the Tei index, a fairly new index of combined systolic and diastolic function, is defined as the sum of isovolumic contraction time (IVCT) and isovolumic relaxation time (IRCT) divided by the ejection time (ET). (L) E/e', ratio between mitral E wave and e' wave. (M) Representative pulsed-wave Doppler waveform of mitral valve leaflet tips (top) and tissue Doppler tracing of the mitral valve measured on the side of the left ventricular wall (bottom). (N) Running distance during exercise exhaustion test ($n = 5, 5, 6, 6$; statistical differences were evaluated by two-way ANOVA followed by Bonferroni post-hoc test). All numerical data were presented as mean ± SEM. Source data are available online for this figure.

under high LDL cholesterol concentrations. (Rigotti et al, 1997). However, we observed that loss of endothelial-specific SR-B1 did not affect plasma HDL-C, TC, and triglyceride (TG) concentrations in $S^{\Delta EC}$ compared to littermate control $S^{fl/fl}$ mice (Appendix Fig. S3H), consistent with a recent report (Huang et al, 2019). In addition, $S^{\Delta EC}$ and $S^{fl/fl}$ mice from the NC or HFpEF group exhibited similar body weight, blood pressure (hypertension), and glucose tolerance capability (obesity) (Appendix Fig. S3I–M). The data show that the worst cardiac diastolic dysfunction in $S^{\Delta EC}$ mice was unlikely due to systemic substrate metabolism (cholesterol, lipid, and glucose) and blood pressure.

To further demonstrate the impact of endothelial-specific SR-B1 deficiency on cardiac structure and function in HFpEF, histological and immunohistological analyses of hearts were performed after a 20-week dietary intervention. The results revealed that, compared to $S^{fl/fl}$ mice, $S^{\Delta EC}$ HFpEF mice had significantly increased ventricular weight to tibial length (Ventricle/TL) and lung wet weight to tibial length (LW/TL) ratios (Fig. 3A,B). In addition, the extent of cardiac fibrosis in $S^{\Delta EC}$ HFpEF mice was significantly increased compared with the hearts from littermate controls (Fig. 3C–F). Therefore, the loss of endothelial SR-B1 promoted cardiac pathological remodeling in HFpEF.

To determine whether the observed phenotype could be rescued through SR-B1 restoration in cardiac endothelial cells, we employed an adeno-associated virus 1 (AAV1)-mediated endothelial delivery system. Based on previous literature, we preselected four AAV serotypes for initial screening: AAV1 (Chen et al, 2005), AAV2 (Qi et al, 2012), AAV9, and the mutant RGDLRVS-AAV9 (Huang et al, 2021). For endothelial-specific expression, we utilized the Cdh5 promoter, a well-established endothelial cell-specific marker (Appendix Fig. S4A). Through immunofluorescence and Western blot analyses 3 weeks after post-injection virus, we identified that the AAV1 serotype combined with the Cdh5 promoter construct demonstrated optimal transduction efficiency and specificity for subsequent gene delivery experiments (Appendix Fig. S4B–E).

$S^{\Delta EC}$ and littermate control $S^{fl/fl}$ mice were subjected to HFD plus ʟ-NAME for 10 weeks, followed by tail vein injection of AAV1-Cdh5 expressing zsGreen or SR-B1 (Fig. 3G). Ten weeks post-injection, we conducted comprehensive cardiac functional assessments through echocardiography and exercise tolerance tests. The results demonstrated that SR-B1 restoration significantly ameliorated the exacerbated cardiac hypertrophy, diastolic dysfunction, and exercise-induced fatigue caused by SR-B1 deficiency (Fig. 3H–K). Furthermore, and of particular importance,

pathological examinations and RT-qPCR analyses confirmed that SR-B1 reconstitution markedly suppressed the progression of cardiac hypertrophy and cardiac fibrosis in SR-B1-deficient mice (Fig. 3L,M). In summary, these findings collectively demonstrate that AAV1-mediated SR-B1 restoration effectively rescues the pathological phenotypes associated with endothelial SR-B1 deficiency.

## Endothelial SR-B1 is dispensable for the pathogenesis of acute HFrEF induced by transverse aortic constriction (TAC) surgery

To investigate the role of endothelial SR-B1 in HFrEF, $S^{\Delta EC}$ and $S^{fl/fl}$ control mice were subjected to TAC surgery. Blood pressure at the aortic arch ligation site was measured using an ultrasonic flow Doppler module, which verified that both TAC groups ($S^{fl/fl}$ TAC and $S^{\Delta EC}$ TAC mice) displayed equivalent pressure gradients across the constriction site (Appendix Fig. S5A). Two weeks post-surgery, both TAC mice groups developed severe dilated heart failure with systolic dysfunction, as evidenced by a significant reduction in FS (mean FS < 20%, Appendix Fig. S5C) and increased LVIDd (Appendix Fig. S5F). Notably, echocardiographic analysis revealed that endothelial SR-B1 deficiency did not alter the progression of cardiac systolic dysfunction or remodeling in this acute HFrEF model (Appendix Fig. S5B–I). Furthermore, pathological examinations and histological analyses demonstrated that SR-B1 knockout did not significantly affect TAC-induced cardiac hypertrophy (Appendix Fig. S5K), pulmonary edema (Appendix Fig. S5L), or fibrosis (Appendix Fig. S5M and 5N). Molecular marker analysis *via* RT-qPCR, including *Bnp*, *Col3a1*, and *Col1a1*, confirmed no significant differences in fibrosis or hypertrophy markers between the two HFrEF groups (Appendix Fig. S5O).

To definitively evaluate the temporal impact of SR-B1, we performed TAC on a new batch of endothelial-specific SR-B1 KO mice. Echocardiography at weeks 2, 4, 6, and 8 post-surgery revealed no significant differences in cardiac function between genotypes (Appendix Fig. S5P–S). Survival analysis showed that mice began to die from the 2nd week post-surgery, with the mortality rate reaching up to 30% by the 8th week (Appendix Fig. S5T). In light of this increased mortality risk, the mice were euthanized. Pathological examination revealed that SR-B1 deficiency did not significantly alter cardiac hypertrophy or pulmonary edema (Appendix Fig. S5U,V).

In summary, endothelial SR-B1 deficiency does not appear to influence disease progression in acute HFrEF induced by TAC.

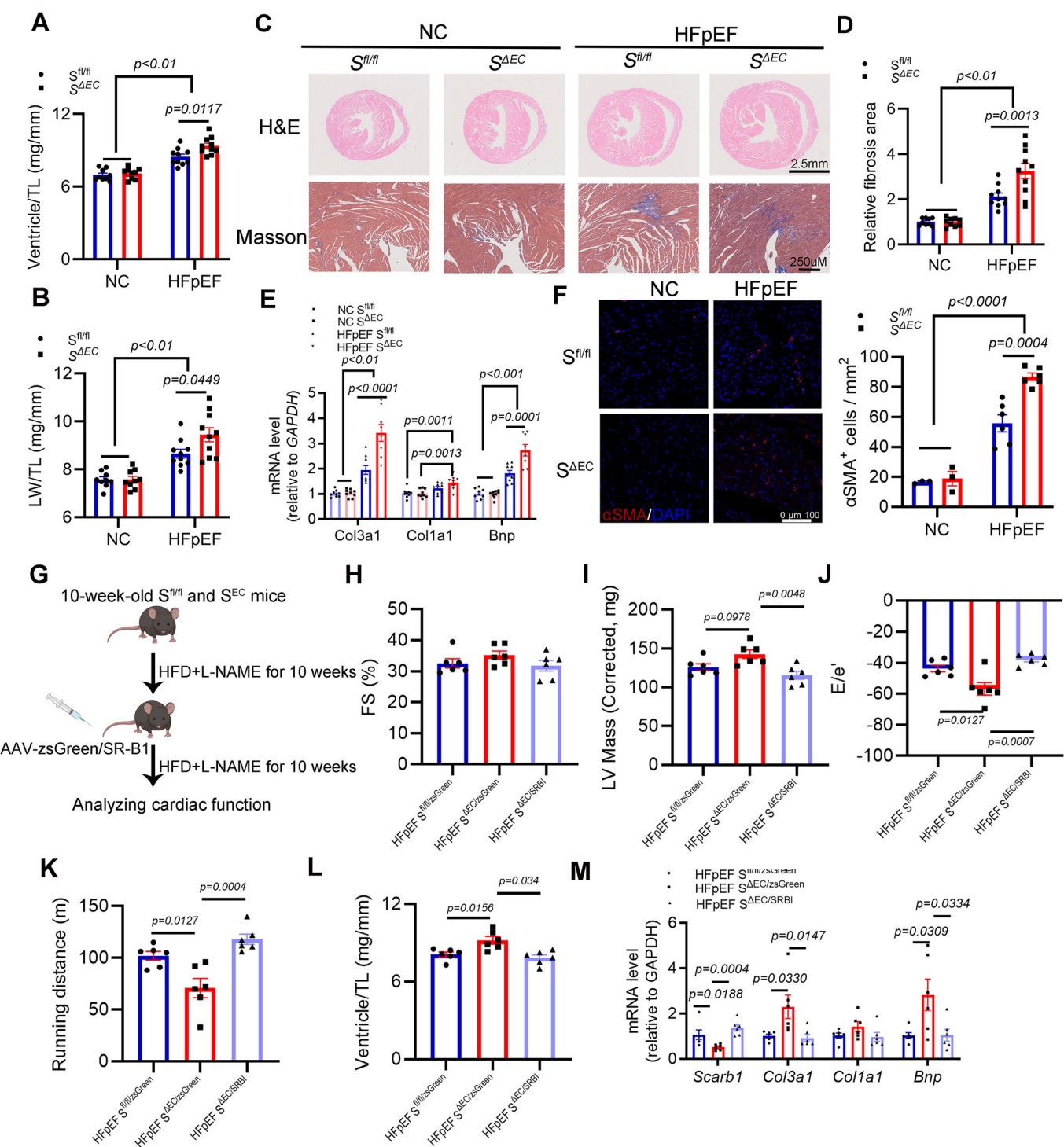

### Lineage-tracing system reveals that endothelial-to-mesenchymal transition does not contribute to HFpEF cardiac diastolic dysfunction aggravated by endothelial SR-B1 loss in vivo

One key feature of HFpEF is systemic inflammation with high concentrations of IL-1β, IL-6, and tumor necrosis factor-alpha (TNF-α) (Paulus, 2020). Since endothelial-to-mesenchymal transition

(EnMT) in vivo is also driven by similar proinflammatory factors (Bischoff, 2019; Evrard et al, 2016; Kovacic et al, 2019), we were interested in investigating whether adult mouse cardiac ECs possess EnMT potential and contribute to fibrosis development in HFpEF, and importantly, if it is driven by endothelial SR-B1 loss. Cell morphology of primary adult mice cardiac ECs isolated from wild-type mice changed dramatically when they were stimulated with TGF-β1 (Fig. 4A), along with significant downregulation of the endothelial

◀ **Figure 3.  Endothelial-specific SR-B1 deficiency aggravates cardiac pathological remodeling, and AAV-mediated restoration of SR-B1 expression in $S^{\Delta EC}$ mice effectively ameliorates cardiac dysfunction and remodeling induced by SR-B1 deletion.**

(A–F) Cardiac pathological examinations were conducted in $S^{fl/fl}$ and $S^{\Delta EC}$ mice euthanized after 20 weeks of either a normal diet or HFD plus L-NAME (A two-way ANOVA followed by Bonferroni post-hoc test was applied to assess the significant difference). (A) Ratio between cardiac ventricle weight and tibia length (TL) in $S^{fl/fl}$ and $S^{\Delta EC}$ mice euthanized after 20 weeks of normal diet or HFD plus L-NAME ($n = 9, 9, 10, 10$). (B) Ratio between wet lung weight and TL ($n = 9, 9, 10, 10$). (C) Representative images of hematoxylin & eosin (H&E; scale bar, 2.5 mm) and Masson's trichrome (MT; scale bar, 250 μm) staining in transversal sections of heart. (D) Fibrosis area in Masson's trichrome-stained transversal sections was analyzed ($n = 9, 9, 10, 10$). (E) Col3a1, Col1a1, and Bnp mRNA abundance in the left cardiac ventricle were measured via RT-qPCR ($n = 9, 9, 9, 8$). (F) Representative images of immunofluorescence staining of αSMA to label myofibroblast co-stained with DAPI in the left cardiac ventricle of $S^{fl/fl}$ and $S^{\Delta EC}$ mice (left panel; scale bar, 100 μm); αSMA-positive cell per mm$^2$ was counted (right panel, $n = 3, 3, 6, 6$). (G–M) Endothelial SR-B1 gain-of-function, achieved through AAV1 tail vein injection in $S^{\Delta EC}$ and $S^{fl/fl}$ mice, was evaluated using echocardiography and pathological analysis ($n = 6$; statistical analyses were performed using one-way ANOVA with Bonferroni's post-hoc test or Kruskal–Wallis test with Dunn's post-hoc test). (G) Schematic illustration of the experimental design. (H) FS. (I) LV mass. (J) E/e' ratio. (K) Running distance. (L) Ratio between cardiac ventricle weight and TL. (M) The mRNA expression levels of Scarb1, Col3a1, Col1a1, and Bnp in the left ventricle were quantified using RT-qPCR. All numerical data were presented as mean ± SEM. Source data are available online for this figure.

marker gene *Cdh5* and upregulation of major characteristic genes of myofibroblasts, such as *Col1a1*, *Col3a1*, and *Acta2* (αSMA) (Fig. 4B). When co-incubated with proinflammatory factors $H_2O_2$ or IL-1β and TGF-β1, adult cardiac ECs exhibited significantly decreased expression of endothelial markers, including CD31 and endothelial nitric oxide synthase (eNOS) (Fig. 4B). Notably, IL-1β, not $H_2O_2$, suppressed the transcriptional upregulation of characteristic myofibroblast markers, including *Acta2* and *Col3a1*, compared with that obtained with incubating TGF-β1 alone. Moreover, TGF-β1 treatment, either alone or in combination with H2O2 or IL-1β, significantly reduced *Scarb1* (SR-B1) transcription. The in vitro findings suggest that adult cardiac ECs possess the potential for EnMT, and SR-B1 is likely involved in this process.

Subsequently, we examined EnMT in HFpEF in vivo. Cells expressing both endothelial marker CD31 and myofibroblast marker αSMA were not detected in cardiac fibrotic regions using double-labeling immunofluorescence (Fig. 4C). Furthermore, a triple-transgenic and lineage-tracing mouse model was constructed to visually trace the potential EnMT fate of cardiac ECs in HFpEF, and to investigate whether SR-B1 participates in the EnMT process (Fig. 4D). In both $S^{fl/fl}/R^{\Delta EC}$ mice and control $S^{fl/+}/R^{\Delta EC}$ mice, cardiac endothelial cells would exhibit sustained and specific expression of red fluorescent protein (RFP) following tamoxifen injection, accompanied by the endothelial-specific knockout of SR-B1. Subsequently, immunofluorescence analysis revealed that RFP expression was exclusively localized to cardiac CD31-positive endothelial cells (ECs) and absent in platelet-derived growth factor receptor α (PDGFRα)-positive cardiac fibroblasts 2 weeks post-tamoxifen injection in healthy control mice (NC $S^{fl/+}/R^{\Delta EC}$) (Fig. 4E). Notably, in lineage-tracing $S^{fl/+}/R^{\Delta EC}$ mice subjected to 20 weeks of HFD plus L-NAME regimens, no co-localization of αSMA and RFP-positive cells was observed in cardiac tissues (Fig. 4F). Next, fluorescence-activated cell sorting (FACS) was used to further confirm the absence of EnMT in the HFpEF heart. Consistent with the immunofluorescence co-localization data, RFP-positive but CD31-negative cells were undetectable in the hearts of both HFpEF groups (Fig. 4G; Appendix Fig. S6A). Thus, EnMT is not responsible for cardiac fibrosis development in HFpEF mice, even in the absence of SR-B1.

## Endothelial SR-B1 deficiency leads to an increase in the inflammatory CMECs subcluster with high expression of CXCL10 revealed using single-cardiac-endothelial-cell RNA sequencing (scecRNA-seq), accompanied by CXCR3-positive T-cell cardiotropism

To investigate the molecular mechanism *via* which endothelial SR-B1 deficiency aggravated adverse cardiac remodeling and diastolic

dysfunction in HFpEF, scecRNA-seq was performed to characterize the transcriptional profiles of ECs purified from triple-transgenic mice (Fig. 5A). First, cardiac ECs were classified into capillary ECs, artery ECs, vein ECs, and lymphatic ECs based on a previously established single-cell transcriptomic atlas (Kalucka et al, 2020) (Fig. 5B; Appendix Fig. S7A). Heatmap analysis of the top 10 markers revealed distinct signatures for the four EC subclusters (Appendix Fig. S7B). Next, we conducted quantitative analysis of *Scarb1* (SR-B1) expression across cardiac endothelial cell subclusters from healthy control mice (NC $S^{fl/+}/R^{\Delta EC}$). The results demonstrated that *Scarb1* was predominantly and significantly expressed in cardiac capillary endothelial cells (Appendix Fig. S7C,D), which was consistent with histological observations. Furthermore, we analyzed *Scarb1* expression in capillary ECs from four group mice (Appendix Fig. S7E). The results demonstrated reduced *Scarb1* expression in double-floxed mice compared to single-floxed controls (NC $S^{fl/fl}/R^{\Delta EC}$ vs NC $S^{fl/+}/R^{\Delta EC}$ or HFpEF $S^{fl/fl}/R^{\Delta EC}$ vs HFpEF $S^{fl/+}/R^{\Delta EC}$), confirming successful *Scarb1* knockout.

Because of the dominant expression of SR-B1 in CMECs, the capillary EC cluster was selected to evaluate further. Hierarchical (unsupervised) clustering identified nine subclusters that were projected onto a two-dimensional uniform manifold approximation and projection (UMAP) plot (Fig. 5C). Composition analysis revealed that the proportion of capillary subcluster 0 increased in the HFpEF group, driven by the loss of endothelial SR-B1 (Fig. 5D); subcluster 3 exhibited the opposite effect (Fig. 5D). The heatmap of enriched Kyoto Encyclopedia for Genes and Genomes (KEGG) pathways showed that signature marker genes in subcluster 0 were associated with proinflammatory pathways, including TNF-, IL-17-, and NF-kappa B-signaling pathways, whereas theses pathways were relatively inactive in subcluster 3 (Appendix Fig. S7F). The top differentially expressed genes in subcluster 0 mainly included chemokines, such as *Cxcl10*, *Cxcl2*, and *Cxcl1* (Fig. 5E). Moreover, the top three differential pathways activated in subcluster 0 were closely related to high expression of these chemokines (Appendix Fig. S7G). Therefore, we propose that SR-B1 deletion drives activation of the inflammatory CMEC subcluster, likely by directly upregulating key chemokine concentrations.

To test this hypothesis, we quantified the transcript levels of *Cxcl10*, *Cxcl1*, and *Cxcl2* in heart tissues from HFpEF mice. The analysis revealed Cxcl10 was the differentially expressed gene with the highest fold change compared to Cxcl1 and Cxcl2 (Fig. 5F). Similarly, CXCL10 protein was significantly upregulated in $S^{\Delta EC}$ HFpEF hearts compared to $S^{fl/fl}$ HFpEF hearts (Fig. 5G). In addition, the mRNA and protein abundances of CXCR3, a specific

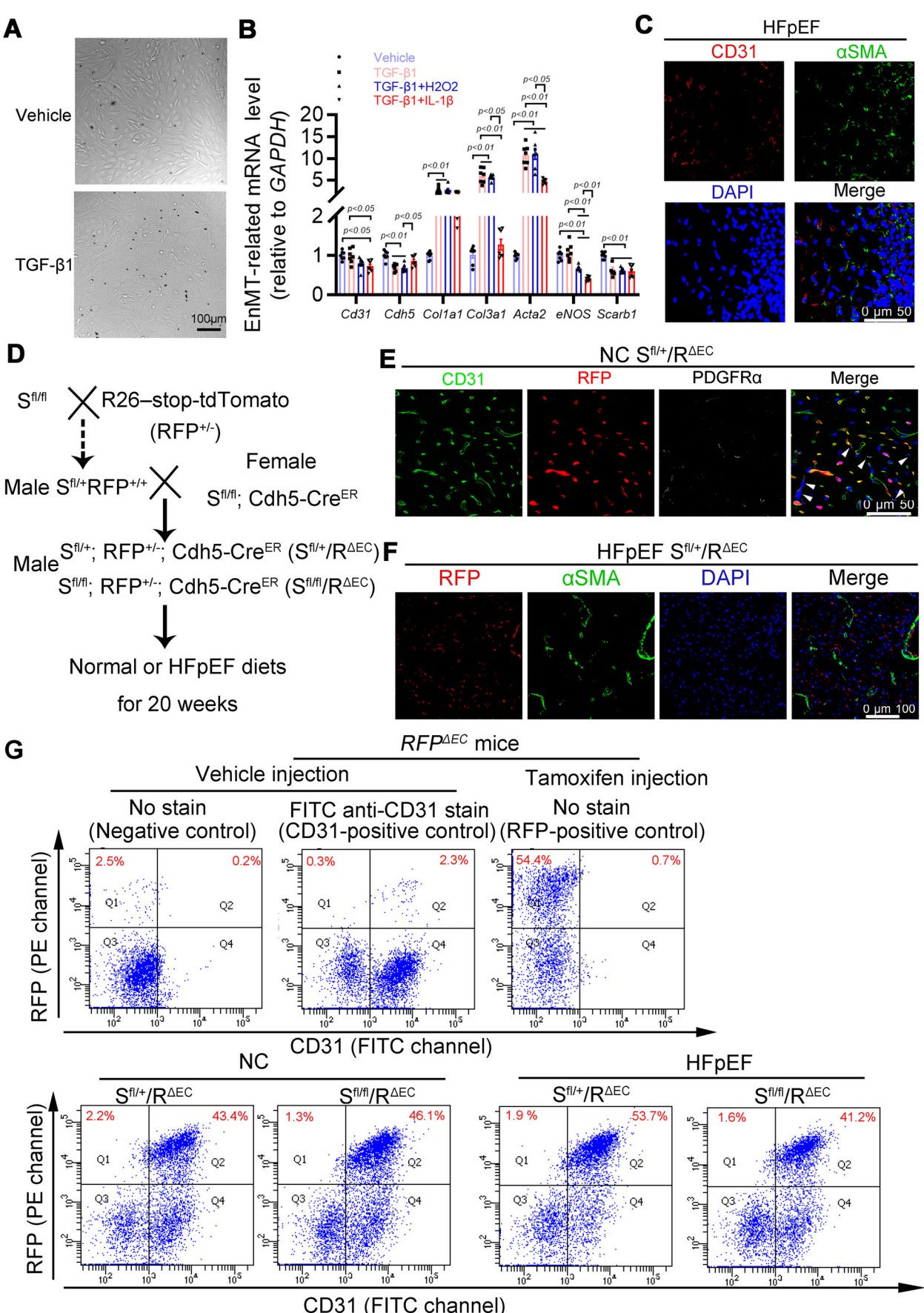

◀ **Figure 4.  Genetic lineage tracing of endothelial cells in vivo reveals that cardiac endothelial cells cannot transdifferentiate into myofibroblast in HFpEF.**

(A) Representative bright-field images of MCECs incubated with vehicle or 10 ng/mL TGF-β1, captured on day 4, are presented. Scale bar, 100 μm. (B) EnMT-related genes and *Scarb1* (SR-B1) mRNA abundance were measured in MCECs isolated from wild-type mice following stimulation with different factors *via* RT-qPCR (n = 7). Statistical analyses were performed using distinct approaches based on data distribution characteristics: *Col1a1* and *Col3a1* expression data were analyzed using the Kruskal–Wallis test followed by Dunn's multiple comparisons test; *Acta2* (αSMA) expression was evaluated using Brown–Forsythe and Welch ANOVA tests with Dunnett's T3 multiple comparisons; and the remaining genes were assessed using ordinary one-way ANOVA with Bonferroni's multiple comparisons correction. (C) Immunofluorescent staining of CD31 and αSMA to investigate the potential ability of MCECs in the cardiac fibrosis area of HFpEF mice. Scale bar, 50 μm. (D) Workflow of experiment to generate endothelial-specific SR-B1-deficient and RFP overexpression mice (a lineage tracing mouse model), investigating the in vivo transdifferentiating fate of endothelial cells in HFpEF in the presence or absence of endothelial SR-B1. (E) Immunofluorescent imaging of CD31 and PDGFRα (fibroblast marker protein) signals co-imaged with red fluorescent protein (RFP) in the cardiac ventricle of $S^{fl/+}/R^{\Delta EC}$ mice fed a normal diet (NC) to confirm the successful construction of lineage-tracing mice. Scale bar, 50 μm. (F) Fluorescent images after immunofluorescence staining of αSMA in the cardiac ventricle of $S^{fl/+}/R^{\Delta EC}$ mice fed HFD plus ʟ- NAME. Scale bar, 100 μm. (G) Single cells from lineage-tracing mice of different groups were FACS sorted according to RFP and CD31 expression. Top panel: negative controls from a heart of $RFP^{\Delta EC}$ mice not injected with tamoxifen (left panel); CD31-positive controls from a heart of $RFP^{\Delta EC}$ mice not injected with tamoxifen (middle panel); RFP-positive controls from a heart of $RFP^{\Delta EC}$ mice 2 weeks after tamoxifen injection (right panel). Bottom panel, double-channel signals were presented from isolated left ventricular cells in mice from four different groups. RFP signal was detected with PE channel (y axis), while CD31 protein expression was measured through FITC channel (x axis). All numerical data were presented as mean ± SEM. Source data are available online for this figure.

receptor for CXCL10, were also significantly higher in $S^{\Delta EC}$ HFpEF hearts than in $S^{fl/fl}$ HFpEF hearts (Fig. 5F,G).

Next, our immunofluorescent staining results confirmed that CXCR3 is predominantly expressed in T cells in the heart (Appendix Fig. S8A), consistent with a single-cell dataset (Appendix Fig. S8B). Furthermore, immunohistochemical staining revealed a significant accumulation of CXCR3-positive cells in the left ventricle of $S^{\Delta EC}$ HFpEF mice compared to their $S^{fl/fl}$ HFpEF counterparts (Fig. 5H). To further confirm that SR-B1 deficiency promotes CXCR3 + T cell infiltration in the heart, flow cytometry analysis was performed (the gating strategy is presented in Appendix Fig. S6B). The results demonstrated that SR-B1 deficiency increased the CXCR3 + T cell/CD3 + T cell ratio (Fig. 5I,J), and enhanced the infiltration of both CXCR3 + T cells (Fig. 5K) and total CD3 + T cells (Appendix Fig. S8C) in HFpEF cardiac left ventricle, without affecting the populations of CD45+ and CD11b+ cells (Appendix Fig. S8C).

Previous studies have demonstrated that T cell infiltration contributes to cardiac hypertrophy and dysfunction, involving peripheral CD4 + T cell expansion (Smolgovsky et al, 2023). However, whether systemic CXCR3 + T cells are also expanded in HFpEF remains unclear. To address this, we conducted flow cytometry analysis on cardiac tissues and blood samples from HFpEF mice and their controls after 10 weeks of dietary intervention (the gating strategy for immune cells of heart or blood is presented in Appendix Fig. S6B,C). As shown in Appendix Fig. S8D,E, in addition to the increased numbers of CD45 +, CD11b +, and CD3+ cells, CXCR3 + T cells were significantly elevated in both cardiac tissues and peripheral blood of HFpEF mice. Notably, while the proportion of CXCR3 + T cells within the total T-cell population was markedly increased in cardiac tissues, no such change was observed in peripheral blood (Appendix Fig. S8D,E). This discrepancy suggests that the CXCR3-positive T-cells cardiotropism may be regulated by the spatiotemporal expression of endogenous chemokines within the local myocardial microenvironment, potentially including CXCR3 ligands such as CXCL9, CXCL10, and CXCL11.

Based on these findings, we propose that CXCL10, produced as a result of inflammatory activation in the cardiac microvascular endothelial cell (CMEC) subcluster driven by endothelial SR-B1 deletion, acts as a critical mediator of increased diastolic dysfunction through CXCR3 + T cell infiltration in the heart.

## The activation of IRF1 likely responsible for CXCL10 upregulation by SR-B1 deletion

We attempted to link SR-B1 and CXCL10 in primary human CMECs (HCMECs). In HCMECs with SR-B1 deficiency, *CXCL10* transcript was upregulated with a greater fold change than *CXCL1* and *CXCL2* (Fig. 6A). Previously, it has been reported that obesity, as a key comorbidity of HFpEF, causes systemic chronic inflammation (Paulus and Tschope, 2013), with significantly high levels of endotoxin (Bowser et al, 2020; Ding et al, 2020). To determine the mechanisms underlying the inverse SR-B1/CXCL10 relationship, the endotoxin lipopolysaccharide (LPS) was used to induce the inflammatory activation of HCMECs, and CXCL10 was also upregulated significantly in HCMECs with SR-B1 knockdown when treated with LPS (Fig. 6B,C). Furthermore, reconstitution of SR-B1 in HCMECs reduced the CXCL10 protein abundance (Fig. 6D). The findings indicate that the presence of SR-B1 prevents CXCL10 expression, thus inhibiting inflammation activation.

We next explored the molecular mechanisms underlying SR-B1-mediated CXCL10 regulation using the PROMO tool v2.0 to predict the potential transcription factors that bind to the *CXCL10* promoter. The promoter sequence (Appendix Fig. S9A), identified through the Ensemble website, was uploaded to PROMO for analysis, and the results are presented in Appendix Fig. S9B. Notably, IRF1 ranked ninth among the predicted transcription factors but emerged as the top-ranked marker gene in the proinflammatory subcluster 0 of capillary endothelial cells (Fig. 5E). Subsequently, chromatin immunoprecipitation followed by qPCR (Chip-qPCR) confirmed that IRF1 can bind to the *CXCL10* promoter (Fig. 6E,F). A dual-luciferase assay further demonstrated that *CXCL10* promoter-driven signals were increased significantly in IRF1-overexpressing cells compared to the control (Fig. 6G). Furthermore, immunofluorescent staining indicated predominant nuclear localization of IRF1 in HCMECs, with intensified signal upon SR-B1 deficiency (Fig. 6H). Therefore, we hypothesized that IRF1 directly mediates the transcriptional upregulation of CXCL10 under conditions of SR-B1 deficiency. An in vitro rescue assay verified that IRF1 is required for CXCL10 protein upregulation following SR-B1 loss (Fig. 6I).

Previous studies have established a causative role for elevated IRF1 in heart failure pathogenesis (Huang et al, 2020; Jiang et al,

                                                             

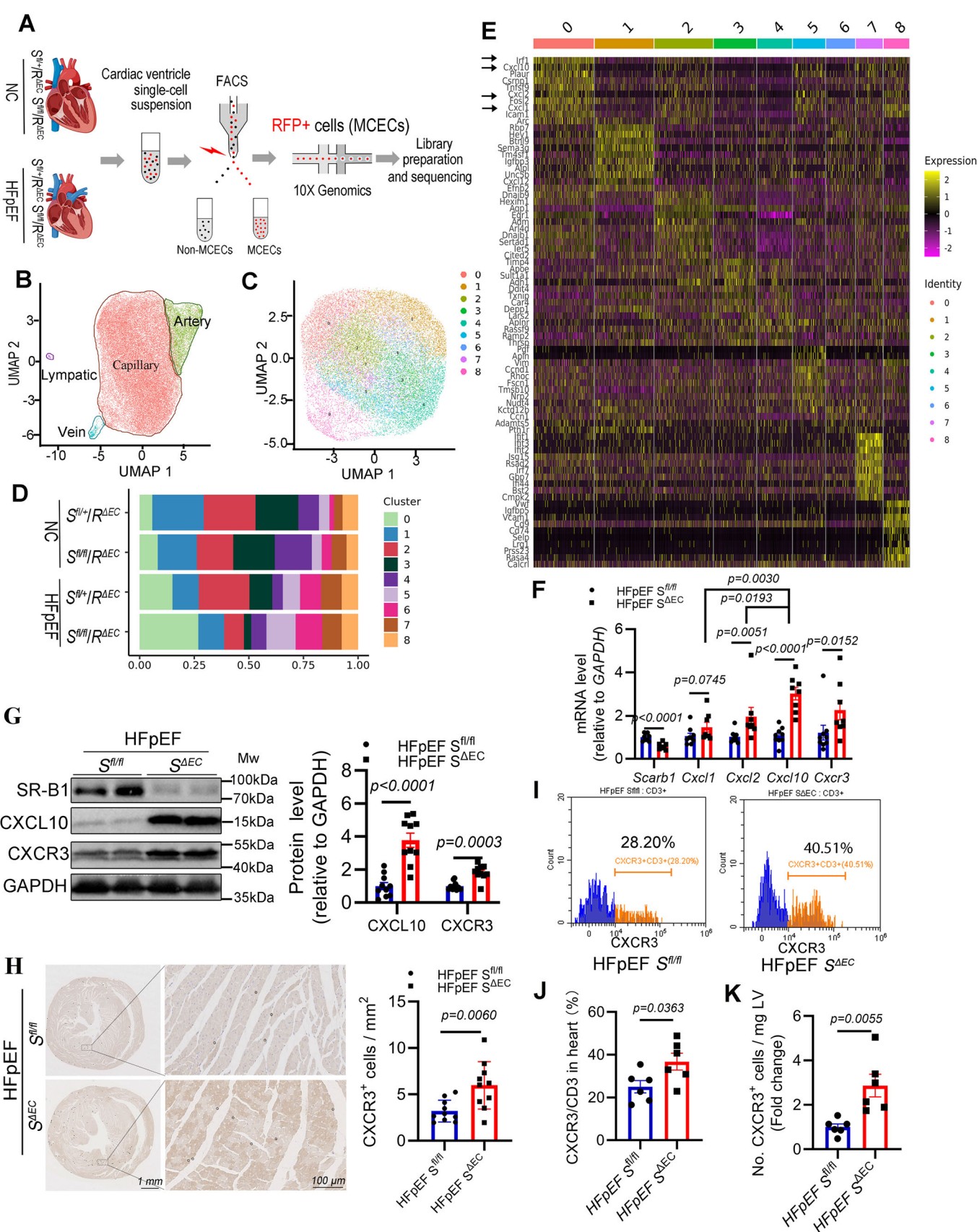

**Figure 5.   Single-cell RNA sequencing reveals that endothelial SR-B1 deletion promotes inflammatory activation of the cardiac microvascular endothelial cell, which is confirmed by increasing cardiac abundance of CXCL10 and CXCR3-positive T cell infiltration in the left cardiac ventricle.**

(A) A schematic diagram of a single-cell transcriptomic sequencing experiment. (B) Projection of cardiac ECs onto UMAP. (C) UMAP visualization of cardiac capillary ECs. (D) The comparison of the percentage of cardiac capillary EC subclusters from left cardiac ventricles of different groups. (E) Heatmap depicts expression levels of differentially expressed marker genes in subclusters of cardiac microvascular ECs. Yellow represents high gene expression, and purple represents low gene expression. (F) *Scarb1* (SR-B1), *Cxcl1*, *Cxcl2*, *Cxcl10*, and *Cxcr3* mRNA abundance were measured in left ventricle of $S^{fl/fl}$ and $S^{\Delta EC}$ HFpEF mice ($n = 9, 8$, the fold change differences between *Cxcl10* and *Cxcl1* or *Cxcl2* were analyzed using a two-way ANOVA followed by Bonferroni post-hoc test, while the remaining comparisons were assessed using Student's *t* test or Mann–Whitney test). (G) Immunoblotting of SR-B1, CXCL10, and CXCR3 in the left ventricle of $S^{fl/fl}$ and $S^{\Delta EC}$ HFpEF mice (left); densitometric analysis of CXCL10 and CXCR3 protein levels ($n = 10$, right). Significant difference was evaluated *via* Welch's *t* test for CXCL10, and the Mann–Whitney test for CXCR3. (H) Representative immunohistochemical images showing anti-CXCR3 staining (left panel; scale bar, 1 mm or 100 μm for zoom-in image), with corresponding quantitative analysis of CXCR3+ cells in the left ventricle of $S^{fl/fl}$ and $S^{\Delta EC}$ HFpEF mice (right panel, $n = 10$, Student's *t* test). (I) Representative flow cytometry histograms depicting the percentage of CXCR3 + T cells among CD3 + T cells in the cardiac left ventricle of $S^{fl/fl}$ and $S^{\Delta EC}$ HFpEF mice, with corresponding quantitative analysis (( J), $n = 6$, Student's *t* test). (K) Flow cytometric quantification of cardiac CXCR3 + T cell ($n = 6$, Student's *t* test). All numerical data were presented as mean ± SEM. Source data are available online for this figure.

2014). Given our in vitro data and this documented detrimental role of IRF1, we hypothesized that in vivo, endothelial-specific knockout of IRF1 would alleviate HFpEF pathology by suppressing CXCL10 upregulation. First, we confirmed significantly elevated IRF1 expression in cardiac endothelial cells from HFpEF hearts compared to healthy controls (Appendix Fig. S10A–C). Second, considering the observed progressive decrease in SR-B1 expression during HFpEF, we employed an RGD-peptide-conjugated nano-particle (RGD-Nano) system (Liu et al, 2019) to specifically knock down IRF1 in cardiac endothelial cells of mice with HFpEF on a wild-type background, rather than in endothelial SR-B1 KO mice.

Initially, we conducted in vitro screening using the mouse endothelial cell line H5V and identified an effective siRNA sequence targeting Irf1 (siIrf1). This sequence significantly down-regulated *Cxcl10* expression to 29% of baseline levels, as confirmed by RT-qPCR (Appendix Fig. S10D). To ensure the specificity of the nanomaterial delivery to cardiac endothelial cells, we conjugated Cy5.5 to RGD-Nano and performed fluorescence imaging on cardiac slices. The results demonstrated that the nanomaterials specifically targeted CD31-positive cardiac endothelial cells (Appendix Fig. S10E). Subsequently, we isolated cardiac endothelial cells from mice injected with RGD-Nano formulating either siIrf1 or its scramble control siRNA (siNC). RT-qPCR analysis confirmed the efficient knockdown of *Irf1* in vivo, which was accompanied by a significant reduction in *Cxcl10* expression (~50%) (Appendix Fig. S10F), consistent with our in vitro findings.

To evaluate the therapeutic potential of IRF1 knockdown, we subjected 10-week-old mice to a 10-week HFD plus L-NAME treatment to induce HFpEF, followed by intravenous injection of siRNA-formulating RGD-Nano particles every 3 days (Appendix Fig. S10G). Prior to treatment, there were no significant differences in body weight, blood glucose, or blood pressure between the two HFpEF groups (Appendix Fig. S10H–K). During the 10-week nanomedicine treatment period, the siNC HFpEF group exhibited severe mortality (5 out of 7 mice died), whereas the siIrf1 group showed a higher survival rate (with only 1 death out of 10 mice). This mortality may be attributed to systolic dysfunction (evidenced by a significant decrease in FS) (Appendix Fig. S10L), because only HFD plus L-NAME regimens did not contribute to a significant reduction in FS, suggesting that long-term RGD-Nano injections might have adversely affected cardiac function. Importantly, pathological examination revealed that IRF1 knockdown significantly attenuated cardiac hypertrophic remodeling, supporting our

hypothesis that elevated IRF1 exacerbates cardiac dysfunction in SR-B1-deficient mice (Appendix Fig. S10N).

In summary, our in vitro findings confirm that IRF1 activation directly drives CXCL10 upregulation when SR-B1 loss, and in vivo IRF1 knockdown effectively inhibits cardiac hypertrophy, highlighting its promising potential as a therapeutic target for HFpEF.

## Endothelial-limited CXCL10 overexpression in vivo promotes CXCR3 + T infiltration in the heart, and aggravates cardiac diastolic dysfunction and pathological remodeling in HFpEF

We attempted to investigate the in vivo role of CXCL10 in SR-B1 deficiency-mediated HFpEF adverse cardiac remodeling through endothelial-specific AAV1 delivery, and to determine whether CXCL10 is sufficient to recruit CXCR3 + T cells into cardiac tissue. Wild-type mice were subjected to HFD plus L-NAME for 10 weeks, after which they received tail vein injections of AAV1 vectors expressing either zsGreen or CXCL10, followed by an additional 10 weeks of the same dietary regimen (Fig. 7A). Echocardiography revealed preserved FS between the two HFpEF groups (Fig. 7B). However, HFpEF mice overexpressing endothelial CXCL10, i.e., AAV1-Cdh5-CXCL10-injected mice, exhibited more severe myocardial remodeling and diastolic dysfunction than AAV1-ZsGreen-injected HFpEF mice (Fig. 7C–H). Exercise exhaustion test confirmed that CXCL10 overexpressed mice exhibited significantly reduced running distances, indicating severe reduction in cardiac functional reserve (Fig. 7I). Further ex vivo assessment revealed that AAV1-CXCL10-injected HFpEF mice had more severe cardiac ventricular hypertrophy and pulmonary edema than AAV1-ZsGreen-injected HFpEF mice (Fig. 7J,K). Histological and molecular analyses revealed that hearts from CXCL10-overexpressing HFpEF mice exhibited more severe fibrosis and hypertrophy (Fig. 7L–N). Therefore, endothelial-selective CXCL10 overexpression compromised cardiac function, which was analogous to the SR-B1 deficiency phenotype.

The pathophysiological roles of CXCL10 in autoimmune diseases and cancer have been well characterized. CXCL10, secreted in a paracrine manner by endothelial cells, fibroblasts, or macrophages, can recruit CXCR3 + T cells into tissues, thereby exacerbating autoimmune inflammatory cascades (Antonelli et al, 2014) or suppressing cancer progression (Tokunaga et al, 2018). However, it remains unclear whether CXCL10 specifically derived

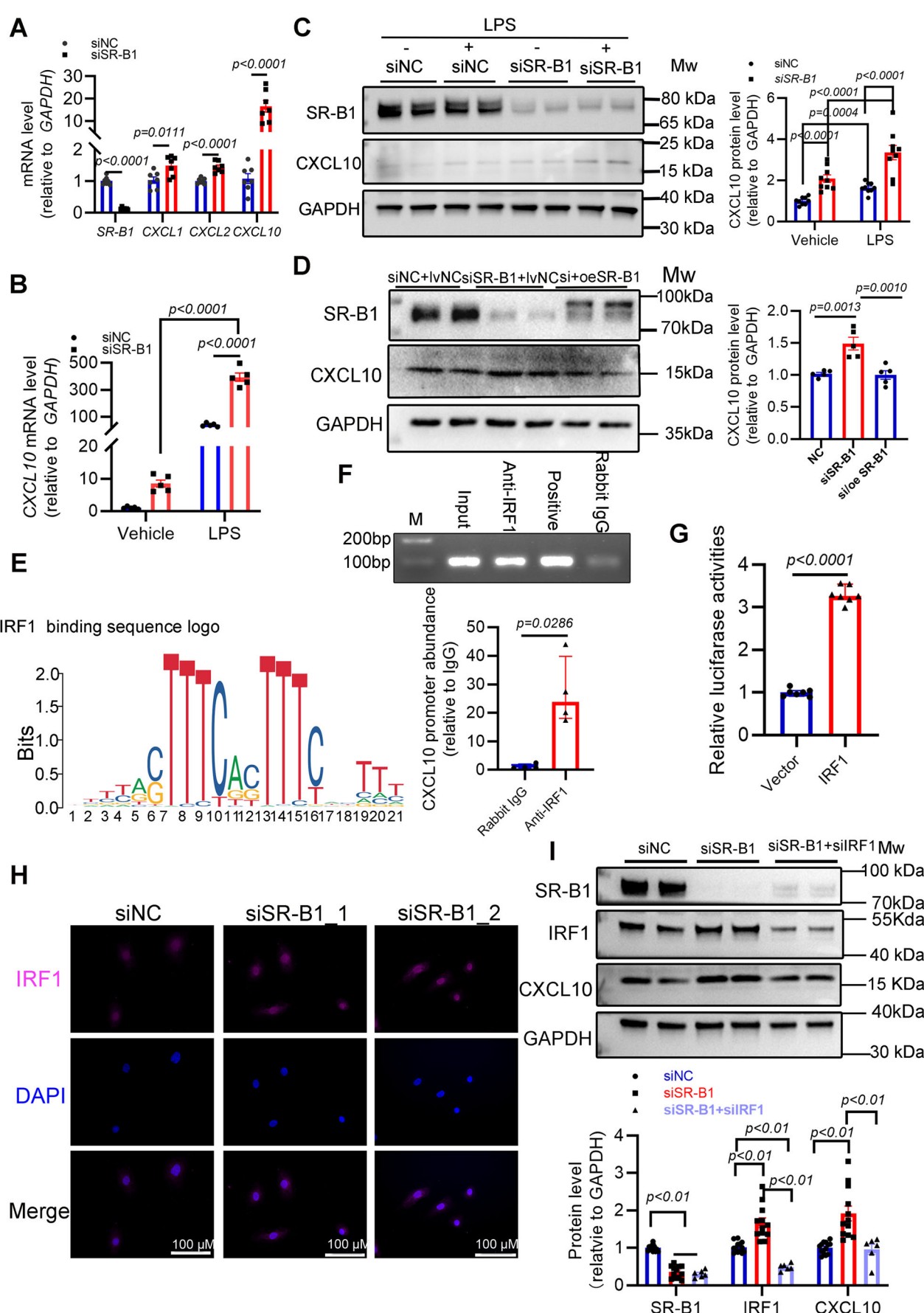

◄ **Figure 6.** The transcriptional factor IRF1 is identified as a direct mediator responsible for CXCL10 transcript upregulated by microvascular endothelial SR-B1 deletion.

(A) *SCARB1* (SR-B1), *CXCL1*, *CXCL2*, and *CXCL10* mRNA abundances were measured in HCMECs transfected with *SCARB1* siRNA (siSR-B1) or a normal control siRNA (siNC). Significant differences were analyzed by Student's *t* test ($n = 7$). (B) *CXCL10* mRNA abundance was determined in different groups *via* RT-qPCR ($n = 5$; statistical differences were evaluated by two-way ANOVA followed by Bonferroni post-hoc test). HCMECs transfected with siNC or siSR-B1 were incubated with lipopolysaccharide (LPS, 20 ng/mL) for 8 h. (C) CXCL10 protein abundance was evaluated in the four groups by immunoblotting ($n = 9$; two-way ANOVA followed by Bonferroni post-hoc test). (D) SR-B1 and CXCL10 protein abundance was assessed by immunoblotting in HCMECs co-transfected with siRNA (siNC or siSR-B1) and lentivirus (overexpression of homoSR-B1 [oeSR-B1] or control lentivirus [lvNC]), and analyzed *via* one-way ANOVA followed by Bonferroni post-hoc test ($n = 5$). (E) IRF1-binding motif obtained from JASPAR website (Matrix ID Ma0050.2). Predicted site was located in *CXCL10* promoter at 79–99 bp. (F) Agarose gel electrophoresis of PCR-amplified products based on chromatin immunoprecipitation assay to investigate the possibility of IRF1 binding to the *CXCL10* gene promoter in HCMECs incubated with LPS (upper panel); *CXCL10* promoter abundance was quantified by qPCR ($n = 4$, lower panel). The Mann–Whitney test was used. (G) Luciferase reporter analysis in 293T cells ($n = 7$). Significant difference was evaluated by Student's *t* test. (H) Immunofluorescent staining of IRF1 co-stained with DAPI to compare nucleus signals of IRF1 in HCMECs when SR-B1 presence or absence. Scale bar, 100 μm. (I) Representative immunoblot images showing protein expression profiles of SR-B1, IRF1, CXCL10, and GAPDH in HCMECs following transfection with control siNC, siSR-B1, or combined siSR-B1 and siIRF1 (upper panel), and quantitative densitometric analysis of relative protein expression levels for SR-B1, CXCL10, and IRF1 (lower panel, $n = 13, 12, 6$, Brown–Forsythe test and Welch ANOVA with Dunnett's T3 multiple comparisons). All numerical data were presented as mean ± SEM. Source data are available online for this figure.

from cardiac endothelial cells can drive CXCR3+ cell infiltration into the heart.

First, we employed Western blot and RT-qPCR analysis and found that CXCL10 and CXCR3 expression was significantly higher in CXCL10-overexpressing hearts compared to ZsGreen-expressing controls (Fig. 7N,O). Next, we utilized flow cytometry to assess immune cell infiltration in HFpEF mice at 5 weeks post-AAV injection (Appendix Fig. S11A). Consistent with mice of 10 weeks post-AAV injection, these CXCL10-overexpressing mice exhibited cardiac hypertrophy and diastolic dysfunction (Appendix Fig. S11B–E). The flow cytometry results demonstrated that endothelial CXCL10 overexpression led to a significant increase in the CXCR3 + /CD3 + T cell ratio and CXCR3 + T cell counts (Fig. 7P–R), as well as a marked elevation in CD45+ leukocytes (Appendix Fig. S11F). CD3 + T cells and CD11b+ myeloid cells remained unchanged (Appendix Fig. S11G,H).

Subsequently, we applied the same strategy to evaluate the effects of CXCL10 on immune cell infiltration and cardiac function in healthy mice (Appendix Fig. S11I). Notably, these mice did not exhibit significant cardiac hypertrophy or diastolic dysfunction (Appendix Fig. S11J–M). Flow cytometry analysis revealed that, apart from a marked increase in the proportion of CXCR3 + / CD3 + T cells, no significant alterations were observed in immune cell subsets (including CD45 + , CD11b + , CD3 + , or total CXCR3+ cell counts) in AAV1-Cdh5-CXCL10-injected mice compared to AAV1-Cdh5-zsGreen-injected controls (Appendix Fig. S11N–S).

In summary, CXCL10 is a critical mediator of pathogenic CXCR3 + T cell infiltration and cardiac dysfunction in HFpEF, but it cannot independently induce significant immune microenvironmental changes or functional impairments in healthy mice. This suggests that CXCL10-mediated recruitment of CXCR3 + T cells into the myocardium likely depends on the interplay between systemic inflammatory signals and cardiac-specific microenvironmental factors.

Despite the higher clinical susceptibility of women to HFpEF (Mishra and Kass, 2021), female sex is paradoxically protective in a widely used preclinical two-hit model (Tong et al, 2019). To investigate whether this protection extends to microvascular inflammatory activation, we assessed key molecules (*Irf1*, *Cxcl1*, *Cxcl2*, *Cxcl10*, *Icam1*) in ovariectomized (OVX) two-hit female mice. No significant changes were detected (Appendix Fig. S12). This absence of microvascular activation aligns with a recent report

(Smolgovsky et al, 2023) showing that XBP1s downregulation—a key finding in male mice—was absent in females. Thus, the lack of molecular changes in our study may indeed be attributed to a fundamental protective mechanism in female mice within this specific model.

## IRF1-CXCL10/CXCR3 axis closely associated with cardiac diastolic dysfunction in patients with HFpEF

To support these preclinical experimental findings, the gene abundance of the SR-B1–IRF1–CXCL10/CXCR3 axis was assessed in the left ventricles of patients with cardiomyopathy-related HFpEF. The characteristics of the HFpEF donors indicated that the HFpEF patients had preserved LVEF, but increased left ventricular diastolic posterior wall thickness (LVPWT), E/e' ratio, and elevated NT-proBNP concentration (Appendix Table S1). The experimental data indicated that the ventricle of HFpEF patient contained high abundances of *BNP* and *COL3A1*, which coordinated with the SR-B1 gene marked downregulation and significant increases in *IRF1*, *CXCL10*, and *CXCR3* abundance (Fig. 8A–C), compared with the abundance in the healthy control group.

We also enrolled 176 individuals (109 HFpEF cases and 67 healthy controls) to investigate the correlation between plasma CXCL10 concentrations and HFpEF (Appendix Table S2). Plasma CXCL10 and N-terminal pro-B-type natriuretic peptide (NT-proBNP) concentrations were measured using an enzyme-linked immunosorbent assay. The CXCL10 and NT-proBNP concentrations in HFpEF plasma were increased significantly compared with control plasma (Fig. 8D; Appendix Fig. S13A). Bivariate linear correlation analysis revealed a positive correlation between plasma CXCL10 and NT-proBNP concentrations ($r = 0.32$, $P < 0.001$, Fig. 8E). Subsequently, univariate and multivariate logistic regression analyses were performed to identify risk factors for HFpEF (Appendix Table S3). Based on the univariate and multivariate logistic regression analysis results, log CXCL10 [OR = 3.28 (95% CI: 1.01–10.65), $P = 0.048$] was a positive factor associated with HFpEF (Fig. 8F). Furthermore, HDL-C concentration in HFpEF plasma was significantly lower (1.20 vs 1.01, $P < 0.001$; Appendix Table S2). Additional Spearman correlation analysis showed that plasma CXCL10 levels were negatively correlated with HDL-C ($r = -0.18$, $P = 0.015$; Appendix Fig. S13B), suggesting that the increase in CXCL10 abundance in HFpEF plasma is likely associated with a low level of HDL-mediated SR-B1 activation. The findings

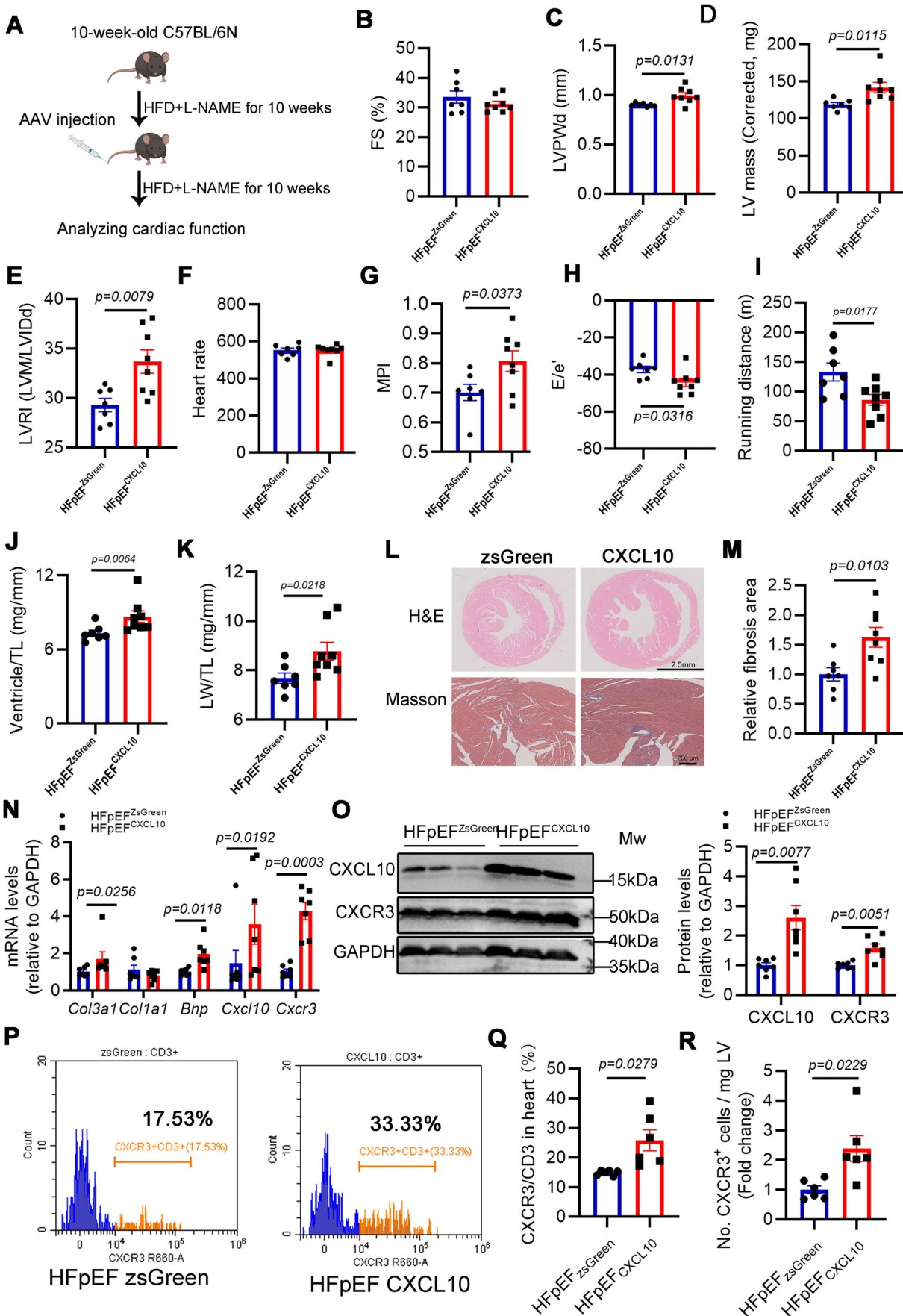

**Figure 7. Endothelial-restricted CXCL10 overexpression deteriorates cardiac diastolic dysfunction and pathological remodeling in HFpEF, correlating with enhanced infiltration of CXCR3 + T cells in the heart.**

(A) A schematic diagram of the experiment workflow to investigate the role of endothelial CXCL10 in HFpEF mice. Echocardiographic evaluation of cardiac functional changes of HFpEF mice infected with AAV1-mediated endothelial-restricted CXCL10 (*HFpEF^CXCL10^*) or control ZsGreen (*HFpEF^ZsGreen^*) delivery ((B–I), $n = 7$, 8; statistical differences were evaluated by Student's $t$ test). (B) FS. (C) LVPWd. (D) LV mass, left ventricular mass. (E) LVRI. (F) Heart rate in parasternal short-axis view. (G) MPI. (H) E/e' ratio. (I) Running distance during the exercise exhaustion test. (J) Ratio between cardiac ventricle weight and tibia length (TL) ($n = 7$, 8; Mann–Whitney test). (K) Ratio between wet lung weight and TL ($n = 7$, 8; Mann–Whitney test). (L) Representative images of H&E (scale bar, 2.5 mm) and MT (scale bar, 250 μm) staining. (M) Fibrosis area in MT-stained transversal sections relative to *HFpEF^ZsGreen^* mice ($n = 7$, 8; Student's $t$ test). (N) *Col3a1*, *Col1a1*, *Bnp*, *Cxcl10*, and *Cxcr3* mRNA abundance in the left cardiac ventricle was measured *via* RT-qPCR ($n = 7$, Mann–Whitney test or Welch's $t$ test). (O) CXCL10 and CXCR3 protein abundances in *HFpEF^ZsGreen^* and *HFpEF^CXCL10^* hearts were measured *via* immunoblotting and evaluated according to the densitometric data ($n = 7$, Welch's $t$ test). (P) Representative flow cytometry histograms depicting the percentage of CXCR3 + T cells among CD3 + T cells in mouse cardiac left ventricle, with corresponding quantitative analysis ((Q), $n = 6$, Welch's $t$ test). (R) Flow cytometric quantification of cardiac CXCR3 + T cells ($n = 6$, Welch's $t$ test). The flow cytometric data were obtained in *HFpEF^ZsGreen^* and *HFpEF^CXCL10^* mice subjected to 5 weeks of HFD plus L-NAME regimens post-injection of zsGreen or CXCL10-expressing AAV. All numerical data were presented as mean ± SEM. Source data are available online for this figure.

demonstrated that the IRF1-CXCL10/CXCR3 axis is involved in clinical HFpEF pathogenesis.

## Discussion

Growing evidence suggests that cardiac microvascular endothelial activation and the infiltration of activated T cells into the myocardium contribute to HFpEF pathogenesis. (Smart et al, 2023). This study is the first to demonstrate the protective role of cardiac microvascular endothelial SR-B1 in mitigating adverse remodeling and diastolic dysfunction, which is mechanistically linked to the suppression of inflammatory activation in CMECs (cardiac microvascular endothelial cells) characterized by reduced secretion of the chemokine CXCL10 in HFpEF. Further in vivo investigations revealed that the CXCL10 secretion mediates the infiltration of specific CXCR3+ activated T cells and exacerbates cardiac dysfunction in HFpEF hearts. Notably, the activation of the SR-B1-CXCL10-CXCR3 signaling axis was also observed in HFpEF patients, and elevated circulating CXCL10 levels were identified as an independent risk factor for HFpEF development. These findings collectively indicate that the SR-B1-CXCL10-CXCR3 axis mediates HFpEF-related cardiac dysfunction by integrating CMEC activation with T-cell cardiotropism (Fig. 8G).

Using Single-cardiac-endothelial-cell transcriptomics, we first defined an inflammatory subcluster from heterogeneous mice CMECs, characterized by high expression of proinflammatory genes, including *Irf1*, *Cxcl1*, *Cxcl2*, and *Cxcl10*, along with inflammatory signaling pathway activation in HFpEF hearts. Notably, SR-B1 deletion promoted an increase in the proportion of the inflammatory subcluster in HFpEF, and accompanied by deteriorative cardiac function. The involvement of SR-B1 in inflammation activation has been shown in other systems. For example, endothelial inflammatory activator LPS downregulated SR-B1 abundance significantly (Baranova et al, 2002). In macrophages or human umbilical vein ECs, SR-B1 deficiency aggravated oxidized LDL- or LPS-induced inflammatory signaling both in vivo and in vitro (Cai et al, 2012; Ren et al, 2017; Tao et al, 2021).

The secretory function of CMECs is an important conduit for crosstalk between CMECs and other cardiac cells in cardiac remodeling (Segers et al, 2018). Microvascular ECs are major participants and regulators of inflammatory processes (Pober and

Sessa, 2007). We demonstrated that CXCL10, a chemokine secreted by cardiac microvascular endothelial cells (CMECs), drives significant infiltration of CXCR3 + T cells into the myocardium, and exacerbating pathological cardiac remodeling and diastolic dysfunction. Compelling evidence indicates that CXCL10 promotes pathogenesis in disorders of chronic inflammation, including diabetes (Javeed et al, 2021), liver fibrosis (Hintermann et al, 2010), atherosclerosis (Heller et al, 2006), multiple sclerosis (Sorensen et al, 1999), and heart transplantation (Hancock et al, 2001) through recruitment of CXCR3-positive effector T cells to target organs. Our data uncovered that cardiac CXCR3 is expressed predominantly by T cells, and CMECs SR-B1 loss specifically promotes the infiltration of CXCR3-positive cells in the left cardiac ventricle. Consistently, a recent study also confirmed that the proportion of CXCR3-positive T cells (γδ T cells) was increased significantly in another type of HFpEF mouse model induced by deoxycorticosterone acetate salt, compared with sham controls(Smart et al, 2023). Moreover, mounting data have shown that T-cell infiltration in the heart drives the pathogenesis of chronic heart failure, especially non-ischemic heart failure (Laroumanie et al, 2014; Nevers et al, 2015; Ngwenyama et al, 2019), and accentuates cardiac diastolic dysfunction (Brassington et al, 2022; Smolgovsky et al, 2023).

The low concentration of nitric oxide (NO) caused by eNOS inactivation is also considered a manifestation of CMEC activation in HFpEF (Paulus, 2020). NO production in vitro in peripheral vascular ECs is regulated by SR-B1-mediated eNOS activation (Assanasen et al, 2005; Mineo et al, 2003; Yuhanna et al, 2001), which also occurs in cardiac ECs (data not shown). In the 'two-hit' model, the eNOS inhibitor L-NAME was used to imitate the hypertension of patients with HFpEF by reducing endothelial NO concentration. However, our results showed that the blood pressure in mice remained unchanged regardless of the diet (normal or HFpEF diet). Thus, we inferred that NO concentration in endothelial cells does not substantively contribute to the cardiac diastolic dysfunction accentuated by SR-B1 deletion.

Cardiac fibrosis is a key feature of the remodeled heart in HFpEF and is a major cause of diastolic dysfunction. Myofibroblasts, the main cause of collagen net formation in the heart, could arise from the transition of cardiac endothelial cells in vitro, driven by inflammatory factors. Unfortunately, in the inflammatory HFpEF development, we did not find that the transition is present in vivo

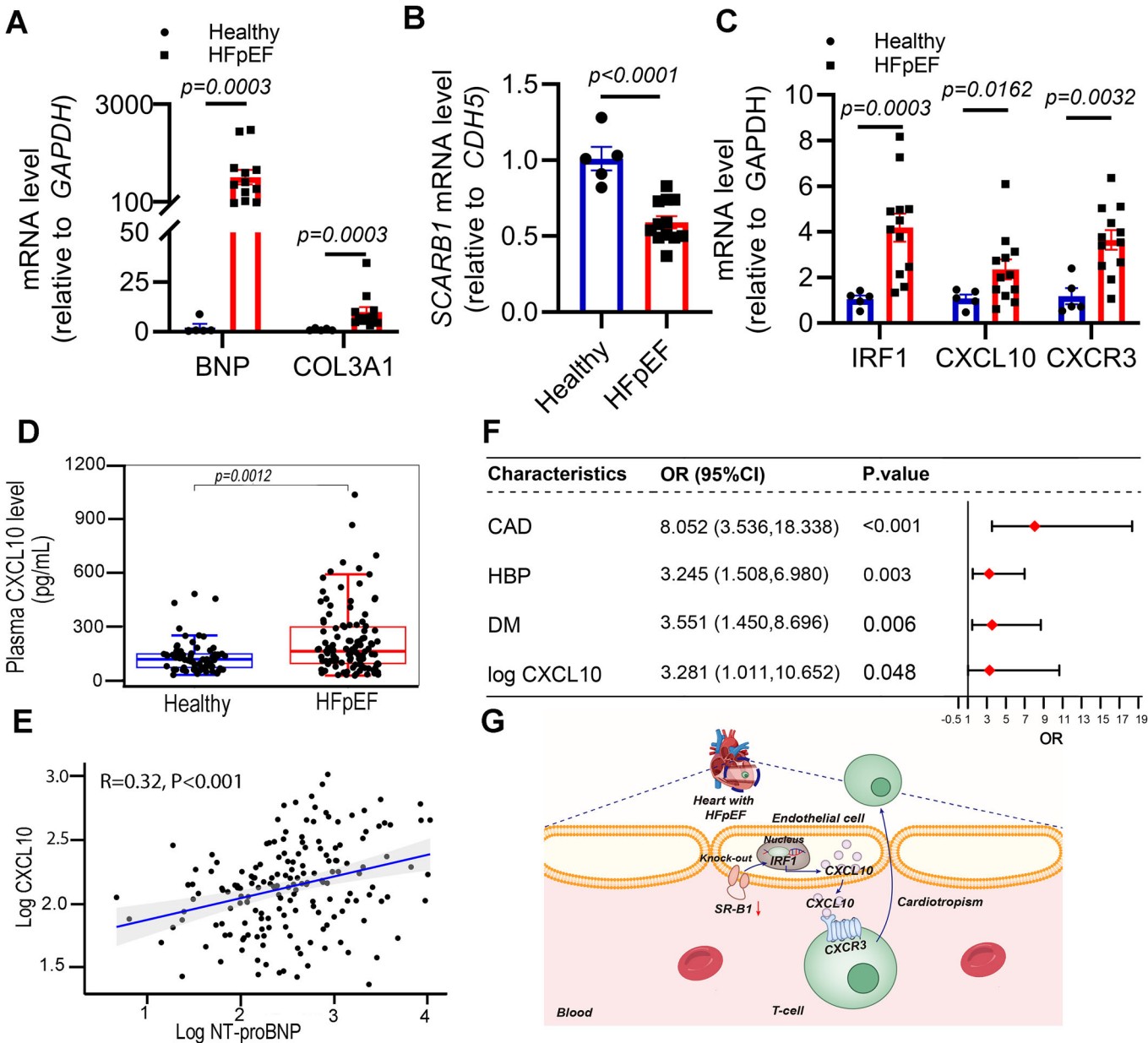

**Figure 8. IRF1-CXCL10/CXCR3 axis implicates in the pathogenesis of clinical HFpEF patients.**

(A) *BNP* and *COL3A1* mRNA abundances were assessed in healthy and HFpEF individuals by RT-qPCR ($n = 5$, 12; Mann–Whitney test). Data were presented as mean ± SEM. (B) *SCARB1* (SR-B1) mRNA abundance was evaluated in healthy and HFpEF individuals ($n = 5$, 12; Student's $t$ test). Data were presented as mean ± SEM. (C) *IRF1*, *CXCL10*, and *CXCR3* mRNA abundances in healthy and HFpEF individuals ($n = 5,12$; Welch's or Student's $t$ test). Data were presented as mean ± SEM. (D) Plasma CXCL10 concentrations were significantly elevated in HFpEF patients (respectively, $n = 67$, 109; Mann–Whitney test). The box plots are defined as follows: Center of box, Median (healthy: 114.75 pg/mL; HFpEF: 159.95 pg/mL). Bounds, Interquartile range (IQR; 25th to 75th percentiles) (healthy: 68.92–145.20 pg/mL; HFpEF: 91.46–296.91 pg/mL); Bounds of whiskers, Lowest and highest values within 1.5 × IQR (lower whisker: healthy 26.91 pg/mL, HFpEF 23.27 pg/mL; upper whisker: healthy 249.27 pg/mL, HFpEF 593.29 pg/mL); Minima and Maxima, Full data range (healthy: 26.91–482.91 pg/mL; HFpEF: 23.27–1039.95 pg/mL). (E) Bivariate linear correlation analysis between log CXCL10 and log NT-proBNP ($R = 0.32$, $P < 0.001$). (F) Forest plot showing the effect of each variable on HFpEF in multivariate logistic regression. OR odds ratio, CI confidence interval. (G) A schematic diagram of the potential mechanism by which SR-B1 deficiency confers IRF1-CXCL10/CXCR3 activation and exacerbation of cardiac diastolic dysfunction. Source data are available online for this figure.

using genetic lineage-tracing technology. Furthermore, our results demonstrated that the crosstalk of CMECs and T cells likely led to the aggravation of cardiac fibrosis. Mounting evidence has shown that T cells both directly and indirectly promote extracellular

matrix deposition and fibrosis in the pathological heart (Bayer et al, 2023; Bradshaw and DeLeon-Pennell, 2020). Thus, we propose that the deterioration of fibrosis in SR-B1-deficient HFpEF heart was attributable to T-cell accumulation and fibroblast activation.

## The present study has several limitations that warrant discussion

The present study has several limitations that warrant discussion. Firstly, acute heart failure and chronic heart failure represent distinct pathological conditions. In our study, the role of SR-B1 was exclusively investigated in the context of acute heart failure. Previous studies have indicated that CXCR3 + T cell infiltration exacerbates cardiac systolic dysfunction (Ngwenyama et al, 2019), and SR-B1 deficiency facilitates the recruitment of CXCR3 + T cells into the heart. Therefore, SR-B1 may also play a potential role in HFrEF. As such, an ideal preclinical model of chronic HFrEF should be utilized in future research to further elucidate the role of SR-B1 in HFrEF. Second, we preliminarily demonstrated that HDL, the primary ligand of SR-B1, reduces CXCL10 expression under inflammatory stimulation. However, the precise mechanism by which SR-B1, a membrane receptor, regulates downstream gene transcription remains unclear and requires further investigation. Third, although we validated that EnMT did not occur in HFpEF in vivo, an in vitro assay indicated that endothelial cells have the capacity to produce collagen. Therefore, future studies should directly evaluate collagen secretion from endothelial cells rather than focusing on the cell transition. Fourth, a critical unresolved question is whether the IRF1-mediated CXCL10 upregulation, identified in vitro, underlies the aggravated diastolic dysfunction in endothelial SR-B1 knockout mice in vivo. Finally, this study was conducted at a single center. Although we reported a correlation between CXCL10 and HFpEF after adjusting for potential confounders, multicenter validation is necessary to confirm these findings. In addition, HFpEF is a heterogeneous syndrome, but our study focused exclusively on a cardiomyopathy-related subtype to evaluate the role of SR-B1-CXCL10-CXCR3 signaling in HFpEF pathogenesis. This narrow focus may introduce bias and limit the generalizability of our results.

In conclusion, SR-B1 could reduce CMECs activation and accumulation of CXCR3-positive T-cells, thereby slowing HFpEF progression. Such results unveil novel potential therapeutic targets and strategies for the clinical management of HFpEF.

## Methods

### Reagents and tools table

| Reagent/ resource | Reference or source | Identifier or catalog number |
|---|---|---|
| **Experimental models** | | |
| C57BL/6 N mice | Charles River Laboratories | |
| SR-B1*flox/flox* mice | Beijing Viewsolid Biotechnology Co. LTD | |
| Cadherin 5 (Cdh5)-Cre mice | Model Animal Research Center of Nanjing University | |
| R26-stop-tdTomato mice | The Jackson Laboratory | Stock #007914 |
| HCMECs | ScienCell | #6000 |

| Reagent/ resource | Reference or source | Identifier or catalog number |
|---|---|---|
| 293 T | Procell | CL-0005 |
| H5V | Provided by Professor Jie Liu (Shanghai East Hospital, Tongji University School of Medicine, China). | |
| **Antibodies** | | |
| **Western blotting** | | |
| Anti-GAPDH | Proteintech | 6004-1-Ig |
| Anti-SR-B1 | Abcam | Ab217318 |
| Anti-CD31 | Abcam | Ab28364 |
| Anti-Cdh5 | Abcam | Ab33168 |
| Anti-β-actin | Proteintech | 66009-1-Ig |
| Anti-CXCL10 | R&D | AF-466-NA |
| Anti-CXCR3 | Abcam | Ab288437 |
| Anti-IRF1 | CST | #8478 s |
| Anti-mouse IgG | MBL | 330 |
| Anti-rabbit IgG | MBL | 458 |
| Anti-goat IgG | Proteintech | SA0001-3 |
| **Immunostaining** | | |
| CXCR3 | Abclone | A11294 |
| CD3E | Abclone | A12415 |
| Anti-SR-B1 | Abcam | Ab217318 |
| Anti-CD31 | Santa Cruz | Sc-376764 |
| Anti-CD31 | Abcam | Ab28364 |
| Anti-PDGFRα | R&D | Af1062 |
| Anti-αSMA | Abcam | Ab7818 |
| Anti-tdTomato | Arigo | ARG55724 |
| Anti-cTnT | Abcam | Ab8295 |
| Anti-cTnl | Abcam | Ab10231 |
| **Primary mouse cardiac endothelial cell isolation** | | |
| Anti-CD31 | BD Pharmingen | 553370 |
| **FACS sorting** | | |
| FITC-conjugated anti-mouse CD31 | Biolegend | 102406 |
| Brilliant Violet 421™ anti-mouse CD3ε Antibody | Biolegend | 100336 |
| PerCP/ Cyanine5.5 anti-mouse CD45 Antibody | Biolegend | 103132 |
| BV605 Rat Anti-CD11b | BD Pharmingen | 563015 |
| APC anti-mouse CD183 (CXCR3) Antibody | Biolegend | 126512 |

| Reagent/ resource | Reference or source | Identifier or catalog number |
|---|---|---|
| TruStain FcX™ (anti-mouse CD16/32) Antibody | Biolegend | 101320 |
| Zombie Green™ Fixable Viability Kit | Biolegend | 423112 |
| Brilliant Violet 510™ anti-mouse CD3 Antibody | Biolegend | 100234 |
| **Chip-qPCR** | | |
| Anti-IRF1 | CST | #8478 s |
| Anti-Histone 3 | CST | #9725t |
| **Oligonucleotides and other sequence-based reagents** | | |
| **Genotyping** | | |
| Mouse SR-B1-F | Tsingke Biotech | CCCTAATCCCTGCCCATA |
| Mouse SR-B1-R | Tsingke Biotech | CAAGGCTCATCTGGAAGG |
| Mouse Cdh5-CreER-F | Tsingke Biotech | CCGGTCGATGCAACGAGTGATGAGG |
| Mouse Cdh5-CreER-R | Tsingke Biotech | GCCTCCAGCTTGCATGATCTCCGG |
| Mouse R26-tdTomato-F | Tsingke Biotech | GGCATTAAAGCAGCGTATCC |
| Mouse R26-tdTomato-R | Tsingke Biotech | CTGTTCCTGTACGGCATGG |
| **Silencing gene expression** | | |
| Human SR-B1 siRNA_1 sense and anti-sense | Genechem | GCUCGUCAACAAGCACUGUUC |
| | | GAACAGUGCUUGUUGACGAGC |
| Human SR-B1 siRNA_2 sense and anti-sense | Genechem | GCCAAGAGAAAUGCUAUUUAU |
| | | AUAAAUAGCAUUUCUCUUGGC |
| Human IRF1 siRNA sense and anti-sense | Genechem | CGAAGACCUUAUGAAGCUCUU |
| | | AAGAGCUUCAUAAGGUCUUCG |
| **RT-qPCR** | | |
| Mouse GAPDH-F | Tsingke Biotech | CCCTAATCCCTGCCCATA |
| Mouse GAPDH-R | Tsingke Biotech | CAAGGCTCATCTGGAAGG |
| Mouse SR-B1-F | Tsingke Biotech | AATGTCCGCATAGACCCGAG |
| Mouse SR-B1-R | Tsingke Biotech | AGGACCTCGTTTGGGTTGAC |
| Mouse Cd31-F | Tsingke Biotech | CCAGAAACATCATCATAACCG |
| Mouse Cd31-R | Tsingke Biotech | CATCGCCACCTTAATAGTTG |
| Mouse Cdh5-F | Tsingke Biotech | ATTGAGACAGACCCCAAACG |
| Mouse Cdh5-R | Tsingke Biotech | TTCTGGTTTTCTGGCAGCTT |
| Mouse Bnp-F | Tsingke Biotech | GAAGGACCAAGGCCTCACAA |
| Mouse Bnp-R | Tsingke Biotech | CGTTCAGCCCAAACGACTG |
| Mouse Cxcl1-F | Tsingke Biotech | ACTCAAGAATGGTCGCGAGG |
| Mouse Cxcl1-R | Tsingke Biotech | GTGCCATCAGAGCAGTCTGT |
| Mouse Cxcl2-F | Tsingke Biotech | GGCGGTCAAAAAGTTTGCCT |

| Reagent/ resource | Reference or source | Identifier or catalog number |
|---|---|---|
| Mouse Cxcl2-R | Tsingke Biotech | TTCTTCCGTTGAGGGACAGC |
| Mouse Cxcl10-F | Tsingke Biotech | TCCCCATCAGCACCATGAAC |
| Mouse Cxcl10-R | Tsingke Biotech | ATTCTCACTGGCCCGTCATC |
| Mouse Col1a1-F | Tsingke Biotech | GCTCCTCTTAGGGGCCACT |
| Mouse Col1a1-R | Tsingke Biotech | ATTGGGGACCCTTAGGCCAT |
| Mouse Col3a1-F | Tsingke Biotech | CTGTAACATGGAAACTGGGGAAA |
| Mouse Col3a1-R | Tsingke Biotech | CCATAGCTGAACTGAAAACCACC |
| Mouse αSMA-F | Tsingke Biotech | GTCCCAGACATCAGGGAGTAA |
| Mouse αSMA-R | Tsingke Biotech | TCGGATACTTCAGCGTCAGGA |
| Mouse eNOS-F | Tsingke Biotech | GCTTGGGATCCCTGGTGTTT |
| Mouse eNOS-R | Tsingke Biotech | GGAGCCACTCCTTTTGATGGA |
| Mouse Cxcr3-F | Tsingke Biotech | AATGCCACCCATTGCCAGTA |
| Mouse Cxcr3-R | Tsingke Biotech | TAGCTCGAAAACGCCTCTGG |
| Homo GAPDH-F | Tsingke Biotech | CTCCTCCACCTTTGACGCTG |
| Homo GAPDH-R | Tsingke Biotech | TCCTCTTGTGCTCTTGCTGG |
| Homo SR-B1-F | Tsingke Biotech | CGTGTCCTTCCTCGAGTACC |
| Homo SR-B1-R | Tsingke Biotech | GCACCCAAGACCAGGATGTT |
| Homo CXCL1-F | Tsingke Biotech | ATCATTGTGAAGGCAGGGGA |
| Homo CXCL1-R | Tsingke Biotech | GCCCCTTTGTTCTAAGCCAGA |
| Homo CXCL2-F | Tsingke Biotech | GCCACTGTGATAGAGGCTGA |
| Homo CXCL2-R | Tsingke Biotech | CTAACACAGAGGGAAACACTGC |
| Homo CXCL10-F | Tsingke Biotech | AGCAGAGGAACCTCCAGTCT |
| Homo CXCL10-R | Tsingke Biotech | ATGCAGGTACAGCGTACAGT |
| Homo BNP-F | Tsingke Biotech | GATCCCCAGACAGCACCTTC |
| Homo BNP-R | Tsingke Biotech | CTGTAACCCGGACGTTTCCA |
| Homo COL3A1-F | Tsingke Biotech | CTTCTCTCCAGCCGAGCTTC |
| Homo COL3A1-R | Tsingke Biotech | GTAGTCTCACAGCCTTGCGT |
| Homo CXCR3-F | Tsingke Biotech | GGAGGTACAGCACGAGTCAC |
| Homo CXCR3-R | Tsingke Biotech | TGCTAAATGACGCCGAGGTT |
| Homo IRF1-F | Tsingke Biotech | GGCTGAGTCTACAGCGTGTC |
| Homo IRF1-R | Tsingke Biotech | AAGTCCAGCTTCTCTGCACC |
| **Chip-PCR** | | |
| Homo CXCL10 promoter-F | Tsingke Biotech | AGAAACAGTTCATGTTTTGGAAAGT |
| Homo CXCL10 promoter-R | Tsingke Biotech | TACGGAATTTCCCTCTGCTCCT |
| **Chemicals, enzymes, and other reagents** | | |
| CXCL10 ELISA kit | Abcam | ab173194-96T |
| ʟ-NAME | Cayman | 80210 |
| 10% kcal% fat diet | Research Diet | D12450J |

| Reagent/ resource | Reference or source | Identifier or catalog number |
|---|---|---|
| 60% kcal% fat diet | Research Diet | D12492 |
| Collagenase IV | Worthington | LS004188 |
| Collagenase II | Worthington | LS004176 |
| DNase I | Roche | 11284932001 |
| Sheep anti-Rat IgG magnetic beads | Invitrogen | 11035 |
| Low-glucose DMEM medium | Invitrogen | 11885076 |
| FBS | Gibco | 10099141 C |
| Penicillin/ Streptomycin solution | Solarbio | P7630 |
| Heparin | Macgene | CC112 |
| ECGS | Macgene | CC019 |
| TGF-β1 | Peprotech | #100-21-10 |
| $H_2O_2$ | Beijing Tongguang Fine Chemicals Company | 106058 |
| IL-1β | Peprotech | 200-01B-2# |
| Endothelial cell medium | ScienCell | #1001 |
| HitransG P | Genechem | REVG005 |
| High-glucose DMEM medium | Gibco | C11995500BT |
| Masson's trichrome stain kit | Solarbio | G1340 |
| Anti-rabbit DAB detection kit | ZSGB-Bio | PV-6001 |
| DAPI | Coolaber | SL1841 |
| RIPA lysis buffer | Beyotime | P0013C |
| BCA protein detecting kit | Pierce | 23227 |
| ECL substrate | Pierce | 32209 |
| Super sensitive ECL luminescence reagent | Meilunbio | MA0186 |
| Violet fluorescent Live/Dead reactive dye | Invitrogen | L34963 |
| Red blood cell lysis buffer | Solarbio | R1010 |
| Zombie Fixable Viability Kit | BioLegend | 423112 |
| Precision Count Beads | BioLegend | 424902 |
| Dual-Luciferase® Reporter assay system | Promega | E1910 |

| Reagent/ resource | Reference or source | Identifier or catalog number |
|---|---|---|
| PEI-Fe₃O₄ magnetic nanoparticles | NANOEAST | Mag2100 |
| Cyanine5.5 NHS ester | Aladdin | C171354 |
| **Software** | | |
| GraphPad Prism | Graphpad | Prism v9.0.0 |
| ImageJ | NIH | Image 1.48 |
| Leica application suite X | Leica | v3.4.6.21481 |
| Digital pathology(NDP) | Humamatsu Photonics | v2 |
| FACSDiva Software | BD Biosciences | v9.0 |
| R Software | R Development Core Team | v4.2.3 |
| Vevo Lab Software | VisualSonics | v3.1.1 |
| Image lab | Bio-RAD | v3.0 |
| PASS | NCSS, LLC | v15.0 |
| **Other** | | |

## Clinical sample collection

All participants were recruited from Qilu Hospital of Shandong University between May 25, 2021 and August 27, 2024 and signed an informed consent form. This study was conducted in accordance with the principles of the Declaration of Helsinki and the guidelines set out in the Department of Health and Human Services Belmont Report. All procedures were reviewed and approved by the Ethics Committee of Shandong University Qilu Hospital (No. KYLL-2020(090) and No. KYLL-2021(299)).

HFpEF was diagnosed according to the European Society of Cardiology guidelines (McDonagh et al, 2021): (1) Symptoms and signs of HF; (2) LVEF ≥ 50%; (3) Objective evidence of cardiac structural and/or functional abnormalities consistent with the presence of LV diastolic dysfunction/raised LV filling pressures, including NTproBNP >125 pg/mL.

Endomyocardial tissue from the left ventricular septum was obtained by a standard left ventricular outflow tract-sparing surgery in patients with left ventricular hypertrophy ($n = 12$). All HFpEF donors meeting diagnostic criteria for HFpEF. Control endomyocardial tissue ($n = 5$) was obtained from unused donor hearts that could not be used for transplantation. All ventricular septal samples obtained were rapidly frozen in liquid nitrogen for future analysis of gene expression with RT-qPCR.

In the study of serum CXCL10, to minimize the possibility that some abnormal conditions may influence the results, patients with any of the following conditions were excluded: (1) History of left ventricular ejection fraction (LVEF) < 50% at any time; (2) Isolated right heart failure due to pulmonary disease; (3) Primary valvular diseases; (4) Previous myocardial infarction within six months; (5) Renal dysfunction, severe infection, severe liver dysfunction, or

multiple organ failure. The control included participants who failed to meet the HFpEF diagnostic criteria for appeal and were recruited from the Health Management Centre in Qilu Hospital of Shandong University. Smoking was defined as "ever smoked" as compared to "never smoked". Smoking is defined as currently or formerly smoking more than one cigarette per day for a duration of over 6 months. Drinking was defined as "ever drunk" as compared to "never drunk". Drinking is defined as currently or formerly having alcoholic beverages more than once a week for a duration of over 6 months. Hypertension was defined as systolic blood pressure (SBP) $\geq 140$ mmHg and/or diastolic blood pressure (DBP) $\geq 90$ mmHg, or current anti-hypertensive drug therapy. Diabetes mellitus, atrial fibrillation, and coronary artery disease were all determined based on the diagnosis in the medical history taken from the participants.

Venous blood samples were obtained from HFpEF participants ($n = 109$) and non-HFpEF participants ($n = 67$) by venipuncture and drawn into EDTA tubes, which were immediately placed on ice before transfer to the laboratory. The tubes were spun at 4000 rpm for 10 min to separate the plasma and then stored at $-80\ °C$ for future batch analysis of CXCL10 expression by enzyme-linked immunosorbent assay.

## CXCL10 plasma-level measurements

Enzyme-linked immunosorbent assay (ELISA) was applied to measure CXCL10 plasma concentrations according to the manufacturer's instructions (CXCL10, ab173194-96T, Abcam).

## Clinical echocardiography

All participants were examined by a GE Vivid E95 or GE Vivid E9 color Doppler ultrasound, following recommendations of the American Society of Echocardiography (Lang et al, 2005; Nagueh et al, 2016). Left ventricular systolic function was assessed by left ventricular ejection fraction (LVEF), and left ventricular diastolic function was assessed by septal and lateral mitral annular peak early diastolic velocity (e'), as well as the ratio of early trans-mitral flow to early medial mitral annular diastolic velocity (E/e'). Pulmonary arterial systolic pressure (PASP) is calculated from the modified Bernoulli equation as $4 \times$ peak TR velocity plus estimated right atrial pressure. Increased left ventricle diastolic posterior wall thickness (LVPWT) implies that the patient suffered left ventricular hypertrophy (LVH). Left ventricular geometry is often classified using relative wall thickness (RWT), calculated as twice the LV posterior wall thickness divided by the LV internal diameter at end-diastole (LVPWT $\times$ 2/LVIDD), and using left ventricular mass index (LVMI), normalized to body surface area or height. Four patterns are described: normal (normal LVMI, RWT $\leq 0.42$), concentric remodeling (normal LVMI, RWT $> 0.42$), concentric hypertrophy (increased LVMI, RWT $> 0.42$), and eccentric hypertrophy (increased LVMI, RWT $\leq 0.42$). Both concentric LVH and concentric remodeling can be observed in HFpEF patients (Pieske et al, 2020).

## Experimental animals

All animal procedures were performed in accordance with the Guidelines for Care and approved by the Ethics Committee of Animal Research, Peking University Health Science Center

(LA2018088 and LA2022151). Experimental animals were randomly assigned to groups, and investigators were blinded to group allocation during all subsequent procedures whenever possible.

C57BL/6N mice were purchased from Charles River Laboratories (Beijing, China). $SR$-$B1^{flox/flox}$ mice were constructed using CRISPR-Cas9 gene-editing technology by Beijing Viewsolid Biotechnology Co., Ltd. (Beijing, China) from C57BL/6N background mice. TEK receptor tyrosine kinase 2 (Tie2)-Cre and cadherin 5 (Cdh5)-Cre mice are commonly used to obtain endothelial-specific expression. However, as Tie2 is expressed in myeloid cells, target gene knockout occurs in non-ECs. Alternatively, during cardiac development, approximately 80% and 20% of cardiac fibroblasts are derived from epicardial cells and EC transdifferentiation, respectively (Moore-Morris et al, 2014). To accurately study the pathologic function of endothelial SR-B1, the tamoxifen-induced $Cdh5$ promoter-driven recombinant enzyme Cre mouse strain ($Cdh5$-$Cre^{ER}$) was selected to obtain endothelial-specific SR-B1 knockout mice. $Cdh5$-$Cre^{ER}$ mice (Zhan et al, 2023) were obtained from Model Animal Research Center of Nanjing University (Nanjing, China), and crossed with $SR$-$B1^{flox/flox}$ ($S^{fl/fl}$) mice to generate endothelial-specific SR-B1 deletion mice ($S^{fl/fl}$; $Cdh5$-$Cre^{ER}$, $S^{\Delta EC}$). Seven-week-old $S^{fl/fl}$ and $S^{\Delta EC}$ mice were administered intraperitoneally with 100 mg/kg Tamoxifen (T5648, Sigma) dissolved in corn oil every 2 days for a total of five times. R26-stop-tdTomato mice were acquired from the Jackson Laboratory (Stock #007914), and crossed with endothelial-restricted SR-B1 deletion to generate endothelial lineage tracing mice. Mice were randomly assigned to experimental groups. Mice were maintained in a climate-controlled environment with a 12-h light/dark cycle and had free access to food and water, in specific pathogen-free (SPF) conditions.

## Heart failure mouse model

The mouse HFpEF model (i.e., the "two-hit" model) was established as described previously (Chen et al, 2022; Schiattarella et al, 2019). In brief, the HFpEF mice were fed with a Rodent Diet with 60% kcal fat diet (60% HFD; D12492, Research Diet) and administered with 0.5 mg/mL L-NAME (80210, Cayman) water (renewed every 3 days, pH = 7.4), whereas the control mice were fed a rodent diet with 10% kcal from fat (D12450J, Research Diet) and normal water.

HFrEF in mice was induced by transverse aortic constriction (TAC) operation, as described previously (Huang et al, 2022). Briefly, 10-week-old mice were anesthetized by an intraperitoneal injection of 20 μL 1.25% tribromoethanol (Avertin) per gram of body weight. Subsequently, the aortic arch was exposed through a midline incision in the anterior neck, and the transverse aorta between the innominate artery and the left common carotid artery was constricted using a 6-0 silk suture, guided by a blunted 27-gauge needle (0.41 μm in diameter). The mice in the sham group underwent the same surgery with no aorta constriction.

## Cell isolation, culture, and treatment

Primary mouse cardiac endothelial cells (MCECs) were isolated and cultured, as described previously (Liu et al, 2019). In brief, male mice were sacrificed, and their hearts were excised in a sterile hood. Then, cardiac ventricle was minced finely, and digested using a

1640-medium solution containing 550 units/mL collagenase IV (LS004188, Worthington), 150 units/mL collagenase II (LS004176, Worthington), 0.9 units/mL dispase II (LS02109, Worthington), and 50 units/mL DNase I (11284932001, Roche) with gentle agitation at 37 °C for 45 min. Subsequently, the cardiac single-cell suspension was obtained by 70 μM cell strainer filtration (#352350, Falcon). Then, the cardiac cells were incubated with magnetic beads (11035, Invitrogen) pre-coated with anti-mouse CD31 antibody (553370, BD Pharmingen, 1:20) with gentle rotation at 4 °C for 30 min. Finally, a magnetic separator (12320D, ThermoFisher) was used to purify the cardiac endothelial cells. The isolated MCECs were plated on plastic plates coated with poly-L-lysine, and cultured in an optimized low-glucose DMEM medium (11885076, Invitrogen) added with 20% fetal bovine serum (FBS, 10099141C, Gibco), 1% penicillin/streptomycin solution (P7630, Solarbio), 100 μg/mL Heparin (CC112, Macgene), and 100 μg/mL ECGS (CC019, Macgene) at 37 °C under a humidified atmosphere containing 5% $CO_2$.

The passage-1 MCECs were used to perform experiments. To better mimic the niche of endothelial cells in HFpEF pathogenesis, MCECs were treated with high glucose and palmitate along with L-NAME for 48 h, replenishing daily. Specifically, a combination of 25 mM glucose and 100 μM palmitate (Gan et al, 2020) was used to simulate HFpEF endothelial cells subjected to high glucose and high lipid treatment conditions (diabetes or obesity). In addition, 500 μM L-NAME was added to inhibit endothelial NOS activity, which is associated with the occurrence of hypertension in "2-hit" HFpEF mice.

To induce endothelial-to-mesenchymal transition (EnMT), MCECs were treated with 10 ng/mL TGF-β1 (#100-21-10; Peprotech), a combination of 10 ng/mL TGF-β1 and 100 μM hydrogen peroxide ($H_2O_2$) (106058; Beijing Tongguang Fine Chemicals Company), or a combination of 10 ng/mL TGF-β1 and 1 ng/mL IL-1β (200-01B-2#; Peprotech) (Evrard et al, 2016; Xiong et al, 2018). Treatments were administered the following morning and repeated every other day for a total duration of 7 days. Subsequently, RNA levels were assessed using reverse transcription quantitative polymerase chain reaction (RT-qPCR).

Human cardiac microvascular endothelial cells (HCMECs, #6000, ScienCell) were cultured in endothelial cell medium (ECM, #1001, ScienCell) added with 5% FBS, 1% penicillin/streptomycin solution and 1% v/v ECGS solution, and studied at the passage-3. When HCMECs were grown at 50-70% confluence, the cells were transfected with Lipofectamine™ 3000 reagent with siRNA (50 pM or 107 pM) to silence SR-B1 expression. If needed, after 4 h of siRNA transfection, SR-B1 was reintroduced using CMV promoter-driven 3xFlag-homo SR-B1 lentivirus (NM_005505.5, Genechem) at a multiplicity of infection (MOI) of 5. HitransG P infection enhancement reagent (REVG005, Genechem) was used to enhance lentiviral infection efficiency, and medium was changed 5 h post-infection. To induce endothelial activation, cells serum-starved with ECM overnight were incubated with 20 ng/mL ultrapure LPS (L2880, sigma) for 8 h in ECM only added with 5% FBS.

The 293T cell line (Procell, Wuhan, China) was negative for mycoplasma contamination and verified by STR profiling. The mouse endothelial cell line H5V was generously provided by Professor Jie Liu (Shanghai East Hospital, Tongji University School of Medicine, Shanghai 200120, China). The 293T and H5V were cultured in high-glucose DMEM medium (C11995500BT, Gibco) supplemented with 10% FBS and 1% penicillin/streptomycin solution.

## Histology, immunohistochemistry, and cell immunofluorescence

Hearts were perfused retrograde with PBS buffer *via* the aorta using a syringe attached a 27-gauge needle to remove any blood, then fixed in 4% paraformaldehyde (PFA) buffer at 4 °C overnight. Hearts were embedded with paraffin and then sliced into 6μm sections at a level of LV papillary muscle. Subsequently, paraffin sections were stained with hematoxylin and eosin (H&E) for detecting cardiac structure, and a Masson's trichrome stain kit (G1340, Solarbio) was used for detecting cardiac fibrosis. Finally, images were obtained by a digital slide scanner (C13140-01, Hamamatsu Photonics). The cardiac fibrosis area was assessed by using ImageJ 1.48 v (National Institutes of Health, USA).

Immunohistochemistry was performed to detect CXCR3 signaling with anti-CXCR3 antibody (A11294, Abclone, 1:100) and an anti-rabbit DAB detection kit (PV-6001, ZSGB-Bio) in the mouse heart according to the manufacturer's protocol, and hematoxylin staining was conducted to label cell nuclei before dehydration and mounting. Only cells displaying both CXCR3 and hematoxylin signaling were counted as CXCR3-positive cells.

Fluorescence staining was conducted on paraffin-embedded mouse heart sections for CXCR3 (A11294, Abclone, 1:100) and CD3E (A12415, Abclone, 1:100), as well as on optimal cutting temperature (OCT)-embedded frozen human heart sections for SR-B1 (ab217318, Abcam, 1:200) and CD31 (ab28364, Abcam, 1:50), using the Treble-Fluorescence immunohistochemical mouse/rabbit kit (RS00035, Immunoway) following the manufacturer's instructions. For the other confocal fluorescence imaging, the fresh human hearts and mouse hearts embedded with OCT were sliced with a microtome for 7 μm and used. Subsequently, the tissue sections were incubated with anti-SR-B1 (ab217318, Abcam, 1:200), anti-CD31 (553370 (Rat IgG), BD Pharmingen, 1:200; sc-376764 (Mus IgG), Santa Cruz, 1:50), conjugated-FICT Isolectin B4 (IB4; L2895, Sigma, 1:5000), anti-cTnI (ab10231, Abcam, 1:200), anti-cTnT (ab8295, abcam, 1:200), anti-αSMA (ab7818, abcam, 1:100), anti-PDGFRα (af1062, R&D, 1:100), anti-tdTomato (ARG55724, Arigo, 1:200) antibody at 4 °C overnight, followed by corresponding goat-source secondary antibody (Invitrogen) for 1 h at room time (RT). Finally, all sections were mounted with the antifade mounting medium with DNA-staining DAPI (SL1841, Coolaber).

Primary HCMECs cultured on a glass-bottom dish were fixed in 4% PFA for 15 min at RT, then permeabilized with 0.1% Triton-100 in PBS for 10 min at RT. Next, 5% BSA in PBS was used to block non-specific antigens for 1 h at RT, then incubated with anti-IRF1 antibody (#8478, CST, 1:200) at 4 °C overnight. The next day, cells were washed with PBST (0.1% Tween20 in PBS) at least three times at RT for 15 min, then incubated with Alexa Fluor 555-conjugated goat anti-rabbit IgG (H + L) antibody (A21428, Invitrogen, 1:500) for 1 h at RT. Finally, cells were mounted with containing DAPI medium (SL1841, Coolaber). Immunostaining images were acquired on a TCS SP8 scanning confocal microscope (Leica, Germany) and Leica Application Suite X v3.5.6.21481 (Leica, Germany).

For negative controls, tissue sections and cultured cells were incubated with antibody diluent without the primary antibody, following an otherwise identical experimental protocol. All negative controls for immunofluorescence imaging are presented in Appendix Fig. S14.

## Real-time quantitative PCR and western blotting

Prior to RNA or protein extraction, the frozen left cardiac ventricle was ground and homogenized in liquid nitrogen. Total RNA from cells and homogenized heart tissue was isolated by using TRIzol reagent (15596018, Invitrogen), and cDNA was synthesized by using PrimeScript RT master mix (RR036A, Takara). RT-qPCR was conducted by using TransStart Top Green qPCR SuperMix (AQ131, TransGen Biotech, China) and a specific primer on an AriaMax Real-Time PCR instrument (Agilent Technologies, Singapore). The 2-ΔΔCt relative quantification method was used to analyze mRNA relative expression level. The primers used in this study are listed in the method "*Reagents and tools table*".

Left cardiac ventricle was prior to protein extracted. Total protein was extracted by using a RIPA lysis buffer (P0013C, Beyotime, China) containing a protease inhibitor cocktail (04693132001, Roche). The BCA protein assay (23227, Pierce) was used to quantify protein concentration. Equal amounts of protein samples were separated by sodium dodecyl sulfate polyacrylamide gel electrophoresis (SDS-PAGE) on 10% or 12% gels, then transferred to NT nitrocellulose membranes (66485, PALL) or 0.2-μm PVDF transfer membrane (ISEQ00010, Millipore). Subsequently, the membranes were block in TBST (0.1% Tween20 in TBS) with 5% non-fat milk at RT for 1 h, then incubated with anti-GAPDH (6004, Proteintech, 1:5000), anti-SR-B1 (ab217318, Abcam, 1:2000), anti-CD31 (ab28364, Abcam, 1:1000), anti-Cdh5 (ab33168, Abcam, 1:1000), anti-β-Actin (66009, Proteintech, 1:5000), anti-CXCL10 (AF-466-NA, R&D, 1:1000), Anti-CXCR3 (ab288437, Abcam, 1:1000), and anti-IRF1 (#8478, CST, 1:2000) antibody at 4 °C overnight. The next day, the membranes were incubated with corresponding HRP-conjugated anti-mouse (330, MBL, 1:5000), anti-rabbit (458, MBL, 1:5000), or anti-goat (SA00001-3, Proteintech, 1:5000) second antibody at RT for 1 h. Finally, the membrane was detected by using ECL substrate (32209, Pierce) or super-sensitive ECL luminescence reagent (MA0186, Meilunbio) on the ChemiDoc XRS+ imaging system (Bio-Rad). The software Image J 1.48 v (National Institutes of Health, USA) was used to quantify protein abundance.

## Murine echocardiography analysis

The echocardiography methods for measuring the cardiac function have been described previously (Chen et al, 2022; deAlmeida et al, 2010). In brief, mice were shaved the day before performing echocardiography. Firstly, to confirm the successful ligation of the transverse aorta, aortic flow velocity (Vmax) across the constriction site was assessed by color and PW Doppler of the Vevo2100 ultrasound system (VisualSonics, Canada) or rodent ultrasound system (Vevo F2) 1 week after TAC operation, and further converted to the pressure gradient according to the modified Bernoulli equation $4 \times Vmax^2$. To evaluate cardiac contraction of HFrEF and HFpEF mice, 1.5% isoflurane inhalant mixed with 1 L/min 100% $O_2$ was applied to make the heart rate stabilizing up to 500 bpm per minute. Next, the maximum cardiac LV length was identified through parasternal long-axis images acquired in B-Mode. Then, the transducer was rotated orthogonal to the left parasternal short-axis view at the papillary muscle level where systolic function was obtained in M-Mode. Ejection fraction (EF) was measured using the Vevo F2 system. For measuring diastolic function, the isoflurane level was increased to stabilize the heart rate to 300-350 bpm and slow the mitral valve. Subsequently, at the apical four-chamber view, diastolic function was obtained by using 20-MHz pulsed-wave and tissue Doppler probe. Finally, Parameters included aortic flow velocity, heart rate, LVEF, LVFS, LVPWd, LV Mass, LVRI, MPI, and E/e', were assessed with Vevo Lab software (Version 3.1.1, VisualSonics, Canada). All parameters were measured at least three times, and means were presented.

## Intraperitoneal glucose-tolerance test

After 8-h fasting, 200 mg per mL glucose in saline was injected intraperitoneally. Glucose concentrations in mice tail blood were measured using a glucose meter before (0) and at 15, 30, 45, 60, and 120 min after glucose administration.

## Mice blood pressure

A non-invasive tail-cuff method was performed to evaluate the mice systolic and diastolic blood pressure through a CODA-8 blood pressure system instrument (Kent Scientific, USA). Before the test, mice were trained for 3 days to adapt to short-term constraints. Firstly, the air tightness of O-cuff and VPR-cuff was pre-tested to ensure normal working state. Then, the heating platform was preheated to 37 °C, then the mouse was placed and confined in the tube as comfortable as possible. Next, blood pressure was measured at the same time every day for at least 4 consecutive days. Finally, the records were collected at least eight times a day per mouse using the CODA v4.1 software, and the mean of each day's measurements was used to represent the blood pressure of the mouse.

To confirm the blood pressure data based on the non-invasive tail-cuff method, 24-h blood pressure was measured using remote radio-telemetry as previously described (Prysyazhna et al, 2012). In brief, radiotelemetric transmitter (HD-X11, Data Sciences International) was surgically implanted in the aortic arch *via* the left carotid artery. After 1 week of individual housing for postoperative recovery, the mouse cages were placed on telemetric receivers. Blood pressure data were then collected through scheduled sampling using the DSI Dataquest A.R.T. telemetry system. Mean arterial pressure values were calculated to generate 24-h blood pressure profiles for quantitative analysis.

## Exercise endurance test

Mice single lane Touchcreen treadmill (Panlab Harvard Apparatus, USA) was employed to investigate the cardiac reserve capacity. Firstly, in the running device with upper slope 20°, mice were trained for 3 days. After fitness training, mice ran at a starting speed of 8 cm/s, and its running speed increased to 11 cm/s for 2 min after 4 min. Subsequently, the speed increased by 3 cm/s every 2 min, with a maximum speed of 35 cm/s, until the mice were exhausted. "Exhausted" refers to the inability of mice to continue

running after ten times of direct contact with the electrically stimulated grid within 10 s.

## Flow cytometry

The endothelial-to-mesenchymal transition was assessed in lineage-tracing mice using flow cytometry. The cardiac single-cell suspension was obtained as the above method "Cell Isolation and Culture". 1 million cells were stained with violet fluorescent Live/Dead reactive dye (L34963, Invitrogen) and a FITC-conjugated anti-mouse CD31 antibody (#102406, Biolegend, 1:100) in a 1 mL of PEB butter (0.5% BSA, and 2 mM EDTA in PBS) for 30 min at 4 °C, respectively. Cells were washed twice before analysis. The negative control and CD31-positive control signals were acquired based on isolated cells from the hearts of a Rosa26-stop-tdTomato; Cdh5-Cre ER mice without Tamoxifen injection. The RFP-positive control signals were obtained based on isolated cells from the hearts of a Rosa26-stop-tdTomato; Cdh5-Cre ER mice 2 weeks after Tamoxifen injection. Compensated data were manually gated to remove debris, non-viable cells, and doublets on a fluorescence-activated cell sorter (FACS) SymphonyS6 (BD, USA). Then, violet fluorescent reactive LIVE/DEAD dye (Ex. 405 nm and Em. 431/28 nm) was used to further exclude dead cells in FACS analysis. FITC channel (Ex. 488 and Em. 515/30) and PE channel (Ex. 561 and Em. 585/15) were employed to detect CD31 and tdTomato signals, respectively. FACSDiva software v9.0 (BD, USA) was used to analyze data.

Changes in immune cell populations in the cardiac left ventricle (LV) and blood were assessed using quantitative flow cytometry. Blood was collected in heparinized tubes (YA1471, Solarbio) after mice were anesthetized. The thoracic cavity was opened, and the heart was perfused with 1640 medium to remove blood. Approximately 30 mg of minced LV tissue was digested for 15 min at 37 °C with 3 mL of solution containing collagenase type IV (300 U/mL, Worthington LS004188) and DNase I (100 U/mL, Solarbio D8071). The isolated cardiac cells and blood cells were subjected to red blood cell lysis buffer (R1010, Solarbio). The lysed cardiac cell suspension was resuspended in 1 mL PBS, while the lysed blood was resuspended in 500 μL PBS. Cells were then stained with Zombie Fixable Viability Kit (423112, BioLegend, 1:1000) for dead cell discrimination and incubated in the dark at 4 °C for 15 min, followed by centrifugation at 800 ×g for 5 min. After Fc blocking with anti-CD16/CD32 (101320, BioLegend, 1:100) for 10 min, cell surface staining was performed in the dark at 4 °C using antibodies against CD3 (CD3e (100336, BioLegend, 1:40) for cardiac cells, CD3 (100234, BioLegend) for blood cells, 1:100), CD11b (563015, BD Pharmingen, 1:200), CD45 (103132, BioLegend), and CXCR3 (126512, BioLegend, 1:100). Samples were washed with FACS buffer and resuspended for flow cytometry analysis. Precision Count Beads (BioLegend 424902) were added to enable absolute cell counting. Flow cytometry data were acquired on a four-laser CytoFLEX LX Flow Cytometer (Beckman Coulter) using CytExpert software and analyzed with the same software.

## Single-cardiac-endothelial-cell RNA sequencing (scecRNA-seq) and analysis

RFP-positive cells from isolated cardiac left ventricle were obtained from FACS, then subjected to single-cell RNA sequencing and analyses by NovelBio Bio-pharm Technology Co., Ltd. The scRNA-Seq libraries were generated using the 10X Genomics Chromium Controller Instrument and Chromium Single Cell 3' V3 Reagent Kits (10X Genomics, Pleasanton, CA). Briefly, cells were concentrated to approximately 1000 cells/μL and loaded into each channel to generate single-cell Gel Bead-In-Emulsions (GEMs). After the RT step, GEMs were broken, and barcoded-cDNA was purified and amplified. The amplified barcoded cDNA was fragmented, A-tailed, ligated with adapters and index PCR amplified. The final libraries were quantified using the Qubit High Sensitivity DNA assay (Thermo Fisher Scientific) and the size distribution of the libraries were determined using a High Sensitivity DNA chip on a Bioanalyzer 2200 (Agilent). All libraries were sequenced by Illumina sequencer (Illumina, San Diego, CA) on a 150 bp paired-end run.

scecRNA-seq data analysis was performed by NovelBio Co.,Ltd. with NovelBrain Cloud Analysis Platform (www.novelbrain.com). We applied fastp with default parameters, filtering the adapter sequence and removed the low-quality reads to achieve clean data. Then, the feature-barcode matrices were obtained by aligning reads to the mouse genome (mm10, Ensembl 100) using CellRanger v5.0.1. We applied the downsample analysis among samples sequenced according to the mapped barcoded reads per cell of each sample and finally achieved the aggregated matrix. Cells contained over 200 expressed genes and mitochondria UMI rate below 20% passed the cell quality filtering and mitochondria genes were removed in the expression table.

Seurat package (version: 3.2.4, https://satijalab.org/seurat/) was used for cell normalization and regression based on the expression table according to the UMI counts of each sample and percent of mitochondria rate to obtain the scaled data. PCA was constructed based on the scaled data with the top 2000 high variable genes and the top 10 principals were used for tSNE construction and UMAP construction. Utilizing the graph-based cluster method, we acquired the unsupervised cell cluster result based the PCA top 10 principal and we calculated the marker genes by FindAllMarkers function with Wilcoxon rank-sum test algorithm under the following criteria:1. lnFC > 0.25; 2. P value < 0.05; 3. min.pct>0.1. In order to identify the cell type detailed, the clusters of the same cell type were selected for re-tSNE analysis, graph-based clustering and marker analysis. Potentially contaminating cells were excluded based on the expression of EC markers (i.e., platelet and EC adhesion molecule 1 and Cdh5). Cardiac ECs were re-clustered and identified according to previously reported marker genes (Kalucka et al, 2020), including capillary, artery, vein, and lymphatic ECs.

To characterize the relative activation of a given gene set, such as pathway activation, we performed QuSAGE (2.16.1) analysis. To identify differentially expressed genes among samples, the function FindMarkers with Wilcoxon rank-sum test algorithm was used under the following criteria:1. lnFC > 0.25; 2. P value < 0.05; 3. min.pct>0.1. To discover the gene co-regulation network, find_gene_modules function of monocle3 was used with the default parameters. Gene ontology (GO) analysis was performed to facilitate elucidating the biological implications of marker genes and differentially expressed genes. We downloaded the GO annotations from NCBI (http://www.ncbi.nlm.nih.gov/), UniProt (http://www.uniprot.org/) and the Gene Ontology (http://www.geneontology.org/). Fisher's exact test was applied to identify the significant GO categories, and FDR was used to correct the P

values. Pathway analysis was used to find out the significant pathway of the marker genes and differentially expressed genes according to the KEGG database. Fisher's exact test was used to select the significant pathway, and the threshold of significance was defined by *P* value and FDR.

## The expression bubble map of *Cxcr3* in the heart

The scRNA-seq dataset GSE236585 (Data ref: Li et al, 2023) was downloaded from the GEO database and included 3 HFpEF and 3 control ventricular samples from male C57BL/6 N mice. Analyses of the count data were performed in R version 4.3.1 using Seurat suite versions 3.0. Quality control was performed sample-wise by filtering out cells with <500 or >7000 feature numbers, >20% mitochondrial genes, and genes that were expressed in less than 3 cells. Data were normalized using the NormalizeData function. The 2000 highest variable genes were identified *via* the FindVariableFeatures method and selected to calculate principal components (PCs). The top 30 PC embeddings were adjusted with the harmony R-package. The harmonized embeddings were utilized for non-linear dimensionality reduction *via* UMAP. A shared nearest neighbor (SNN) graph was constructed based on the harmonized PCA space, followed by cell clustering with clustering resolution 1.5. Final clusters were visualized on UMAP embeddings. Cell type markers were calculated with the FindAllMarkers function in Seurat and cell types were manually annotated based on known heart cells markers. The expression of *Cxcr3* gene across all cell types in mixed samples was visualized using Dotplot function.

## Transcription factor prediction and chip-qPCR

The method was performed as described previously (Zhou et al, 2022). First, homo CXCL10 promoter sequence was identified *via* Ensembl website (http://asia.ensembl.org/index.html) located on 76023600-76023801 of chromosome 4. Then, the promoter sequence was loaded on PROMO website tool (https://alggen.lsi.upc.es/cgi-bin/promo_v3/promo/promoinit.cgi?dirDB=TF_8.3) (Farre et al, 2003; Messeguer et al, 2002) to predict potential transcription factors. The detailed binding site was displayed on the PROMO tool, and further validated by the JASPAR website (https://jaspar.genereg.net). Predicted site was located in CXCL10 Promoter at 79–99 bp, so Chip-qPCR primer for CXCL10 promoter was designed. Chromatin immunoprecipitation (Chip) assay was performed in HCMECs using the CHIP KIT (ab500, abcam) combined with anti-IRF1 antibody (#8478, CST, 1:50) according to the manufacturer's instructions. A anti-Histone 3 antibody (#925t, CST, 1:50) was used as a positive control. qPCR was conducted by using TransStart Top Green qPCR SuperMix (AQ131, TransGen Biotech, China) and specific primer on an AriaMax Real-Time PCR instrument (Agilent Technologies, Singapore). The 2-ΔCt relative quantification method was used to analyze CXCL10 promoter abundance. The PCR product yield was also assessed by agarose gel electrophoresis.

## Dual-luciferase reporter assays

The capacity of IRF1 to upregulate CXCL10 transcripts was evaluated as described previously (Cui et al, 2021). In brief, HEK293T were transiently co-transfected with pGL3-firely-por-moter-hCXCL10, pRL-TK Renilla and pcDNA3.1-IRF1-3xflag or pcDNA3.1-3xFlag plasmids by Lipofectamine™ 3000 reagent. After 1 day, transfected cells were harvested for luciferase activity assay using Dual-Luciferase® Reporter assay system (E1910, Promega). The transcriptional activity was indicated as the ratio of firefly luciferase activity to renilla luciferase activity.

## Generation and administration of adeno-associated virus (AAV)

To achieve endothelial-specific gene overexpression, we first cloned CXCL10 cDNA (NM_021274: 76–372), SR-B1 cDNA (NM_016741.3: 137–1666 bp), and control ZsGreen into the endothelial-specific expression vector pAAV-Cdh5. Given the lack of reported AAV serotypes that efficiently target cardiac endothelial cells, we preselected four serotypes—AAV1-cap, AAV2-cap, RGDLRVS-AAV9-cap (generously provided by Professor Yu Huang, School of Biomedical Sciences, Chinese University of Hong Kong), and AAV9-cap (as a control)—based on previous studies (Chen et al, 2005; Huang et al, 2021; Qi et al, 2012; Varadi et al, 2012). The pAAV-Cdh5-ZsGreen vector was packaged into each of these four AAV serotypes and subsequently purified by Bio-lifespan Co., LTD. These AAVs expressing ZsGreen were then administered at varying doses ($0.5 \times 10^{12}$, $1 \times 10^{12}$, and $3 \times 10^{12}$ vector genomes (vgs) per mouse, with an average body weight of 20 g) *via* tail vein injection. Immunofluorescence and western blot analyses were employed to identify the optimal AAV serotype and injection dose. Ultimately, AAV1 was selected for constructing AAVs expressing CXCL10, SR-B1, or ZsGreen (as a control), referred to as AAV1-Cdh5-SR-B1, AAV1-Cdh5-CXCL10, and AAV1-Cdh5-ZsGreen, respectively. These were administered at a dose of $1.5 \times 10^{11}$ vgs per gram of mouse body weight.

## RGD-peptide nanoparticle endothelial delivery system

The delivery system was applied in vivo to achieve specific gene knockdown in mouse cardiac endothelial cells (Liu et al, 2019). Briefly, RGD-peptide synthesized by GL Biochem Ltd. (Shanghai, China) was conjugated to PEI-Fe$_3$O$_4$ magnetic nanoparticles (Mag2100, NANOEAST, Nanjing, China) *via* carbodiimide-mediated amidation (using EDC/NHS), hereafter referred to as RGD nanoparticles. The endothelial-targeting specificity of RGD nanoparticles is attributed to the RGD-peptide, which serves as a ligand for integrins (highly expressed on endothelial cells). To validate the endothelial-targeting capability, RGD-peptide nanoparticles were labeled with Cyanine5.5 NHS ester (C171354, Aladdin, China). Subsequently, synthetic mouse IRF1-siRNA or scramble-siRNA (Tsingke, China) was complexed with RGD-peptide nanoparticles at a mass ratio of 1:5 in RNase-free saline for 1 h at room temperature, yielding IRF1-knockdown endothelial nucleic acid drugs. For administration, RGD nanoparticles loaded with siRNA (dose: 2 µg siRNA per gram of body weight) were injected *via* the tail vein every 3 days.

## Statistical analysis

GraphPad Prism v.9.0.0 (GraphPad, La Jolla, CA, USA) was used for statistical analyses. All experiments were performed with at least three biological replicates, unless otherwise stated. Normality of the data and homogeneity of variance were assessed *via* the Shapiro–Wilk test and F test, respectively. For normally distributed

**The paper explained**

**Problem**

Heart failure with preserved ejection fraction (HFpEF) is a common heart disease with limited treatment options. This study investigates how dysfunction of cardiac blood vessel cells contributes to HFpEF development.

**Results**

SR-B1 protein, mainly expressed in heart blood vessel cells, was found to be significantly reduced in HFpEF. Its loss triggers inflammatory activation through the CXCL10 secretion. This inflammation recruits CXCR3 + T-cells into the heart via CXCL10 signaling, worsening heart relaxation. Restoring SR-B1 in blood vessel cells prevented these changes and improved heart function in experimental models.

**Impact**

The study identifies the SR-B1–CXCL10–CXCR3 axis as a key driver of HFpEF. Blood CXCL10 levels may serve as a diagnostic marker, and targeting this pathway offers promising new therapeutic strategies for HFpEF treatment.

data, values are presented as mean ± standard error of the mean, and the standard Student's $t$ test (equal variance) or Welch's $t$ test (unequal variance) was used to determine differences between two groups, and one-way Analysis of Variance (ANOVA) followed by Bonferroni post-hoc test was used for testing differences among more than two groups. For data that were not distributed normally or did not display homogeneity of variance, values are presented as medians and interquartile ranges (IQR). The Mann–Whitney $U$ test was applied for two groups, and the Kruskal–Wallis with Dunn multiple comparisons test was used for more than two groups. For the analysis of longitudinal blood pressure data, a mixed-effects model with repeated measures was employed to assess differences. In the results with more than two groups and two factors involved, a two-way ANOVA test followed by a Bonferroni post-hoc test was used to analyze differences. The categorical variables are expressed as numbers and percentages (%). The Chi-Square test was used for categorical variables. The association of CXCL10 and NT-proBNP was evaluated using bivariate linear correlation (Pearson correlation) analysis after logarithmic transformation, and the relationship between CXCL10 and HDL-C was assessed using Spearman correlation analysis. Univariate logistic regression analyses were performed to identify risk factors for HFpEF pathogenesis. Variables with $P$ values < 0.05 after univariate logistic regression analyses were introduced into a multivariate logistic regression analysis. Odds ratios (OR) with 95% confidence intervals (CI) were calculated for the relative risk. A $P$ value < 0.05 was considered statistically significant. The respective sample sizes ($n$) and statistical methods are provided in the figure legends. To maintain figure clarity, partial $P$ value ranges are shown in the figures, with the corresponding exact $P$ values listed in Appendix Table S4.

## Data availability

Our single-cell RNA-seq data have been deposited in the GEO database under accession number GSE317186. To access these data, visit: https://www.ncbi.nlm.nih.gov/geo/query/acc.cgi?acc=GSE317186.

The source data of this paper are collected in the following database record: biostudies:S-SCDT-10_1038-S44321-026-00405-9.

## Peer review information

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

## Acknowledgements

This study was supported by the National Key R&D Program of China (2023YFA1800904; 2025ZD0549600); the National Natural Science Foundation of China (U25A2011); the Natural Science Foundation of Beijing, China (L232031; J230039, 7242163) to LZ and MZ, by grants from Fundamental Research Funds for the Central Universities, Peking University Medicine Sailing Program for Young Scholars' Scientific & Technological Innovation (BMU2023YFJHPY027) and the National Natural Science Foundation of China (U22A20286) to YW, by grants from the National Nature Science Foundation of China (81870364), Guangdong Pharmaceutical University Shenzhen Hospital (Longgang) and Guangdong Pharmaceutical University Joint Fund (University joint fund) project (LGSY 202302), Special Fund for Medical and Health Technology Innovation Projects, Longgang District Science and Technology Innovation Program, Shenzhen (Funding Grant: LGKCYLWS2022035, LGKCYLWS2025-34; Non-Funding Grant: LGWJ2024-93), and Key Medical Discipline Construction Site in Longgang District to MX, by grants from the National Natural Science Foundation of China (82201314), and Fundamental Research Funds for the Central Universities starting fund (BMU2022RCZX038) to TW, and by grants from the China Postdoctoral Science Foundation (No. 2021M691944) and Shandong Provincial Natural Science Foundation (ZR2022MH052) to XY.

## Author contributions

**Yufei Wu**: Conceptualization; Resources; Data curation; Formal analysis; Funding acquisition; Investigation; Visualization; Methodology; Writing—original draft. **Xiaomei Yang**: Conceptualization; Data curation; Formal analysis; Supervision; Funding acquisition. **Yu Bai**: Resources; Data curation; Formal analysis; Visualization; Methodology; Writing—original draft. **Chenze Li**: Resources; Methodology. **Peng Wang**: Data curation; Methodology. **Qing Xu**: Data curation; Methodology. **Hui Li**: Data curation; Formal analysis; Methodology. **Xiaoli Rao**: Resources; Methodology. **Yangkai Xu**: Investigation; Visualization. **Jie Chen**: Resources; Data curation. **Huanhuan Cao**: Investigation. **Qi Zhang**: Investigation. **Mingming Zhao**: Supervision; Funding acquisition; Methodology. **Rui Zhan**: Supervision; Methodology. **Xue Fan**: Investigation. **Yuedong Hou**: Data curation. **Jie Liu**: Methodology. **Hong S Lu**: Writing—review and editing. **Tianyun Wang**: Resources; Funding acquisition; Methodology. **Wei Dong Gao**: Writing—review and editing. **Linzhang Huang**: Supervision; Methodology. **Han Xiao**: Methodology. **Lingyun Zu**: Methodology. **Alan Daugherty**: Writing—review and editing. **Mingguo Xu**: Conceptualization; Supervision; Funding acquisition; Investigation; Writing—review and editing. **Lemin Zheng**: Conceptualization; Supervision; Funding acquisition; Investigation; Project administration; Writing—review and editing.

Source data underlying figure panels in this paper may have individual authorship assigned. Where available, figure panel/source data authorship is listed in the following database record: biostudies:S-SCDT-10_1038-S44321-026-00405-9.

## Disclosure and competing interests statement

The authors declare no competing interests.

