## [Peer Review File · EMBO Molecular Medicine]

Microvascular endothelial SR-B1 protects against HFpEF by inhibiting T-cell cardiotropism

Yufei Wu, Xiaomei Yang, Yu Bai, Chenze Li, Peng Wang, Qing Xu, Hui Li, Xiaoli Rao, Yangkai Xu, Jie Chen, Huanhuan Gao, Qi Zhang, Mingming Zhao, Rui Zhan, Xue Fan, Yuedong Hou, Jie Liu, Hong Lu, Tianyun Wang, Wei Dong Gao, Linzhang Huang, Han Xiao, Lingyun Zu, Alan Daugherty, Mingguo Xu, and Lemin Zheng

Corresponding authors: Lemin Zheng (zhengl@bjmu.edu.cn) , Mingguo Xu (docjxzhu@stu.edu.cn)

Review Timeline:

Submission Date:	8th Jan 26
Editorial Decision:	12th Jan 26
Revision Received:	22nd Jan 26
Editorial Decision:	9th Feb 26
Revision Received:	10th Feb 26
Accepted:	19th Feb 26

Editor: *Zeljko Durdevic*

Transaction Report:

(Note: The manuscript was previously reviewed at another journal. As EMBO Press has a transfer agreement with that journal, revision was invited based on the reports from that previous external submission. With the exception of the correction of typographical or spelling errors that could be a source of ambiguity, letters and reports are not edited. Depending on transfer agreements, referee reports obtained elsewhere may or may not be included in this compilation. Referee reports are anonymous unless the Referee chooses to sign their reports.)

12th Jan 2026

Dear Prof. Zheng,

Thank you for the submission of your revised manuscript to EMBO Molecular Medicine. I am pleased to inform you that we will be able to accept your manuscript pending reformatting in accordance with the journal's requirements. Please check the points below as well as our Author Guidelines for more information

<https://www.embopress.org/page/journal/17574684/authorguide#manuscriptpreparation>. Please note that all information about material and methods must be included in the main Methods section of the manuscript. In addition, for the next submission please upload a point-by-point response to the last referees' report. Further, when submitting the revised manuscript please be sure to add institutional email address for Mingguo Xu in the manuscript and our submission system

I look forward to receiving a revised version of your manuscript as soon as possible.

Yours sincerely,

Zeljko Durdevic

Zeljko Durdevic
Senior Editor
EMBO Molecular Medicine

*** Instructions to submit your revised manuscript ***

When preparing your revised manuscript, please refer to our guidelines: <https://link.springer.com/journal/44321/submission-guidelines#cms-Revised-submissions>. We perform an initial quality control of all revised manuscripts before re-review; failure to include requested items will delay the evaluation of your revision.

We require:

- 1) A .docx formatted version of the manuscript text (including legends for main figures, EV figures and tables). Please make sure that the changes are highlighted to be clearly visible.
- 2) Individual production quality figure files as .eps, .tif, .jpg (one file per figure). For guidance, download the 'Figure Guide PDF': <https://media.springernature.com/original/springer-cms/rest/v1/content/27825798/data/v1>.
- 3) A .docx formatted letter INCLUDING the reviewers' reports and your detailed point-by-point responses to their comments. As part of the EMBO Press transparent editorial process, the point-by-point response is part of the Review Process File (RPF), which will be published alongside your paper.
- 4) A complete author checklist, which you can download from our author guidelines. Please insert information in the checklist that is also reflected in the manuscript. The completed author checklist will also be part of the RPF.

6) It is mandatory to include a 'Data Availability' section after the Materials and Methods. Before submitting your revision, primary datasets produced in this study need to be deposited in an appropriate public database, and the accession numbers and database listed under 'Data Availability'. Please remember to provide a reviewer password if the datasets are not yet public.

7) For data quantification: please specify the name of the statistical test used to generate error bars and P values, the number (n) of independent experiments (specify technical or biological replicates) underlying each data point and the test used to calculate p-values in each figure legend. The figure legends should contain a basic description of n, P and the test applied. Graphs must include a description of the bars and the error bars (s.d., s.e.m.).

9) Our journal encourages inclusion of *data citations in the reference list* to directly cite datasets that were re-used and obtained from public databases. Data citations in the article text are distinct from normal bibliographical citations and should directly link to the database records from which the data can be accessed. In the main text, data citations are formatted as follows: "Data ref: Smith et al, 2001" or "Data ref: NCBI Sequence Read Archive PRJNA342805, 2017". In the Reference list, data citations must be labeled with "[DATASET]". A data reference must provide the database name, accession number/identifiers and a resolvable link to the landing page from which the data can be accessed at the end of the reference.

12) Author contributions: You will be asked to provide CRediT (Contributor Role Taxonomy) terms in the submission system. These replace a narrative author contribution section in the manuscript.

13) A Conflict of Interest statement should be provided in the main text.

14) Every published paper includes a 'Synopsis' to further enhance discoverability. Synopses are displayed on the journal webpage and are freely accessible to all readers. They include a short stand first (maximum of 300 characters, including space) as well as 2-5 one-sentences bullet points that summarizes the paper. Please write the bullet points to summarize the key NEW findings. They should be designed to be complementary to the abstract - i.e. not repeat the same text. We encourage inclusion of key acronyms and quantitative information (maximum of 30 words / bullet point). Please use the passive voice. Please attach these in a separate file or send them by email, we will incorporate them accordingly.

15) Include a Reagents and Tools Table as part of the Methods section, which can be downloaded from our author guidelines.

Photos 400-800 DPI

*Additional important information regarding figures and illustrations can be found at
<https://media.springernature.com/original/springer-cms/rest/v1/content/27825798/data/v1>

Response to editors' comments

Pages 1 to 23: for Editors

Pages 24 to 46: for Reviewer 1#

Pages 47 to 64: for Reviewer 2#

Pages 65 to 77: for Reviewer 3#

We sincerely appreciate the valuable feedback and insightful suggestions from the editors and reviewers, which have been instrumental in enhancing our work. In the revised manuscript, all modifications have been highlighted in yellow for clarity. Below are our point-by-point responses to the editors' comments, with the original comments in black and our responses in blue.

Next, we carefully address the following highlighting concerns:

Comment 1: Demonstration of the cardiac immune response in deficient or sufficient SR-B1 mice, as suggested by Reviewer #1. (Reviewer #1: "Supplemental Figure 8A is also insufficient to support this axis. Flow cytometric quantitative characterization is needed, and if the conclusion the authors make is supported, then the data should not be hidden in the supplement.")

Response: We sincerely appreciate the reviewer's valuable suggestions. We have incorporated the results from *Supplemental Fig. 8A* into *Fig. 5H* in the revised version. After 20 weeks of HFD plus L-NAME regimens, we performed flow cytometric sorting of immune cells in the hearts of SR-B1-deficient mice. Consistent with the CXCR3+ immunohistochemistry findings (*Fig. 5H*), flow cytometric quantitative analysis revealed that SR-B1 knockout significantly promoted the infiltration of CXCR3+ T cells into the hearts of HFpEF mice, but not CD45+ and CD11b+ immune cell populations (*Fig. 5I-5K, and Extended Data Fig. 7C*). The specific results are detailed on **page 11, lines 11–17** of the revised manuscript, as follows:

"To further confirm that SR-B1 deficiency promotes CXCR3+ T cell infiltration in the heart, flow cytometry analysis was performed (the gating strategy is presented in *Supplementary Fig. 1B*). The results demonstrated that SR-B1 deficiency increased the CXCR3+ T cell/CD3e+ T cell ratio (*Fig. 5I and 5J*), and enhanced the infiltration of both CXCR3+ T cells (*Fig. 5K*) and total T cells (*Extended Data Fig. 7C*) in HFpEF cardiac left ventricle, without affecting the populations of CD45+ and CD11b+ cells (*Extended Data Fig. 7C*)."

Fig. 5I, Representative flow cytometry histograms depicting the percentage of CXCR3+ T cells among CD3+ T cells in cardiac left ventricle of $S^{fl/fl}$ and $S^{\Delta EC}$ HFpEF mice, with corresponding quantitative analysis (**J**, $n=6$, Student's t test). **K**, Flow cytometric quantification of cardiac CXCR3+ T cell ($n=6$, Student's t test).

C Heart from $S^{fl/fl}$ and $S^{\Delta EC}$ HFpEF Mice

Extended Data Fig. 7C, Quantitative flow cytometric analysis of immune cell populations, including CD45+ leukocytes, CD11b+ myeloid cells, and CD3+ T lymphocytes, in the left ventricular tissue of $S^{fl/fl}$ and $S^{\Delta EC}$ HFpEF mice ($n=6$, Student's t test).

Comment 2: Address the lack of cardiomyocytes in the non-CMEC scRNA-seq data set with further experimental evidence such as snRNASeq, as suggested by Reviewer #2. (Reviewer #2: “I am concerned by the apparent lack of cardiomyocytes in their non-CMEC scRNA-Seq data set (myocytes are too large for sorting). Most groups now either use snRNASeq, or isolate cardiomyocyte nuclei prior to scRNASeq. This must be addressed.”)

Response: We sincerely apologize for any lack of clarity in our initial description. The single-cell RNA sequencing (scRNA-seq) in our study was performed using mice with an *RFP^{AEC}* background, in which endothelial cells express red fluorescent protein (RFP) following tamoxifen injection. As shown in **Figure 5A**, sequencing was conducted exclusively on RFP-positive cells, which explains the absence of cardiomyocytes in our dataset.

We fully acknowledge the important concern regarding the technical considerations of cardiomyocyte single-cell sequencing. To clarify, our tissue dissociation method involved mechanical mincing followed by enzymatic digestion at 37°C (as detailed in the **Supplementary Methods** section). This approach does not preserve viable cardiomyocytes, whereas the isolation of living cardiomyocytes typically requires the Langendorff retrograde perfusion technique. However, while the Langendorff method is effective for obtaining viable cardiomyocytes, it can compromise the efficiency of non-cardiomyocyte isolation.

The two objectives of performing single-cell sequencing on mice with RFP-labeled endothelial cells: first, to determine whether cardiac endothelial cells undergo phenotypic transitions during the progression of HFpEF; and second, because tamoxifen-induced endothelial SR-B1 deletion does not occur uniformly across all cells, further sorting of RFP-positive endothelial cells by flow cytometry allows for a more accurate investigation of the underlying molecular mechanisms.

Fig.5 A, A schematic diagram of single-cell transcriptomic sequencing experiment.

Comment 3: Larger sample sizes, as requested by Reviewer #2 and #3.

(1) In response to Reviewer #2's concern about the adequacy of our sample sizes, we provide the following point-by-point clarification:

1) **The major point 1 of Reviewer #2** "Fig 1F shows n=3 only claiming that HFpEF mice have reduced SRB1 but why not a more robust number?"

Response: In the current revised version, Fig. 1F corresponds to Fig. 1E. For Fig. 1E (n = 3), we would like to clarify that each sample (n = 1) represents a pooled mixture of cardiac endothelial cells isolated from at least 2 mice, as indicated in the corresponding figure legend. The isolation method was consistent with that employed for single-cell sequencing of pooled mixed samples.

Moreover, in this revised manuscript, we also investigated the change of *Scarb1* (SR-B1) transcriptional level in different stages of HFpEF progression, validating significant downregulation of *Scarb1* mRNA level in cardiac endothelial cells in HFpEF mice than control mice (Fig. 1F).

Fig. 1F, *Scarb1* mRNA abundance in mouse cardiac ECs at various time points following administration of a high-fat diet (HFD) combined with L-NAME drink (n=5-6 / group, Welch's or Student's t-test).

2) **The major point 4 of Reviewer #2** "Figure 4. Why only n=3?"

Response: In **Figure 4B**, we have added the sample size (n = 7). As illustrated in the updated figure (**Fig. 4B**), the results remain consistent with our earlier findings, confirming that cardiac coronary endothelial cells can undergo transdifferentiation into myofibroblasts in vitro, and that SR-BI expression is downregulated during this phenotypic transition.

B
Fig.4 B, EnMT-related genes and *Scarb1* (SR-B1) mRNA abundance were measured in MCECs isolated from wild-type (WT) mice following stimulation with different factors *via* RT-qPCR (n=7). Statistical analyses were performed using distinct approaches based on data distribution characteristics: *Coll1a1* and *Col3a1* expression data were analyzed using the Kruskal-Wallis test followed by Dunn's multiple comparisons test; *Acta2* (α SMA) expression was evaluated using Brown-Forsythe and Welch ANOVA tests with Dunnett's T3 multiple comparisons; and the remaining genes were assessed using ordinary one-way ANOVA with Bonferroni's multiple comparisons correction.

3) The minor point 6 of Reviewer #2 “Extended data Fig 2D n=3 insufficient if you are trying to convince me that these groups are not different”

Response: In *Extended Data Fig. 2D*, we have expanded the sample size from 3 to 6. As demonstrated in the updated figure, the results align consistently with our earlier findings, confirming that endothelial SR-B1 deletion mediated by *Cdh5-Cre* does not significantly alter SR-B1 protein levels in macrophages following tamoxifen injection.

Extended Data Fig. 2D, Immunoblotting of SR-B1 in peritoneal macrophages isolated from $S^{fl/fl}$ and $S^{\Delta EC}$ mice 2 weeks after tamoxifen administration (upper panel); densitometric analysis of SR-B1 protein level (n=6, lower panel). Significant difference was evaluated *via* Student's t test.

Figure legend: The newly added Western blot results(n=3)

4) The minor point 7 of Reviewer #2 “*Extended Data Fig. 5 - I am convinced the TAC mice have HFrEF but please increase n to more than 5*”

Response: In the current version, *Extended Data Fig. 5A–L* corresponds to the previous *Fig. 5*. In *Extended Data Figure 5A–L*, we have increased the sample size from $n = 5$ to $n = 8–9$. The results remain consistent with our earlier findings, confirming that endothelial SR-B1 deletion

does not affect the progression of TAC-induced cardiac hypertrophic remodeling, systolic function, or pulmonary edema.

Extended Data Fig. 5 Endothelial-specific SR-B1 is dispensable for acute cardiac systolic dysfunction and pathological remodeling induced by TAC. All cardiac function index parameters obtained from echocardiography were in mice 2 weeks after TAC operation (A–L, $n=8,8,9,9$). A, Representative color Doppler images of aortic arch in sham and TAC mice (upper, the white arrow indicates the ligation site); pressure gradient across the constriction site was calculated according to the modified Bernoulli equation $4 \times V_{max}^2$, and V_{max} is the maximum peak aortic velocity obtained from Pulsed-wave Doppler imaging in mice 1 week after TAC operation (lower). B, Representative M-mode images in the parasternal short axis view. C, FS, left ventricular fractional shortening. D, LVPWd, left ventricular posterior wall diastolic thickness. E, Left ventricular volume at the end of diastole. F, LVIDd, left ventricular internal diastolic diameter. G, LV mass, left ventricular mass. H, LVRI, left ventricular remodeling index. I, Heart rate in parasternal short axis view. J, Body weight. K, Ratio between cardiac ventricle weight and tibia length (TL). L, Ratio between wet lung weight and TL in $S^{fl/fl}$ and $S^{\Delta EC}$ mice sacrificed 2 weeks after TAC operation ($n=8,8,9,9$).

(2) **The point 4 of Reviewer #3** “Figure 2B shows that the knockdown is about 80% at the protein level. How many mice were tested? No qPCR data are shown.”

Response: In Figure 2B (n = 5 vs 3), similar to Figure 1F, the n = 1 actually refers to a pooled sample from 2–3 mice, consistent with the method used for single-cell sequencing of pooled mixed samples.

Furthermore, we have now added qPCR (n=11) data to assess the SR-B1 knockout efficiency in the heart, as shown in Extended Data Fig. 2C.

C

Extended Data Fig. 2C, *Scarb1* (SR-B1) mRNA abundance was assessed in cardiac ventricle of $S^{fl/fl}$ and $S^{\Delta EC}$ mice two weeks post-tamoxifen administration (n=11, Mann-Whitney test).

Comment 4: Demonstration of HFpEF rescue with endothelial SR-B1 delivery, as suggested by Reviewer #2. (The major comment 6 “The authors have explored the molecular mechanisms underlying the impact of SR-B1 deficiency in vivo, and of SR-B1-mediated CXCL10 regulation in vitro. However, they do not investigate whether the phenotype can be rescued by replacing the SR-B1 in CMECs (could be achieved by AAV-mediated approach). They have the capability to do this, as they do show that the HFpEF phenotype is exacerbated in AAV1-CXCL10-injected HFpEF mice. Showing that they can rescue HFpEF with endothelial SR-B1 delivery would really hit the nail on the head when it comes to showing that cardiac microvascular endothelial activation contributes to HFpEF pathogenesis.”)

Response: In our study, we restored SR-B1 expression in endothelial SR-B1-deficient mice ($S^{\Delta EC}$) by delivering SR-B1 mRNA via an AAV1 carrying the Cdh5 promoter-initiated expressing vector, as illustrated in Fig. 3G. Echocardiographic analysis revealed that SR-B1 reintroduction effectively rescued cardiac hypertrophic remodeling and diastolic dysfunction induced by

endothelial SR-B1 deficiency (**Fig. 3H-M**). Treadmill endurance tests further demonstrated that SR-B1 supplementation significantly improved the impaired cardiac reserve caused by SR-B1 deficiency, leading to a notable increase in running distance (**Fig. 3N**). Moreover, the heart-to-tibia length ratio and lung wet weight to tibia length (**Fig. 3O and 3P**) confirmed that SR-B1 reconstitution effectively prevented further deterioration of cardiac function and lung edema. As shown in **Fig. 3Q**, SR-B1 restoration also substantially attenuated the elevation of pathological markers in the heart. In summary, these findings collectively demonstrate that AAV1-mediated SR-B1 restoration effectively rescues the pathological phenotypes associated with endothelial SR-B1 deficiency. The findings related to this section are presented in the revised manuscript on **page 7 (lines 25-29)** and **page 8 (lines 1-5)**.

Fig. 3 (G-Q) Endothelial SR-B1 gain-of-function, achieved through AAV1 tail vein injection in $S^{\Delta EC}$ and $S^{fl/fl}$ mice, was evaluated using echocardiography and pathological analysis (n=6; statistical analyses were performed using one-way ANOVA with Bonferroni's post-hoc test or Kruskal-Wallis test with Dunn's post-hoc test). **G**, Schematic illustration of the experimental design. **H**, FS, left ventricular fractional shortening. **I**, LVPWd, left ventricular posterior wall diastolic thickness. **J**, LV mass, left ventricular mass. **K**, LVRI, left ventricular remodeling index. **L**, Heart rate in parasternal short axis view. **M**, Diastolic function assessed by E/e' ratio. **N**, Running distance. **O**, Ratio between cardiac ventricle weight and TL. **P**, Ratio between wet lung weight and TL. **Q**, mRNA expression levels of *SR-B1*, *Col3a1*, *Col1a1*, and *Bnp* in the left ventricle were quantified using RT-qPCR.

Comment 5: Demonstration of rescue of cardiac function with EC-specific knock out of IRF1, as suggested by Reviewer #3. (The comment 1 “*The authors interpret the CHIP data as causality for IRF1 driving CXCL10, but this is not correct. If the author’s hypothesis is correct, an EC-specific KO of IRF1 should correct the problem and rescue the phenotype.*”)

Response: We sincerely appreciate the valuable suggestions. We will address the concern from the following three aspects.

(1) To investigate the role of IRF1 in the pathological progression of HFpEF, we employed a targeted endothelial delivery system using RGD-peptide nanoparticles (RGD-Nano) to specifically knock down IRF1 in cardiac endothelial cells of mice, as previously described (*Liu et al., Circulation, 2019*)¹. Initially, we conducted *in vitro* screening using the mouse endothelial cell line H5V and identified an effective siRNA sequence targeting *Irf1* (siIrf1) (*Supplementary Table I*). This sequence significantly downregulated *Cxcl10* expression to 29% of baseline levels, as confirmed by RT-qPCR (*Extended Data Fig. 9A*). To ensure the specificity of the nanomaterial delivery to cardiac endothelial cells, we conjugated Cy5.5 to RGD-Nano and performed fluorescence imaging on cardiac slices. The results demonstrated that the nanomaterials specifically targeted CD31-positive cardiac endothelial cells (*Extended Data Fig. 9B*). Subsequently, we isolated cardiac endothelial cells from mice injected with RGD-Nano formulating either siIrf1 or its scramble control siRNA (siNC). RT-qPCR analysis confirmed the efficient knockdown of *Irf1 in vivo*, which was accompanied by a significant reduction in *Cxcl10* expression (approximately 50%) (*Extended Data Fig. 9C*), consistent with our *in vitro* findings.

To evaluate the therapeutic potential of IRF1 knockdown, we subjected 10-week-old mice to a 10-week HFD plus L-NAME treatment to induce HFpEF, followed by intravenous injection of siRNA-formulating RGD-Nano particles every three days (*Extended Data Fig. 9D*). Prior to treatment, there were no significant differences in body weight, blood glucose, or blood pressure between the two HFpEF groups (*Extended Data Fig. 9E-9H*). During the 10-week nanomedicine treatment period, the siNC HFpEF group exhibited severe mortality (5/7 mice), whereas the siIrf1 group showed a higher survival rate (1/12 mice). This mortality in the siNC group may be attributed to systolic dysfunction (evidenced by a significant decrease in FS) (*Extended Data Fig. 9I*), because only HFD plus L-NAME regimens did not contribute to a significant reduction in FS, suggesting that long-term RGD-Nano injections might have adversely affected cardiac function. Importantly, pathological examination revealed that IRF1 knockdown significantly attenuated cardiac hypertrophic remodeling (*Extended Data Fig. 9K*), supporting our hypothesis that elevated IRF1 exacerbates cardiac dysfunction in SR-B1-deficient mice. These findings are presented in the revised manuscript on **page 12 (line 29)** and **page 13 (lines 1-29)**.

Extended Data Fig. 9 Knockdown of IRF1 specifically in cardiac endothelial cells significantly ameliorates cardiac pathological remodeling in HFpEF mice. A, Quantitative analysis of *Irf1* and *Cxcl10* mRNA expression levels in mouse endothelial cell line H5V following transfection with either scramble control siRNA (siNC) or IRF1-specific siRNA (siIRF1) ($n = 7$, Student's t-test). B, Representative immunofluorescence microscopy images demonstrating co-localization (indicated by white arrows) of CD31⁺ endothelial cells (green) and Cy5.5-labeled RGD peptide-conjugated magnetic nanoparticles (red) in cardiac tissue sections. Nuclei were counterstained with DAPI (blue). Scale bar, 100 μm . C, Targeted delivery of IRF1-siRNA using RGD-conjugated nanoparticles significantly reduces *Irf1* and *Cxcl10* mRNA expression in isolated cardiac endothelial cells from wild-type mice, as quantified by RT-qPCR ($n = 5$; Student's t-test). D, Workflow of the experimental design to investigate the role of endothelial-specific IRF1 in the HFpEF pathogenesis. E, Body weight ($n=7, 7, 10$). F, Blood glucose during intraperitoneal glucose tolerance test (ipGTT) ($n=7, 7, 10$), with corresponding area under the curve (G). H, Systolic blood pressure ($n=6, 6, 7$) of mice subjected to 10 weeks of HFD plus L-NAME regimens (One-way ANOVA with Bonferroni's multiple comparisons test). At 10 weeks post-injection of RGD nanoparticles, cardiac systolic function parameters were assessed by echocardiography, including (I) fractional shortening (FS) and (J) left ventricular remodeling index (LVRI) ($n= 7, 2, 9$, differences between sham group and HFpEF-siIRF1 group only analyzed using Student's t-test). K, Ratio of cardiac ventricle weight to tibia length (TL) ($n=7, 7, 10$, One-way ANOVA with Bonferroni's multiple comparisons test).

(2) To clarify whether IRF1 rescue CXCL10 expression upregulation following SR-B1 knockdown, we knock-downed the expression of IRF1 in SR-B1 deficient HCMECs in the revised manuscript. As shown in **Fig. 6H**, we observed a significant upregulation of CXCL10 expression in SR-B1-deficient HCMECs, which was reversed upon IRF1 knockout. This result further supports the involvement of IRF1 in SR-B1-mediated CXCL10 upregulation.

Fig. 6H, Representative immunoblot images showing protein expression profiles of SR-B1, IRF1, CXCL10, and GAPDH in HCMECs following transfection with control siNC, siSR-B1, or combined siSR-B1 and siIRF1 (upper panel), and quantitative densitometric analysis of relative protein expression levels for SR-B1, CXCL10, and IRF1 (lower panel, $n = 6$, Brown-Forsythe test and Welch ANOVA with Dunnett's T3 multiple comparisons).

(3) Finally, we would like to clarify the rationale behind our selection of IRF1. In addition to predictions from PROMO (*Extended Data Fig. 9B*), ChIP-qPCR (*Fig. 6F*), and luciferase assays (*Fig. 6F*), our single-cell RNA sequencing data revealed that IRF1 is the top 1 high-expressed gene in capillary subcluster 0 (*Fig. 5E*), a subpopulation of inflammatory microvascular endothelial cells that is a key focus of our study.

Fig. 5E, Heatmap depicts expression levels of differentially expressed marker genes in sub-clusters of cardiac microvascular ECs. Yellow represents high gene expression and purple represents low gene expression.

In summary, our *in vivo* and *in vitro* findings demonstrate that IRF1 plays a critical role in SR-B1-mediated CXCL10 upregulation and contributes to the pathological hypertrophic progression of HFpEF. Through targeted endothelial delivery of IRF1 siRNA using RGD-Nano nanoparticles, we observed significant improvements in cardiac hypertrophy and survival rates in HFpEF mice. However, the high mortality rate and severe systolic dysfunction in the control group precluded a definitive assessment of the effects of IRF1 knockdown on diastolic dysfunction. Therefore, we acknowledge the need for further experimental validation to fully elucidate the role of IRF1 in diastolic dysfunction in HFpEF. Additionally, we have explicitly addressed this limitation in the "Limitations" section of our manuscript on **page 18, lines 14-19**.

Comment 6: Inclusion of appropriate controls, as suggested by Reviewer #3.

(1) The point 2 of Reviewer #3 “*In figure 1A, B and all other immunofluorescence images, specificity controls are missing.*”

Response: We sincerely appreciate the valuable suggestion. All immunofluorescence images, specificity controls for all immunofluorescence experiments are provided in **Supplementary Fig. 4**.

Supplementary Fig. 4 All negative controls for immunofluorescence imaging are presented, with each panel clearly labeled to indicate the corresponding fluorescent image number. Tissue sections were incubated with antibody diluent in the absence of the primary antibody, while all other experimental conditions remained consistent.

(2) **The point 5 of Reviewer #3** “While the floxed mice without *Cre* are a good control, the other control (*Cre*⁺ and *SR-B1*^{fl/+} or ^{+/+}) is also needed, because none of these mice are genetically pure and there could be contributions from the genome around the *Cre* site.”

Response: We sincerely appreciate the valuable suggestion. Regarding the concern about the potential contribution of the genome around the *Cre* site, we consider this unlikely to significantly affect our study findings.

Our single-cell RNA sequencing results are based on a lineage tracing mouse model (as shown in *Fig. 4D* and *Fig. 5A*), where *SR-B1*^{fl/+} and *RFP*^{+/-} with *Cre* mice serve as controls of *SR-B1*^{fl/fl} and *RFP*^{+/-} with *Cre* mice. The key difference between the two groups lies in whether the FloxP sites are present on one or both alleles, rather than the presence or absence of *Cre*. This addresses your concern about the appropriateness of the controls.

Furthermore, the single-cell data, supported by this control model, revealed that the secretory factor CXCL10 and IRF1 are key molecules regulated by SR-B1 in inflammatory microvascular endothelial cells. These findings have been robustly validated in subsequent studies using an alternative control (*S*^{fl/fl} without *Cre*) and *in vitro* cell experiments.

Therefore, we conclude that the SR-B1 deficiency-mediated activation of the IRF1-CXCL10 axis and the resulting infiltration of CXCR3⁺ T cells into the heart are independent of the presence of *Cre* in the control mice.

Fig. 4D, Workflow of experiment to generate endothelial-specific SR-B1 deficient and RFP over-expression mice (a lineage tracing mouse model), investigating the *in vivo*

transdifferentiating fate of endothelial cells in HFpEF in the presence or absence of endothelial SR-B1.

Fig.5 A, A schematic diagram of single-cell transcriptomic sequencing experiment.

Comment 7: Clarification of the time course of EC SR-B1 downregulation and its impact on remodelling, as suggested by Reviewer #1.

Response: We sincerely appreciate the valuable concern. Our investigations into cardiac function and *Scarb1* expression at different stages of HFpEF revealed a progressive decline in *Scarb1* mRNA levels, which correlated with the worsening of HFpEF-associated cardiac hypertrophy and dysfunction.

Fig. 1F, *Scarb1* mRNA abundance in MCECs at various time points following administration of a high-fat diet (HFD) combined with L-NAME drink.

Comment 8 Alignment of the time points for comparison in the HFrEF and HFpEF models, as suggested by Reviewer #1. (The major comment 3 “The authors draw comparisons between preclinical HFpEF and HFrEF by comparing the two hit model to TAC (respectively). However, the time points chosen for analysis are not comparable – the authors measure changes in cardiac pathology in 20 weeks of high fat diet/L-NAME and compare to no changes in cardiac pathology after 2 weeks of TAC. Particularly since the earlier 10 week high fat diet/L-NAME comparisons don’t result in statistically significant changes with SR-B1 deletion, a longer time point post-TAC should be used to convincingly demonstrate this phenotype is HFpEF-specific.”)

Response: We sincerely appreciate this valuable suggestion. From a temporal perspective, it is indeed less rigorous to directly compare the two time points. We attempted to extend the postoperative monitoring period in our model, but this approach proved unsuitable. Specifically, we conducted a new batch of TAC surgeries under standard aortic ligation pressure (using a blunted 27-gauge needle)^{2,3} on SR-B1 endothelial-specific knockout mice, increasing the sample size to n=8–9. Consistent with our previous findings, both TAC groups developed severe dilated HFrEF two weeks post-surgery, as evidenced by a significant reduction in fractional shortening (FS; mean FS <20%, *Extended Data Fig. 5C*) and markedly increased left ventricular internal diastolic diameter (LVIDd, *Extended Data Fig. 5F*). This acute HFrEF phenotype may be strain-specific, as our SR-B1 floxed mice are based on the C57BL/6N strain. According to previous studies, the **Rong Tian group**⁴ has demonstrated that the C57BL/6N strain typically exhibits a rapid pathological progression of acute heart failure compared to the C57BL/6J strain, with a postoperative mortality rate reaching approximately 60% by seven weeks. Therefore, we believe that extending the observation period is not sustainable for our study due to high mortality. Based on the current data, we concur with the suggestion that a direct comparison between preclinical HFpEF and HFrEF is not appropriate. Consequently, we have revised the relevant descriptions in the manuscript and emphasized on the relationship between SR-B1 and acute heart failure. Furthermore, we have added this limitation in the "Limitations" section of our manuscript on **page 18 (lines 8-14)**, as follows:

"The present study has several limitations that warrant discussion. Firstly, acute heart failure and chronic heart failure represent distinct pathological conditions. In our study, the role of SR-B1 was exclusively investigated in the context of acute heart failure. Previous studies have indicated that CXCR3+ T cell infiltration exacerbates cardiac systolic dysfunction⁵, and SR-B1 deficiency facilitates the recruitment of CXCR3+ T cells into the heart. Therefore, SR-B1 may also play a potential role in HFrEF. As such, an ideal preclinical model of chronic HFrEF should be utilized in future research to further elucidate the role of SR-B1 in HFrEF."

Extended Data Fig. 5C, FS, left ventricular fractional shortening.

Extended Data Fig. 5F, LVIDd, left ventricular internal diastolic diameter.

Comment 9: Further mechanistic evidence to support the findings related to Endo-MT, as suggested by Reviewer #1. (The major comment 4 “Additionally, the decrease of SR-B1 in response to TGF-B1 could be a response to treatment or an effect, not necessarily a driver for Endo-MT. Additional mechanistic experiments are required to support this statement related to Endo-MT.”)

Response: Thank you for the valuable advice. We have utilized the gold standard *in vivo* method for studying Endo-MT—an endothelial fluorescence reporter HFpEF mouse model. Both flow cytometry and immunofluorescence results have consistently demonstrated that the presence or absence of endothelial SR-B1 does not induce Endo-MT in the hearts of HFpEF mice (**Fig. 4F and 4G**).

Fig. 4F, Fluorescent images after immunofluorescence staining of α SMA in the cardiac ventricle of $S^{fl/+}/R^{\Delta EC}$ mice fed HFD plus L -NAME. Scale bar, 100 μ m **G**, Single cells from lineage-tracing mice of different groups were FACS sorted according to RFP and CD31 expression. Top panel: negative controls from a heart of $RFP^{\Delta EC}$ mice not injected by tamoxifen (left panel); CD31-positive controls from a heart of $RFP^{\Delta EC}$ mice not injected by tamoxifen (middle panel); RFP-positive controls from a heart of $RFP^{\Delta EC}$ mice 2 weeks after tamoxifen injection (right panel).

Bottom panel, double-channel signals were presented from isolated left ventricular cells in mice from different four groups. RFP signal was detected with PE channel (y axis), while CD31 protein expression was measured through FITC channel (x axis).

Comment 10: Quantification of the SR-B1 expression, as suggested by Reviewer #1. (Reviewer #1: “The authors use histology to demonstrate SR-B1 expression is largely in capillary endothelial cells. This would be strengthened using their single cell data set in which they subset cells as capillary/venous/arterial/lymphatic endothelial cells. Can the authors report SR-B1 expression in each of these clusters to provide quantitative data to support their histological findings?”)

Response: We analyzed the expression of SR-B1 in the NC group ($S^{fl/+}; RFP^{AEC}$) based on our single-cell data, as shown in *Extended Data Fig. 7C and 7D*. The expression of *Scarb1* is significantly higher in capillary endothelial cells, rather than in venous, arterial, or lymphatic endothelial cells.

Extended Data Fig. 7C, UMAP visualization depicting *Scarb1* (SR-B1) expression patterns across four distinct cardiac endothelial cell (EC) clusters. **D**, Quantitative analysis of SR-B1 expression levels in four EC clusters from the NC $S^{fl/+}; RFP^{AEC}$ control group.

Comment 11: Clarification of the methodology throughout, as suggested by Reviewer #2.

(1) The major point 2 of Reviewer #2 “*Mouse studies methods -a more detailed description of the method used to induce HFpEF in mice should be provided?*”

Response: We sincerely apologize for any inconvenience caused. The detailed methodology for inducing HFpEF is described in the "Heart Failure Mouse Model" section of the revised manuscript Methods (Page 19, lines 13–24).

(2) The major point 4-1 of Reviewer #2 “*More importantly, the methods describing the cell work in panels A-C is nowhere to be found - are these from WT mice, mice with endothelial-specific SR-B1 deficiency, mice with HFpEF or both? What were the concentrations of each intervention (TGF- β , H₂O₂, IL-1 β)?*”

Response: The methodological details corresponding to this section have now been incorporated into the *Supplementary Methods* under the heading "Cell Isolation, Culture, and Treatment" on page 3, lines 19–25 of *Supplementary Information*, as follows:

“To induce endothelial-to-mesenchymal transition (EnMT), mouse cardiac endothelial cells (MCECs) were treated with 10 ng/mL TGF- β 1 (#100-21-10; Peprotech), a combination of 10 ng/mL TGF- β 1 and 100 μ M hydrogen peroxide (H₂O₂) (106058; Beijing Tongguang Fine Chemicals Company), or a combination of 10 ng/mL TGF- β 1 and 1 ng/mL IL-1 β (200-01B-2#; Peprotech).^{6,7} Treatments were administered the following morning and repeated every other day for a total duration of 7 days. Subsequently, RNA levels were assessed using reverse transcription quantitative polymerase chain reaction (RT-qPCR).”

(3) The major point 4-2 of Reviewer #2 “**Fig. 4B**-*Why was different stats used for Colla1 (Kruskal-Wallis one-way ANOVA with Dunn (is this Dunnett's or Dunn's??) multiple comparisons test, and one-way ANOVA followed by Bonferroni post-hoc test for other genes? Surely the data cannot be normally distributed on n=3 (essential for using parametric statistics).*”

Response: We sincerely apologize for the statistical methodological errors in our initial analysis. The revised statistical approach is detailed in the updated figure legends, as follows:

B
Fig.4 B, EnMT-related genes and *Scarb1* (SR-B1) mRNA abundance were measured in MCECs isolated from wild-type (WT) mice following stimulation with different factors *via* RT-qPCR (n=7). Statistical analyses were performed using distinct approaches based on data distribution characteristics: *Col1a1* and *Col3a1* expression data were analyzed using the Kruskal-Wallis test followed by Dunn's multiple comparisons test; *Acta2* (α SMA) expression was evaluated using Brown-Forsythe and Welch ANOVA tests with Dunnett's T3 multiple comparisons; and the remaining genes were assessed using ordinary one-way ANOVA with Bonferroni's multiple comparisons correction.

(4) **The minor point 1 of Reviewer #2** “With the Methods only included as a supplement and not in the main text, the authors should provide some brief insights at least into what was done for each aspect of the Results> For example, the definition of endothelial-specific SR-B1 deletion mice (*S^{AEC}*) should be clearly provided in the main text.”

Response: We have relocated the sections on ethics, animal models, and statistical methods to the revised manuscript on **page 19 (lines 8–29)** and **page 20 (lines 1-16)**. Additionally, we have included brief explanations in each key results section to enhance clarity. For instance, in the Results 2 “*Endothelial-specific SR-B1 deficiency aggravates diastolic dysfunction in HFpEF*”, the definition of endothelial-specific SR-B1 deletion mice (*S^{AEC}*) is provided on **page 5 (line 29)** and **page 6 (lines 1-16)**, as follows:

“To directly investigate the role of endothelial SR-B1 in HFpEF pathogenesis, endothelial-specific SR-B1 knockout mice (*S^{fl/fl}*; *Cdh5-Cre^{ER}*; *S^{AEC}*) were designed based on a

tamoxifen-inducible Cre-loxP system (Fig. 2A). Briefly, *SR-B1^{fllox/+}* (*S^{fl/+}*) mice were constructed by inserting loxP sequences into the flanking introns of exon 2 (158 bp) of the mouse *Scarb1* (SR-B1) gene using CRISPR-Cas9 gene-editing technology. Subsequently, tamoxifen-inducible endothelial-specific cadherin 5 (*Cdh5*)-*Cre^{ER}* mice were crossed with *S^{fl/+}* mice to generate *S^{fl/+}* *Cdh5-Cre^{ER}* mice. Then, *S^{AEC}* and littermate control *S^{fl/fl}* mice were obtained from the cross of *S^{fl/+}* *Cdh5-Cre^{ER}* mice and *S^{fl/+}* mice, which were confirmed by DNA genotyping (*Extended Data Fig. 2A and Supplementary Table 1*). In contrast to a previous report⁸, this loxP-inserting strategy did not disrupt SR-B1 gene expression (*Extended Data Fig. 2B*). Finally, SR-B1 protein abundance and knockout specificity were evaluated, demonstrating that endothelial SR-B1 was specifically deleted, with approximately 70% reduction observed two weeks after tamoxifen injection (*Fig. 2B, Extended Data Fig. 2C and D*).”

Response to the referees' comments

To Reviewer #1

Wu et al report that endothelial cell SR-B1 is downregulated in response to high fat diet/L-NAME, and show that endothelial cell deletion of SR-B1 exacerbates diastolic dysfunction, cardiac fibrosis, and cardiac hypertrophy in this model of pre-clinical cardiometabolic HFpEF. They also report that patient myocardium has significantly decreased gene expression of SR-B1 and increased plasma levels of circulating CXCL10 compared to control patients. The authors associate the EC downregulation in mice with increased expression of CXCL10 through the transcription factor IRF1 and show that CXCL10 cardiac overexpression exacerbates adverse cardiac remodeling. Several limitations listed below preclude the presented data to support the conclusion that CXCL10 levels induced by endothelial SR-B1 induce T cell pathology in HFpEF, as there is no supportive data of the cardiac immune response in deficient or sufficient SR-B1 mice, and the comparisons with the HFrEF model chosen are weak.

Response: We sincerely thank you for your highly valuable suggestions, which have significantly contributed to the improvement of our manuscript. Due to the addition of new data, the layout of the figures in our article has undergone minor adjustments. All textual modifications have been highlighted with a yellow background for easy reference. Below are our point-by-point responses with the original comments in black and our responses in blue.

Major Comments:

Comment 1# (addressed through three sub-points: (1)-(3))

The main limitation is that the model proposed (CXCL10-CXCR3 T cell axis when Endothelial cell SR-B1 is downregulated) is not supported by the data:

Sub-point (1) What are the levels of infiltrated T cells and other immune cells when CXCL10 is overexpressed? Do they change with HFD+ LNAME or is CXCL10 sufficient to attract CXCR3+ T cells to the heart? Are CXCR3+ T cells expanded by HFD+ LNAME Systemically. Figure 7 lacks all this critical information, using a model that is used as a standard in vivo chemotaxis study that may be altered by HFD+LNAME.

Response: Thank you very much for your valuable suggestions. In response to your questions, we will address them in two parts:

1) First, regarding the role of CXCL10 in mediating immune cell chemotaxis to the heart in HFpEF mice or healthy mice, particularly its effect on CXCR3+ T cells. To investigate this, we employed an endothelial-specific AAV1 delivery system to overexpress CXCL10 or the green fluorescent protein zsGreen in cardiac endothelial cells of both HFpEF and healthy mice. Subsequently, we used flow cytometry to analyze the levels of immune cells in the mouse hearts (*Fig. 7P-7R and Extended Data Fig. 10*). In sum, the results demonstrated that, compared to the control group injected with zsGreen, CXCL10 overexpression in HFpEF hearts

significantly promoted the accumulation of CXCR3+ T cells in the heart. However, in healthy mice, CXCL10 overexpression did not significantly increase the number of CXCR3+ T cells. The detailed results are presented on **page 14 (lines 26-29)** and **page 15 (lines 1-15)**, as follows:

“We utilized flow cytometry to assess immune cell infiltration in HFpEF mice at 5 weeks post-AAV injection (*Extended Data Fig. 10A*). Consistent with mice of 10 weeks post-aaav injection, these CXCL10-overexpressing mice exhibited cardiac hypertrophy and diastolic dysfunction (*Extended Data Fig. 10B-E*). The flow cytometry results demonstrated that endothelial CXCL10 overexpression led to a significant increase in the CXCR3+/CD3+ T cell ratio and CXCR3+ T cell counts (*Fig. 7P-7R*), as well as a marked elevation in CD45+ leukocytes (*Extended Data Fig. 10F*). CD3+ T cells and CD11b+ myeloid cells remained unchanged (*Extended Data Fig. 10G and 10H*).

Subsequently, we applied the same strategy to evaluate the effects of CXCL10 on immune cell infiltration and cardiac function in healthy mice (*Extended Data Fig. 10I*). Notably, these mice did not exhibit significant cardiac hypertrophy or diastolic dysfunction (*Extended Data Fig. 10J–10M*). Flow cytometry analysis revealed that, apart from a marked increase in the proportion of CXCR3+/CD3+ T cells, no significant alterations were observed in immune cell subsets (including CD45+, CD11b+, CD3+, or total CXCR3+ cell counts) in AAV1-Cdh5-CXCL10-injected mice compared to AAV1-Cdh5-zsGreen-injected controls (*Extended Data Fig. 10N-10S*).”

Fig. 7P, Representative flow cytometry histograms depicting the percentage of CXCR3+ T cells among CD3+ T cells in mouse cardiac left ventricle, with corresponding quantitative analysis (**Q**, n=6, Welch’s t test). **R**, Flow cytometric quantification of cardiac CXCR3+ T cells (n=6, Welch’s t test). The flow cytometric data was obtained in HFpEF_{ZsGreen} and HFpEF_{CXCL10} mice subjected to 5 weeks of HFD plus L-NAME regimens post-injection of zsGreen or CXCL10-expressing AAV.

Extended Data Fig. 10. The role of CXCL10 in cardiac function and immune cell infiltration in heart was investigated in both HFpEF disease state (A-H) and healthy state (I-S). **A**, Workflow of the experimental design to investigate the role of CXCL10 during HFpEF pathogenesis. Echocardiographic parameters are shown in: **(B)** FS, left ventricular fractional shortening (n=6, Mann-Whitney test); **(C)** LVRI, left ventricular remodeling index (n=6, Student's t-test); **(D)** E/e' ratio between mitral E wave and e' wave (n=6, Welch's t-test). **E**, Ratio of cardiac ventricle weight to tibia length (TL) (n=6; Student's t-test). Flow cytometric quantification of **(F)** CD45+ cells, **(G)** CD11b+ cells, and **(H)** CD3+ T cells in the left ventricle (n=6, Student's t-test). **I**, Workflow of the experimental design to investigate the role of CXCL10 in healthy hearts. Cardiac function was assessed by echocardiography, shown in: **(J)** FS, left ventricle fractional shortening; **(K)** LVRI, left ventricular remodeling index; **(L)** E/e' ratio between mitral E wave and e' wave. **M**, Ratio of cardiac ventricle weight to tibia length (TL). Flow cytometric quantification of **(N)** CD45+ T cells, **(O)** CD11b+ cells, **(P)** CD3+ cells, **(Q)** CXCR3+ cells, and **(R)** CXCR3+/CD3+ ratio in the left ventricle. **(S)** Representative western blot images of cardiac CXCL10 protein expression (right panel) and corresponding quantification (left panel). N=5 **(J-S)**, significant differences were analyzed using Student's t-test.

2) Second, regarding the effects of HFD+LNAME on cardiac and systemic immune cells in the heart and blood. Overall, we found that HFD+LNAME induced infiltration of various immune cell populations in the heart, and systemic expansion of immune cells in blood, including CD45+, CD11b, CD3+, and CXCR3+ T cells. These findings are consistent with those previously reported by Smolgovsky et al. in *JCI*, 2023⁹. The specific findings are presented on **page 11 (lines 18-29)**, as follows:

“Previous studies have demonstrated that T cell infiltration contributes to cardiac hypertrophy and dysfunction, involving peripheral CD4+ T cell expansion⁹. However, whether systemic CXCR3+ T cells are also expanded in HFpEF remains unclear. To address this, we conducted flow cytometry analysis on cardiac tissues and blood samples from HFpEF mice and their controls after 10 weeks of dietary intervention (the gating strategy for immune cells of heart or blood is presented in *Supplementary Fig. 1B and 1C*). As shown in *Extended Data Fig. 7D-7E*, in addition to the increased numbers of CD45+, CD11b+, and CD3+ cells, CXCR3+ T cells were significantly elevated in both cardiac tissues and peripheral blood of HFpEF mice. Notably, while the proportion of CXCR3+ T cells within the total T cell population was markedly increased in cardiac tissues, no such change was observed in peripheral blood (*Extended Data Fig. 7D and 7E*). This discrepancy suggests that the CXCR3-positive T-cells cardiotropism may be regulated by the spatiotemporal expression of endogenous chemokines within the local myocardial microenvironment, potentially including CXCR3 ligands such as CXCL9, CXCL10, and CXCL11.”

Extended Data Fig. 7D, Flow cytometric analysis of immune cell infiltration in cardiac tissues from mice subjected to a 10-week HFD diet plus L-NAME compared with control group (n = 6, Student's t-test). **E**, Absolute immune cell counts by flow cytometry in peripheral blood comparing 10-week HFpEF with controls. (n=6, Student's t-test or Mann-Whitney test).

Sub-point (2) Supplemental Figure 8A is also insufficient to support this axis. Flow cytometric quantitative characterization is needed, and, if the conclusion the authors make is supported, then the data should not be hidden in the supplement.

Response: Thank you for your valuable suggestions. We have incorporated the results from *Supplemental Fig. 8A* into *Fig. 5H* in the revised version. After 20 weeks of HFD plus L-NAME regimens, we performed flow cytometric sorting of immune cells in the hearts of SR-B1-deficient mice. Consistent with the CXCR3+ immunohistochemistry findings (*Fig. 5H*), flow cytometric quantitative analysis revealed that SR-B1 knockout significantly promoted the infiltration of CXCR3+ T cells into the hearts of HFpEF mice, but not CD45+ and CD11b+ immune cell populations (*Fig. 5I-5K*, and *Extended Data Fig. 7C*). The specific results are detailed on **page 11, lines 11–17** of the revised manuscript, as follows:

“To further confirm that SR-B1 deficiency promotes CXCR3+ T cell infiltration in the heart, flow cytometry analysis was performed (the gating strategy is presented in *Supplementary Fig. 1B*). The results demonstrated that SR-B1 deficiency increased the CXCR3+ T cell/CD3e+ T cell ratio (*Fig. 5I and 5J*), and enhanced the infiltration of both CXCR3+ T cells (*Fig. 5K*) and total T cells (*Extended Data Fig. 7C*) in HFpEF cardiac left ventricle, without affecting the populations of CD45+ and CD11b+ cells (*Extended Data Fig. 7C*).”

Fig. 5I, Representative flow cytometry histograms depicting the percentage of CXCR3+ T cells among CD3+ T cells in cardiac left ventricle of $S^{fl/fl}$ and $S^{\Delta EC}$ HFpEF mice, with corresponding quantitative analysis (J, n=6, Student's t test). K, Flow cytometric quantification of cardiac CXCR3+ T cell (n=6, Student's t test).

C Heart from $S^{fl/fl}$ and $S^{\Delta EC}$ HFpEF Mice

Extended Data Fig. 7C, Quantitative flow cytometric analysis of immune cell populations, including CD45+ leukocytes, CD11b+ myeloid cells, and CD3+ T lymphocytes, in the left ventricular tissue of $S^{fl/fl}$ and $S^{\Delta EC}$ HFpEF mice (n=6, Student's t test).

Sub-point (3) Figure 5 ignores cluster 5, which seems to be important based on gene expression for the proposed mechanism. If the rationale for is that T cells mediate diastolic dysfunction in response to HFD+LNAME, as reported by Smolgovsky et al in *JCI*, 2023, and the conclusion is that EC SR-B1 downregulation contributes to this effect, then this work should be cited, and additional experiments and careful analysis of the seq data need to be done.

Response: Thank you very much for your valuable suggestions. During our single-cell data analysis, we also noted this issue. Compared to capillary cluster 0, we believe that the increase in the proportion of cluster 5 cells induced by SR-B1 deficiency may represent a compensatory adaptation in the progression of HFpEF. This hypothesis is supported by the following **four** points:

1) Under normal dietary conditions (corresponding to two NC group), the proportion of cluster 5 remained unchanged, while the proportion of Cluster 0 increased, consistent with the trends observed in the HFpEF state (*Fig. 5D*).

Fig. 5 D, the comparison of the percentage of cardiac capillary EC sub-clusters from left cardiac ventricles of different groups.

2) In the HFpEF state, cluster 0 not cluster 5 accounted for the largest proportion of CMECs (*Fig. 5D*).

3) The signaling pathways activated in Cluster 5 include leukocyte transendothelial migration, tight junction, and focal adhesion (*Extended Data Fig.6F*). If SR-B1 deficiency indeed promoted Cluster 5, it could theoretically lead to increased infiltration of various immune cell populations, including CD11b+ cells. However, our flow cytometry data demonstrated that SR-B1 loss primarily enhanced the infiltration of CXCR3+ T cells into the myocardium (*Fig. 5I and Extended Data Fig. 7C*).

Extended Data Fig.6F, Heatmap showing KEGG signaling pathways enriched in sub-clusters of cardiac microvascular ECs.

Fig. 5I, Representative flow cytometry histograms depicting the percentage of CXCR3+ T cells among CD3+ T cells in cardiac left ventricle of $S^{fl/fl}$ and $S^{\Delta EC}$ HFpEF mice, with corresponding quantitative analysis (J, n=6, Student's t test). K, Flow cytometric quantification of cardiac CXCR3+ T cell (n=6, Student's t test).

C Heart from $S^{fl/fl}$ and $S^{\Delta EC}$ HFpEF Mice

Extended Data Fig. 7C, Quantitative flow cytometric analysis of immune cell populations, including CD45+ leukocytes, CD11b+ myeloid cells, and CD3+ T lymphocytes, in the left ventricular tissue of $S^{fl/fl}$ and S^{dEC} HFpEF mice (n=6, Student's t test).

4) Through *ex vivo* and *in vivo* experiments, we confirmed that the expression levels of marker genes associated with Cluster 0 were significantly elevated, particularly CXCL10, which exhibited the highest fold change. AAV1-Cdh5-mediated overexpression of CXCL10 drove infiltration of CXCR3+ T cells into myocardium (**Fig. 7P-7R**), which aligns well with the flow cytometry results from SR-B1-deficient mice (**Fig. 5I and Extended Data Fig. 7F**). Therefore, we conclude that Cluster 0 is the primary CMEC subpopulation responsible for cardiac dysfunction induced by SR-B1 deficiency.

Fig. 7P, Representative flow cytometry histograms depicting the percentage of CXCR3+ T cells among CD3+ T cells in mouse cardiac left ventricle, with corresponding quantitative analysis (**Q**, n=6, Welch's t test). **R**, Flow cytometric quantification of cardiac CXCR3+ T cells (n=6, Welch's t test). The flow cytometric data was obtained in HFpEFZsGreen and HFpEFCXCL10 mice subjected to 5 weeks of HFD plus L-NAME regimens post-injection of zsGreen or CXCL10-expressing AAV.

Indeed, these data cannot completely negate your suggestion. Regarding your suggestion, we hope to conduct some more systematic research in future studies.

Additionally, in our revised manuscript, we have referenced the study by Smolgovsky et al. (*JCI*, 2023)⁹. Our data demonstrate that the expansion of immune cells in the peripheral blood and the infiltration of diverse immune cell populations into the hearts of HFpEF mice are entirely consistent with the findings reported in their study. In addition to the reported abnormal expansion of peripheral immune T cells⁹, the endogenous secretion of chemokines in hearts—triggered by microvascular endothelial inflammatory activation—also plays a critical role in the recruitment of activated T cells to the cardiac tissue.

Comment 2# (addressed through two sub-points: (1)-(2))

Sub-point (1) What is the time course of downregulation of EC SR-B1 and its effects on remodeling? If the proposed chemokine axis is the main one driving this, this timely response is important.

Response: We sincerely appreciate your valuable concern. Our investigations into cardiac function and *Scarb1* expression at different stages of HFpEF revealed a progressive decline in *Scarb1* mRNA levels, which correlated with the worsening of HFpEF-associated cardiac hypertrophy and dysfunction.

Fig. 1F, *Scarb1* mRNA abundance in MCECs at various time points following administration of a high-fat diet (HFD) combined with L-NAME.

Sub-point (2) Moreover, AAV overexpression does not target ECs exclusively, so the effects may be independent on the EC signals.

Response: In our AAV1 delivery system, the CXCL10 or SR-B1 expression is driven by the endothelial cell-specific promoter *Cdh5*. This expression vector was provided by Professor Yu Huang from the School of Biomedical Sciences at the Chinese University of Hong Kong, and has been previously utilized for specific *in-vivo* overexpression in mouse aortic endothelial cells (*Huang et al., Circulation Research, 2021*)¹⁰. In the revised manuscript, we confirmed through immunofluorescence analysis of heart sections from mice injected with AAV1 expressing *zsGreen* that *zsGreen* was predominantly expressed in cardiac CD31+ endothelial cells (*Extended Data Fig. 4C*). Moreover, a recent study reported that the high hepatic tropism of AAV vectors leads to low-level expression leakage in the liver even when using the cardiac-specific promoter *Tnnt2* (*Yang et al., Circulation, 2024*)¹¹. Whether AAV1-*Cdh5*-mediated gene delivery causes leakage in other organs has not been reported. This potential leakage issue may require further systematic investigation. However, at least in the heart, our current delivery system does achieve specific targeting of endothelial cells.

C

Extended Data Fig. 4C, Representative fluorescence microscopy images showing ZsGreen signals mainly localized in CD31-positive cells in cardiac ventricular sections of wild-type (WT) mice following AAV1-mediated gene delivery. Scale bar, 50 μm .

Comment 3#

The authors draw comparisons between preclinical HFpEF and HFrfEF by comparing the two hit model to TAC (respectively). However, the time points chosen for analysis are not comparable – the authors measure changes in cardiac pathology in 20 weeks of high fat diet/L-NAME and compare to no changes in cardiac pathology after 2 weeks of TAC. Particularly since the earlier 10 week high fat diet/L-NAME comparisons don't result in statistically significant changes with SR-B1 deletion, a longer time point post-TAC should be used to convincingly demonstrate this phenotype is HFpEF-specific.

Response: We sincerely appreciate your valuable suggestions. From a temporal perspective, it is indeed less rigorous to directly compare the two time points. We attempted to extend the postoperative monitoring period in our model, but this approach proved unsuitable. Specifically,

we conducted a new batch of TAC surgeries under standard aortic ligation pressure (using a blunted 27-gauge needle)^{2,3} on SR-B1 endothelial-specific knockout mice, increasing the sample size to n=8–9. Consistent with our previous findings, both TAC groups developed severe dilated HFrEF two weeks post-surgery, as evidenced by a significant reduction in fractional shortening (FS; mean FS <20%, *Extended Data Fig. 5C*) and markedly increased left ventricular internal diastolic diameter (LVIDd, *Extended Data Fig. 5F*). This acute HFrEF phenotype may be strain-specific, as our SR-B1 floxed mice are based on the C57BL/6N strain. According to previous studies, the **Rong Tian group**⁴ has demonstrated that the C57BL/6N strain typically exhibits a rapid pathological progression of acute heart failure compared to the C57BL/6J strain, with a postoperative mortality rate reaching approximately 60% by seven weeks. Therefore, we believe that extending the observation period is not sustainable for our study due to high mortality. Based on the current data, we concur with the suggestion that a direct comparison between preclinical HFpEF and HFrEF is not appropriate. Consequently, we have revised the relevant descriptions in the manuscript and emphasized on the relationship between SR-B1 and acute heart failure. Furthermore, we have added this limitation in the "Limitations" section of our manuscript on **page 18 (lines 8-14)**, as follows:

"The present study has several limitations that warrant discussion. Firstly, acute heart failure and chronic heart failure represent distinct pathological conditions. In our study, the role of SR-B1 was exclusively investigated in the context of acute heart failure. Previous studies have indicated that CXCR3+ T cell infiltration exacerbates cardiac systolic dysfunction⁵, and SR-B1 deficiency facilitates the recruitment of CXCR3+ T cells into the heart. Therefore, SR-B1 may also play a potential role in HFrEF. As such, an ideal preclinical model of chronic HFrEF should be utilized in future research to further elucidate the role of SR-B1 in HFrEF."

Extended Data Fig. 5C, FS, left ventricular fractional shortening.

Extended Data Fig. 5F, LVIDd, left ventricular internal diastolic diameter.

Comment 4# (addressed through two sub-points: (1)-(2))

Sub-point (1) In Figure 4, there is no flow cytometry gating strategy to support the Endo-MT data.

Response: The gating strategy for Fig. 4G was displayed in Supplementary Fig. 1A.

A Gating strategy for the EnMT data

Supplementary Fig. 1A Gating strategy for flow cytometry of EnMT. Red fluorescence protein (RFP) from cardiac endothelial ell was detected with PE channel (y axis), and endothelial marker CD31 detected with FITC channel (x axis).

Sub-point (2) Additionally, the decrease of SR-B1 in response to TGF-B1 could be a response to treatment or an effect, not necessarily a driver for Endo-MT. Additional mechanistic experiments are required to support this statement related to Endo-MT.

Response: Thank you for your valuable advice. We have utilized the gold standard *in vivo* method for studying Endo-MT—an endothelial fluorescence reporter HFpEF mouse model. Both flow cytometry and immunofluorescence results have consistently demonstrated that the presence or absence of endothelial SR-B1 does not induce Endo-MT in the hearts of HFpEF mice (Fig. 4F and 4G).

Fig. 4F, Fluorescent images after immunofluorescence staining of α SMA in the cardiac ventricle of $S^{fl/+}/R^{\Delta EC}$ mice fed HFD plus L-NAME. Scale bar, 100 μ m **G**, Single cells from lineage-tracing mice of different groups were FACS sorted according to RFP and CD31 expression. Top panel: negative controls from a heart of $RFP^{\Delta EC}$ mice not injected by tamoxifen (left panel); CD31-positive controls from a heart of $RFP^{\Delta EC}$ mice not injected by tamoxifen (middle panel); RFP-positive controls from a heart of $RFP^{\Delta EC}$ mice 2 weeks after tamoxifen injection (right panel). Bottom panel, double-channel signals were presented from isolated left ventricular cells in mice from different four groups. RFP signal was detected with PE channel (y axis), while CD31 protein expression was measured through FITC channel (x axis).

Comment 5#

In figure 8B, there is no difference in patients with HFpEF expression of SR-B1, which seems to not support the conclusion.

Response: Thank you for your valuable suggestion. We collected additional samples, including one control heart tissue and five cardiomyopathy-related HFpEF heart tissues, and newly performed RT-qPCR analysis. The results demonstrated that the ventricles of HFpEF patients exhibited significantly higher levels of *BNP* and *COL3A1*, accompanied by marked downregulation of the SR-B1 gene ($p < 0.0001$) and significant increases in the abundance of *IRF1*, *CXCL10*, and *CXCR3* (Fig. 8A–C), compared to the healthy control group.

Fig. 8A, *BNP* and *COL3A1* mRNA abundances were assessed in healthy and HFpEF individuals by RT-qPCR (n=5,12; Mann-Whitney test). **B**, *SCARB1* (SR-B1) mRNA abundance relative to an endothelial marker gene was evaluated in healthy and HFpEF individuals (n=5,12; Student's t test). **C**, *IRF1*, *CXCL10*, and *CXCR3* mRNA abundances in healthy and HFpEF individuals (n=5,12; Welch's or Student's t test).

Comment 6#

The authors use histology to demonstrate SR-B1 expression is largely in capillary endothelial cells. This would be strengthened using their single cell data set in which they subset cells as capillary/venous/arterial/lymphatic endothelial cells. Can the authors report SR-B1 expression in each of these clusters to provide quantitative data to support their histological findings?

Response: We analyzed the expression of SR-B1 in the NC group ($S^{fl/+}; RFP^{AEC}$) based on our single-cell data, as shown in *Extended Data Fig. 7C and 7D*. The expression of *Scarb1* is significantly higher in capillary endothelial cells, rather than in venous, arterial, or lymphatic endothelial cells.

Extended Data Fig. 7C, UMAP visualization depicting *Scarb1* (SR-B1) expression patterns across four cardiac endothelial cell (EC) clusters. **D**, Quantitative analysis of SR-B1 expression levels in four EC clusters from the NC $S^{fl/+} RFP^{AEC}$ control group.

Minor Comments:

Comment 1# (addressed through three sub-points: (1)-(3))

Figure legends do not have all the information necessary for interpretation of data.

Sub-point (1) Figure 1F legend does not specify if these data are from the two hit model, from humans v mice....

Response: We sincerely appreciate your suggestion. In response, we have provided more detailed information in the legend for the corresponding figure (now *Fig. 1E*, previously *Fig. 1F*), as follows:

“Relative SR-B1 protein abundance in MCECs isolated from 20-week-old two-hit HFpEF mice and control mice was measured by immunoblotting (upper panel). Statistical analysis was performed using Welch’s t-test (lower panel; n = 3 independent experiments using pooled samples, with each data point representing 2–3 mouse hearts).”

Sub-point (2) Scale bars for IF images are really hard to see, and the units of the scale bar should be specified in figure legends.

Response: We sincerely apologize for any inconvenience caused by our oversight. We have uniformly increased the size of all scale bars and their corresponding fonts in the images, and we have added annotations in the respective figure legends. For example, as shown in *Fig. 1A* and *Fig. 1B*:

Fig.1 legends A, Immunofluorescence staining of SR-B1 in mouse cardiac left ventricle co-stained with fluorescence-conjugated isolectin B4 (IsoB4) to label endothelial cells and DAPI to label nuclei. Scale bar, 25 μm. **B**, Double-staining of SR-B1 and CD31 in cardiac left ventricle of a healthy human. Scale bar, 100 μm.

Sub-point (3) Supplementary Figure 1E does not specify how long after TAC these samples were obtained.

Response: We have updated the legend for *Fig. 1F* (previously *Fig. 1E*) as follows: “SR-B1 abundance was measured in mouse cardiac ECs from sham-operated or TAC-induced HFpEF mice (mean FS=18%) at 6 weeks post-surgery via RT-qPCR (n=4).”

Comment 2# (addressed through two sub-points: (1)-(2))

Sub-point (1) The authors make conclusions regarding groups, yet the statistics between those groups are often not shown. For instance, page 7 of results concludes "...Cxcl10 was the differentially expressed gene with highest fold change compared to Cxcl1 and Cxcl2 (Fig 5G)." yet the statistics comparing expression of each of these genes in Figure 5G are not shown.

Response: We sincerely apologize for the inappropriate description in our original text. Our intention was to highlight that *Cxcl10* exhibited the highest fold change among the measured chemokines. To provide a more scientifically rigorous comparison, we analyzed the fold change differences between *Cxcl10* and *Cxcl1* or *Cxcl2* using a two-way ANOVA followed by Bonferroni post-hoc test. As shown in **Fig. 5F**, *Cxcl10* was identified as the differentially expressed gene with the highest fold change compared to *Cxcl1* ($p = 0.0068$) and *Cxcl2* ($p = 0.0030$).

Fig. 5F *Scarb1* (SR-B1), *Cxcl1*, *Cxcl2*, *Cxcl10*, and *Cxcr3* mRNA abundance were measured in left ventricle of $S^{fl/fl}$ and $S^{\Delta EC}$ HFpEF mice (n=5, the fold change differences between *Cxcl10* and *Cxcl1* or *Cxcl2* were analyzed using ANOVA followed by Bonferroni post-hoc test, while the remaining comparisons were assessed using Student's t-test).

Sub-point (2) Additionally, statistics between the red bars (SR-B1 endothelial cell KO groups) are not indicated in Figure 3.

Response: We have refined the comparative analysis between the red bars (SR-B1 endothelial cell KO groups), as presented in **Fig. 2** and **Fig. 3**. For instance, as shown in **Fig. 3A**, compared to $S^{\Delta EC}$ NC mice, $S^{\Delta EC}$ HFpEF mice exhibited a significant increase in heart weight ($p < 0.05$). Another point to clarify is that if significant differences are observed between any of the NC group mice and any of the HFpEF group mice, we will present the results uniformly using a

format similar to that in *Fig. 3A*, rather than employing a pairwise comparison format. This approach enhances clarity and conciseness in the presentation of the data.

Fig.3A Ratio between cardiac ventricle weight and tibia length (TL) in $S^{fl/fl}$ and $S^{\Delta EC}$ mice sacrificed after 20 weeks of normal diet or HFD plus L -NAME (n=5, 5, 6, 6).

Comment 3# (addressed through two sub-points: (1)-(2))

Sub-point (1) Figure 4G: please provide gating strategy in supplemental information for these plots.

Response: The gating strategy for *Fig. 4G* was displayed in *Supplementary Fig. 1A*.

A Gating strategy for the EnMT data

Supplementary Fig. 1A Gating strategy for flow cytometry of EnMT. Red fluorescence protein (RFP) from cardiac endothelial ell was detected with PE channel (y axis), and endothelial marker CD31 detected with FITC channel (x axis).

Sub-point (2) Additionally, consider changing annotation of the plots at the top, as it is difficult to understand what “blank”, “only CD31”, and “only RFP” relates to.

Response: We sincerely appreciate your valuable suggestions. In accordance with the figure legends, we have revised the annotations in **Fig. 4G** as you indicated.

Fig. 4G, Single cells from lineage-tracing mice of different groups were FACS sorted according to RFP and CD31 expression. Top panel: negative controls from a heart of *RFP^{ΔEC}* mice not injected by tamoxifen (left panel); CD31-positive controls from a heart of *RFP^{ΔEC}* mice not injected by tamoxifen (middle panel); RFP-positive controls from a heart of *RFP^{ΔEC}* mice 2 weeks after tamoxifen injection (right panel). Bottom panel, double-channel signals were presented from isolated left ventricular cells in mice from different four groups. RFP signal was detected with PE channel (y axis), while CD31 protein expression was measured through FITC channel (x axis).

Comment 4#

Figure 6I: include DAPI only panels since the merge is shown

Response: We sincerely appreciate your suggestion. We have added DAPI panels to *Fig. 6I*.

Fig. 6I Immunofluorescent staining of IRF1 co-stained with DAPI to compare nucleus signals of IRF1 in HCMECs when SR-B1 presence or absence. Scale bar, 100 μm.

Comment 5#

Supplementary figure 2A, SR-B1 western blots – the top panel (SR-B1 expression) appears to be two blots with two different exposures next to each other? Why does the exposure seem to differ so strikingly between the heart and liver samples? If this is two blots stitched together, a line must be added to indicate this, as well as explanation in the figure legend. If it is one blot and the liver samples happen to have more background, extend the height of the blots so that this is more evident.

Response: The SR-B1 blot in *Supplementary Fig. 2A* (which corresponds to Extended Data *Fig. 2B* in our current revised version) is derived from a single blot. To clarify this, we have provided an overexposed image in Extended Data *Fig. 2B*. Based on literature reports and our experimental results, we consider that the pronounced intensity difference observed in the Western blot is primarily due to the fact that SR-B1 is predominantly expressed in endothelial cells rather than parenchymal cells (cardiomyocytes) in cardiac tissue. In contrast, in liver, SR-B1 is highly expressed in hepatocytes, where it plays a critical role in reverse cholesterol transport¹².

Extended Data Fig. 2B, Immunoblotting of SR-B1 in the heart and liver of *S^{fl/fl}* mice and their wild-type (WT) littermates (left panel); densitometric analysis of SR-B1 protein abundance (n=3, right panel). Significant difference was evaluated *via* Student’s t test.

Comment 6#

Supplemental Figure 8C: specify what the origin of this single cell data set is. Is this a resting vs diseased heart? Murine or human? Please indicate in the figure legend, in results section, and in methods section.

Response: Thank you for your valuable suggestions. *Supplemental Fig. 8C* has now been relocated to *Extended Data Fig. 7B*. We have incorporated the relevant descriptions in the figure legend, results section, and methods section as follows:

In the Figure Legend:

“The bubble map illustrates CXCR3 expression in a mixed cardiac sample, including healthy control mice and HFpEF mice, derived from a wild-type single-cell transcriptome database (GSE236586).”

In the Results Section:

“Our histoimmunofluorescence results confirmed that CXCR3 is predominantly expressed in cardiac T cells (*Extended Data Fig. 7A*), consistent with a single-cell dataset derived from the hybridization of control and diseased murine hearts (*Extended Data Fig. 7B*).”

In the Methods Section:

“ The detailed analytical steps are presented in the *Supplementary Methods* section under “*The expression bubble Map of Cxcr3 in the Heart.*”

Comment 7#

Page 4 of results section “...HFpEF mice exhibited a tendency of more cardiac hypertrophy and worse diastolic dysfunction than...” – this text is in reference to the data in which the changes were not statistically significant. This conclusion in the text needs to be softened and indicate these changes are not statistically significant.

Response: We appreciate your suggestion and have adopted a more objective description as follows:

“Notably, *S^{ΔEC}* HFpEF mice exhibited a consistent trend toward more severe cardiac hypertrophy (as evidenced by LV mass and left ventricular remodeling index (LVRI) parameters) and worse diastolic dysfunction (indicated by myocardial performance index (MPI) and E/e’ ratio) compared

to $S^{fl/fl}$ HFpEF mice, but these differences did not reach statistical significance.”

To Reviewer #2

Major remarks:

The authors demonstrate that mice with endothelial-specific SR-B1 deficiency show exacerbated HFpEF in mice (on E/e', cardiac remodelling and fatigue in particular). Large, elegant body of work with predominantly nice, clean images but there are examples where more evidence could be provided in support of their statements as outlined below which would strengthen the manuscript noticeably.

Response: We extend our heartfelt gratitude for your insightful suggestions, which have greatly enhanced the quality of our manuscript. With the inclusion of new data, the figure arrangement in our article has been slightly reorganized. All textual revisions in our manuscript have been marked with a yellow background for clarity. Below are our point-by-point responses with the original comments in black and our responses in blue.

Specific comments:

Comment 1. The authors claim Fig 1 shows SR-B1 is predominantly present in cardiac microvascular endothelial cells and significantly decreased in HFpEF heart.

Sub-point (1) Whilst the Western blots and qPCR data convincingly show SRB1 is higher in MCECs than non-MCECs, I am less convinced by panels A and B and the merged image of SRB1 and IsoB4.

Response: Thank you very much for your suggestions. In *Fig. 1A*, co-localization regions of SR-B1 and IsoB4 were marked with white straight arrows. In *Fig. 1B*, we replaced the CD31 antibody (ab28364, Abcam) and performed co-localization imaging of SR-B1 and CD31 on frozen human heart sections using a new method based on the Treble-Fluorescence immunohistochemical mouse/rabbit kit (RS0035, Immunoway), which significantly enhanced the quality of the initial immunofluorescence data. Similarly, co-localization regions of SR-B1 and CD31 were also annotated with white straight arrows. Overall, the immunofluorescence staining results in both mouse and human hearts were consistent, demonstrating that SR-B1 is predominantly expressed in microvascular endothelial cells rather than endothelial cells of large vessels.

Fig. 1A, Immunofluorescence staining of SR-B1 in mouse cardiac left ventricle co-stained with fluorescence-conjugated isolectin B4 (IsoB4) to label endothelial cells and DAPI to label nuclei. Scale bar, 25 μm .

Fig. 1B, Double-staining of SR-B1 and CD31 in cardiac left ventricle of a healthy human. Scale bar, 100 μm .

Moreover, we analyzed the expression of SR-B1 in the NC group ($S^{fl/+}; RFP^{AEC}$) based on our single-cell data, as shown in *Extended Data Fig. 7C and 7D*. The expression of *Scarb1* is significantly higher in capillary endothelial cells, rather than in venous, arterial, or lymphatic endothelial cells. Taken together, our findings indicate that SR-B1 is primarily localized in cardiac microvascular endothelial cells.

Extended data fig. 7 C, UMAP visualization depicting *Scarb1* (SR-B1) expression patterns across four distinct cardiac endothelial cell (EC) clusters. **D**, Quantitative analysis of SR-B1 expression levels in four EC clusters from the NC $S^{fl/+} RFP^{ΔEC}$ control group.

Sub-point (2) Fig 1F shows n=3 only claiming that HFpEF mice have reduced SRB1 but why not a more robust number? Particularly when the full data in the supplement clearly has more n than just 3. Where are the full blots? and the Mw markers? How do we know the bands detect what the authors claim they do?

Response: In the current revised version, Fig. 1F corresponds to Fig. 1E. For Fig. 1E (n = 3), we would like to clarify that each sample (n = 1) represents a pooled mixture of cardiac endothelial cells isolated from at least 2 mice, as indicated in the corresponding figure legend. The isolation method was consistent with that employed for single-cell sequencing of pooled mixed samples.

Moreover, in this revised manuscript, we also investigated the change of *Scarb1* (SR-B1) transcriptional level in different stages of HFpEF progression, validating significant downregulation of *Scarb1* mRNA level in cardiac endothelial cells in HFpEF mice than control mice (**Fig. 1F**).

Fig. 1F, *Scarb1* mRNA abundance in mouse cardiac ECs at various time points following administration of a high-fat diet (HFD) combined with L-NAME drink (n=5-6 / group, Welch's or Student's t-test).

The source data corresponding to **Fig. 1E** are presented below, demonstrating: (1) the SR-B1 protein band migrating between 70-100 kDa, and (2) the GAPDH loading control band appearing at 35-40 kDa.

Figure Legend: Original western blot images corresponding to Fig. 4E.

Fig.1 E, Relative SR-B1 protein abundance in MCECs isolated from 20 weeks of two-hit HFpEF mice and control mice was measured by immunoblotting (upper panel) and was statistically analyzed *via* Welch's t test (lower panel, n = 3 independent experiments using pooled samples, with each data point representing 2-3 mouse hearts).

Comments 2

Sub-point (1) Mouse studies methods - a more detailed description of the method used to induce HFpEF in mice should be provided?

Response: We sincerely apologize for any inconvenience caused. The detailed methodology for inducing HFpEF is described in the "Heart Failure Mouse Model" section of the revised manuscript Methods (Page 19, lines 13–24).

Sub-point (2) What was the specifications of the diet (was fat content by weight or by caloric intake, what was the other components of the diet eg starch, sucrose, protein, vitamins and minerals, and how did these constituents compare to the chow diet - there is considerable range in what's considered normal chow in mice) and was L-NAME used (and what dose) to increase BP or was a different model used?

Response: In the HFpEF model, a rodent diet with 60% kcal from fat (60% HFD; D12492, Research Diets) was used to induce obesity, while a rodent diet with 10% kcal from fat (D12450J, Research Diets) served as the standard chow diet. In this study, elevated blood pressure in mice was consistently induced by administering L-NAME at a concentration of 0.5 mg/mL in drinking water (pH = 7.4). The specific compositions of the 60% HFD and standard chow diet are as follows:

Formula		
Product #D12492	gm%	kcal%
Protein	26.2	20
Carbohydrate	26.3	20
Fat	34.9	60
Total		100
	kcal/gm	5.24
Ingredient	gm	kcal
Casein, 30 Mesh	200	800
L-Cystine	3	12
Corn Starch	0	0
Maltodextrin 10	125	500
Sucrose	68.8	275.2
Cellulose, BW200	50	0
Soybean Oil	25	225
Lard*	245	2205
Mineral Mix S10026	10	0
DiCalcium Phosphate	13	0
Calcium Carbonate	5.5	0
Potassium Citrate, 1 H2O	16.5	0
Vitamin Mix V10001	10	40
Choline Bitartrate	2	0
FD&C Blue Dye #1	0.05	0
Total	773.85	4057

Formula		
Product #D12450J	gm%	kcal%
Protein	19.2	20
Carbohydrate	67.3	70
Fat	4.3	10
Total		100
	kcal/gm	3.85
Ingredient	gm	kcal
Casein, 30 Mesh	200	800
L-Cystine	3	12
Corn Starch	506.2	2024.8
Maltodextrin 10	125	500
Sucrose	68.8	275.2
Cellulose, BW200	50	0
Soybean Oil	25	225
Lard*	20	180
Mineral Mix S10026	10	0
DiCalcium Phosphate	13	0
Calcium Carbonate	5.5	0
Potassium Citrate, 1 H2O	16.5	0
Vitamin Mix V10001	10	40
Choline Bitartrate	2	0
FD&C Yellow Dye #5	0.04	0
FD&C Blue Dye #1	0.01	0
Total	1055.05	4057

Figure legend: 60% HFD formula (Left) and Chow diet Formula(right).

Sub-point (3) It is insufficient just to say "HFpEF diet" and to refer to this in a figure legend.

Response: Thank you for your suggestions. All references to the HFpEF diet in the manuscript and figure legends have been revised to "HFD plus L-NAME regimens".

Sub-point (4) What was the sex of the mice (both males and females should be included). What age wear the mice studied at? This is important not just for conventional scientific standards of rigor in this era, but also because the clinical data they show to confirm their data is a mix of males and females (and HFpEF is the HF phenotype more frequently seen in women) and the patients are middle-aged and older, not young adults as these mice likely are.

Response: We sincerely appreciate your valuable suggestions. All HFpEF mice used in our study were male, and tissue collection and cardiac function assessments were performed at 15–30 weeks of age. This model has been widely adopted because it successfully recapitulates many clinical and pathological features of HFpEF patients by incorporating common comorbidities such as obesity and hypertension (*Schiattarella et al., Nature, 2019; Cochran et al., Circulation, 2023*)^{13, 14}. These features include systemic inflammation, cardiac fibrosis, myocardial hypertrophy, and diastolic dysfunction.

We acknowledge its limitations, as you pointed out. First, it fails to induce diastolic dysfunction in female mice, even after ovariectomy¹⁵. Second, while HFpEF primarily affects elderly patients, the model utilizes adult or middle-aged mice. Nevertheless, the core pathological feature we focused on—systemic inflammation—was effectively reproduced in our model. Furthermore, our clinical data strongly support the findings from this preclinical HFpEF model.

Therefore, these limitations are unlikely to significantly impact the conclusions of our current study.

Comment 3. Thank you for showing individual data points but would be easier to visualise if these were eg black to distinguish them from the mean and error bars. Please remove all p values where $P > 0.05$.

Response: Thank you for your valuable suggestions. In response to your comments, we have modified all bar graphs by changing the individual data points to black to clearly distinguish them from the mean and error bars. Additionally, we have removed nearly all $P > 0.05$ values, retaining only a few values with $P < 0.10$ to highlight potential trends between groups.

Comment 4

Sub-point (1) Figure 4. "Cardiac endothelial cells cannot transdifferentiated into myofibroblast in HFpEF". Clearly the grammar in the title for this figure legend requires correcting (Minor point).

Response: Thank you for your valuable suggestions. Regarding the minor grammatical point, the sentence has been revised as follows:

" Cardiac endothelial cells cannot transdifferentiate into myofibroblast in HFpEF".

Sub-point (2) More importantly, the methods describing the cell work in panels A-C is nowhere to be found - are these from WT mice, mice with endothelial-specific SR-B1 deficiency, mice with HFpEF or both? What were the concentrations of each intervention (TGF- β , H₂O₂, IL-1 β)?

Response: We sincerely appreciate your insightful suggestions. The methodological details corresponding to this section have now been incorporated into the *Supplementary Methods* under the heading "Cell Isolation, Culture, and Treatment" on page 3, lines 19–25 of *Supplementary Information*, as follows:

“To induce endothelial-to-mesenchymal transition (EnMT), mouse cardiac endothelial cells (MCECs) were treated with 10 ng/mL TGF- β 1 (#100-21-10; Peprotech), a combination of 10 ng/mL TGF- β 1 and 100 μ M hydrogen peroxide (H₂O₂) (106058; Beijing Tongguang Fine Chemicals Company), or a combination of 10 ng/mL TGF- β 1 and 1 ng/mL IL-1 β (200-01B-2#; Peprotech).^{6,7} Treatments were administered the following morning and repeated every other day for a total duration of 7 days. Subsequently, RNA levels were assessed using reverse transcription quantitative polymerase chain reaction (RT-qPCR).”

Sub-point (3) Why only n=3? Why was different stats used for *Col1a1* (Kruskal-Wallis one-way ANOVA with Dunn (is this Dunnett's or Dunn's??) multiple comparisons test, and one-way ANOVA followed by Bonferroni post-hoc test for other genes? Surely the data cannot be normally distributed on n=3 (essential for using parametric statistics).

Response: In **Figure 4B**, we have added the sample size (n = 7). As illustrated in the updated figure (**Fig. 4B**), the results remain consistent with our earlier findings, confirming that cardiac coronary endothelial cells can undergo transdifferentiation into myofibroblasts in vitro, and that SR-BI expression is downregulated during this phenotypic transition. We sincerely apologize for the statistical methodological errors in our initial analysis. The revised statistical approach is detailed in the updated figure legends, as follows:

Fig.4 B, EnMT-related genes and *Scarb1* (SR-B1) mRNA abundance were measured in MCECs isolated from wild-type (WT) mice following stimulation with different factors *via* RT-qPCR (n=7). Statistical analyses were performed using distinct approaches based on data distribution characteristics: *Col1a1* and *Col3a1* expression data were analyzed using the Kruskal-Wallis test followed by Dunn's multiple comparisons test; *Acta2* (α SMA) expression was evaluated using Brown-Forsythe and Welch ANOVA tests with Dunnett's T3 multiple comparisons; and the remaining genes were assessed using ordinary one-way ANOVA with Bonferroni's multiple comparisons correction.

Comment 5. I am concerned by the apparent lack of cardiomyocytes in their non-CMEC scRNA-Seq data set (myocytes are too large for sorting). Most groups now either use snRNASeq, or isolate cardiomyocyte nuclei prior to scRNASeq. This must be addressed.

Response: We sincerely apologize for any lack of clarity in our initial description. The single-cell RNA sequencing (scRNA-seq) in our study was performed using mice with an *RFP^{AEC}* background, in which endothelial cells express red fluorescent protein (RFP) following tamoxifen injection. As shown in **Figure 5A**, sequencing was conducted exclusively on RFP-positive cells, which explains the absence of cardiomyocytes in our dataset.

We fully acknowledge the important concern regarding the technical considerations of cardiomyocyte single-cell sequencing. To clarify, our tissue dissociation method involved mechanical mincing followed by enzymatic digestion at 37°C (as detailed in the **Supplementary Methods** section). This approach does not preserve viable cardiomyocytes, whereas the isolation of living cardiomyocytes typically requires the Langendorff retrograde perfusion technique. However, while the Langendorff method is effective for obtaining viable cardiomyocytes, it can compromise the efficiency of non-cardiomyocyte isolation.

The two objectives of performing single-cell sequencing on mice with RFP-labeled endothelial cells: first, to determine whether cardiac endothelial cells undergo phenotypic transitions during the progression of HFpEF; and second, because tamoxifen-induced endothelial SR-B1 deletion does not occur uniformly across all cells, further sorting of RFP-positive endothelial cells by flow cytometry allows for a more accurate investigation of the underlying molecular mechanisms.

Fig.5 A, A schematic diagram of single-cell transcriptomic sequencing experiment.

Comment 6. The authors have explored the molecular mechanisms underlying the impact of SR-B1 deficiency in vivo, and of SR-B1-mediated CXCL10 regulation in vitro. However, they do not investigate whether the phenotype can be rescued by replacing the SR-B1 in CMECs (could be achieved by AAV-mediated approach). They have the capability to do this, as they do show that the HFpEF phenotype is exacerbated in AAV1-CXCL10-injected HFpEF mice. Showing that they can rescue HFpEF with endothelial SR-B1 delivery would really hit the nail on the head when it comes to showing that cardiac microvascular endothelial activation contributes to HFpEF pathogenesis.

Response: In our study, we restored SR-B1 expression in endothelial SR-B1-deficient mice (S^{AEC}) by delivering SR-B1 mRNA via an AAV1 carrying the Cdh5 promoter-initiated expressing vector, as illustrated in *Fig. 3G*. Echocardiographic analysis revealed that SR-B1 reintroduction effectively rescued cardiac hypertrophic remodeling and diastolic dysfunction induced by endothelial SR-B1 deficiency (*Fig. 3H–M*). Treadmill endurance tests further demonstrated that SR-B1 supplementation significantly improved the impaired cardiac reserve caused by SR-B1 deficiency, leading to a notable increase in running distance (*Fig. 3N*). Moreover, the ratios of heart-to-tibia length and lung wet weight to tibia length (*Fig. 3O and 3P*) confirmed that SR-B1 reconstitution effectively prevented further deterioration of cardiac function and lung edema. As shown in *Fig. 3Q*, SR-B1 restoration also substantially attenuated the elevation of pathological markers in the heart.

In summary, these findings collectively demonstrate that AAV1-mediated SR-B1 restoration effectively rescues the pathological phenotypes associated with endothelial SR-B1 deficiency. The findings related to this section are presented in the revised manuscript on **page 7 (lines 25-29)** and **page 8 (lines 1-5)**.

Fig. 3 (G-Q) Endothelial SR-B1 gain-of-function, achieved through AAV1 tail vein injection in $S^{\Delta EC}$ and $S^{fl/fl}$ mice, was evaluated using echocardiography and pathological analysis (n=6; statistical analyses were performed using one-way ANOVA with Bonferroni's post-hoc test or Kruskal-Wallis test with Dunn's post-hoc test). **G**, Schematic illustration of the experimental design. **H**, FS, left ventricular fractional shortening. **I**, LVPWd, left ventricular posterior wall diastolic thickness. **J**, LV mass, left ventricular mass. **K**, LVRI, left ventricular remodeling index. **L**, Heart rate in parasternal short axis view. **M**, Diastolic function assessed by E/e' ratio. **N**, Running distance. **O**, Ratio between cardiac ventricle weight and TL. **P**, Ratio between wet lung weight and TL. **Q**, mRNA expression levels of *SR-B1*, *Col3a1*, *Col1a1*, and *Bnp* in the left ventricle were quantified using RT-qPCR.

Comment 7. The authors touch briefly on eNOS and reduced NO bioavailability in the discussion. Have they evaluated nitrosative stress in their model and the impact of the various interventions on this?

Response: Thank you for your valuable concern. We did not further investigate the impact of endothelial SR-B1 deficiency on cardiac nitrosative stress, primarily for three reasons.

First, previous study has demonstrated that nitrosative stress exacerbates cardiac dysfunction in HFpEF mainly due to the abnormal upregulation of iNOS in cardiomyocytes rather than eNOS (*Schiattarella et al., Nature, 2019*)¹⁶.

Second, unlike iNOS, endothelial eNOS primarily regulates vascular blood pressure. In the revised manuscript, we employed radiotelemetry to more accurately assess 24-hour blood pressure in endothelial SR-B1-deficient mice and control mice, yet no significant differences were observed (*Extended Data Fig.3K and 3L*).

Third, SR-B1 deletion is known to inhibit, rather than promote, NO production. The generation of nitrosative stress requires an excessive amount of NO combined with reactive oxygen species (ROS).

In summary, the current research findings do not support the hypothesis that nitrosative stress serves as a core potential risk factor for the deterioration of cardiac function in SR-B1 knockout mice.

Extended Data 3K, The 24-hour mean blood pressure in 10-week-old $S^{fl/fl}$ and S^{dEC} healthy mice was monitored by radiotelemetry at 3 weeks following tamoxifen injection (n = 5-6). Longitudinal blood pressure data were analyzed using a mixed-effects model with repeated measures. **L**, The 24-hour blood pressure profiles of $S^{fl/fl}$ and S^{dEC} HFpEF mice, which underwent a 20-week regimen of high-fat diet (HFD) combined with L-NAME treatment, were monitored using radiotelemetry at 3 weeks post-tamoxifen injection (n = 4; a mixed-effects model with repeated measures).

Comment 8. Please provide a limitations section

Response: The “*Limitations*” section has been incorporated into the revised manuscript and can be found on **page 18 (lines 8-29) and page 19 (line 1)**, as follows:

“**The present study has several limitations that warrant discussion.** The present study has several limitations that warrant discussion. Firstly, acute heart failure and chronic heart failure represent distinct pathological conditions. In our study, the role of SR-B1 was exclusively investigated in the context of acute heart failure. Previous studies have indicated that CXCR3+ T cell infiltration exacerbates cardiac systolic dysfunction⁵, and SR-B1 deficiency facilitates the recruitment of CXCR3+ T cells into the heart. Therefore, SR-B1 may also play a potential role in HFrEF. As such, an ideal preclinical model of chronic HFrEF should be utilized in future research to further elucidate the role of SR-B1 in HFrEF. Second, our investigation into the therapeutic potential of targeting the IRF1-CXCL10/CXCR3 signaling axis for HFpEF was limited. While our findings suggest that nanoparticle-mediated IRF1 knockdown attenuates hypertrophic remodeling in HFpEF, the high mortality rate and severe systolic dysfunction precluded a definitive assessment of its effects on diastolic dysfunction. Future studies are needed to explore whether targeting this signaling axis can ameliorate HFpEF progression. Third, we preliminarily demonstrated that HDL, the primary ligand of SR-B1, reduces CXCL10 expression under inflammatory stimulation. However, the precise mechanism by which SR-B1, a membrane receptor, regulates downstream gene transcription remains unclear and requires further investigation. Fourth, although we validated that EnMT did not occur in HFpEF *in vivo*, an *in vitro* assay indicated that endothelial cells have the capacity to produce collagen. Therefore, future studies should directly evaluate collagen secretion from endothelial cells rather than focusing on the cell transition. Finally, this study was conducted at a single center. Although we reported a correlation between CXCL10 and HFpEF after adjusting for potential confounders, multicenter validation is necessary to confirm these findings. Additionally, HFpEF is a heterogeneous syndrome, but our study focused exclusively on a cardiomyopathy-related subtype to evaluate the role of SR-B1-IRF1-CXCL10/CXCR3 signaling in HFpEF pathogenesis. This narrow focus may introduce bias and limit the generalizability of our results.”

Comment 9. The manuscript has 23 authors and 20 distinct addresses yet not author attribution is provided. Please clarify who did what to warrant authorship. It is a large body of work and likely involved many people, but each author's role should be clarified.

Response: We deeply appreciate your valuable suggestions. The author contributions are as follows:

Lemin Zheng and Mingguo Xu conceptualized, designed, and supervised the study. Yufei Wu, Xiaomei Yang, Yu Bai, Chenze Li, and Peng Wang performed the experiments and analyzed the data. Qing Xu assisted in performing echocardiography. Hui Li and Han Xiao contributed to radiotelemetry for blood pressure monitoring. Xiaoli Rao and Tian Wang conducted the GWAS summary statistics analysis. Jie Chen performed single-cell transcriptomic sequencing data analysis. Huanhuan Cao, Qi Zhang, Mingming Zhao, Rui Zhan, Jie Liu, and Linzhang Huang assisted in conducting in vivo experiments. Fan Xue, Yue Hou, and Lingyun Zu supported the collection and analysis of clinical data. Weidong Gao, Hong S. Lu, and Alan Daugherty provided critical revisions to the manuscript. Yufei Wu and Lemin Zheng wrote the manuscript with input from all authors.

Minor comments:

Comment 1. With the Methods only included as a supplement and not in the main text, the authors should provide some brief insights at least into what was done for each aspect of the Results> For example, the definition of endothelial-specific SR-B1 deletion mice (S^{AEC}) should be clearly provided in the main text.

Response: We have relocated the sections on ethics, animal models, and statistical methods to the revised manuscript on **page 19 (lines 8–29) and page 20 (lines 1-16)**. Additionally, we have included brief explanations in each key results section to enhance clarity. For instance, in the Results 2 “*Endothelial-specific SR-B1 deficiency aggravates diastolic dysfunction in HFpEF*”, the definition of endothelial-specific SR-B1 deletion mice (S^{AEC}) is provided on **page 5 (line 29) and page 6 (lines 1-16)**, as follows:

“To directly investigate the role of endothelial SR-B1 in HFpEF pathogenesis, endothelial-specific SR-B1 knockout mice ($S^{fl/fl}; Cdh5-Cre^{ER}; S^{AEC}$) were designed based on a tamoxifen-inducible Cre-loxP system (Fig. 2A). Briefly, $SR-B1^{fllox/+}$ ($S^{fl/+}$) mice were constructed by inserting loxP sequences into the flanking introns of exon 2 (158 bp) of the mouse *Scarb1* (SR-B1) gene using CRISPR-Cas9 gene-editing technology. Subsequently, tamoxifen-inducible endothelial-specific cadherin 5 (*Cdh5*)- Cre^{ER} mice were crossed with $S^{fl/+}$ mice to generate $S^{fl/+} Cdh5-Cre^{ER}$ mice. Then, S^{AEC} and littermate control $S^{fl/fl}$ mice were obtained from the cross of $S^{fl/+} Cdh5-Cre^{ER}$ mice and $S^{fl/+}$ mice, which were confirmed by DNA genotyping (*Extended Data Fig. 2A and Supplementary Table 1*). In contrast to a previous report⁸, this loxP-inserting strategy did not disrupt SR-B1 gene expression (*Extended Data Fig. 2B*). Finally, SR-B1 protein abundance and knockout specificity were evaluated, demonstrating that endothelial SR-B1 was specifically

deleted, with approximately 70% reduction observed two weeks after tamoxifen injection (*Fig. 2B, Extended Data Fig. 2C and D*).”

Comment 2. Why does the Supplement refer the reader to an online supplement - surely this is it?

Response: We sincerely apologize for this textual error. The term "Online" has been removed from the supplement.

Comment 3. Please better define the difference in drinking status for the adults - "ever drunk" vs "never drunk" is insufficient (are you trying to say regular drinkers or alcoholics? and/or do those who are "never drunk", completely non-drinkers or do they a drink occasionally?)

Response: We greatly appreciate your valuable suggestions. We have redefined the terms "Smoking" and "Drinking" in the Supplementary methods on page 1, lines 26–30, as follows:

“Smoking was defined as “ever smoked” as compared to “never smoked”. Smoking is defined as currently or formerly smoking more than one cigarette per day for a duration of over 6 months. Drinking was defined as “ever drunk” as compared to “never drunk”. Drinking is defined as currently or formerly having alcoholic beverages more than once a week for a duration of over 6 months.”

Comment 4. Please remove all reference to EF data in mice unless these were obtained using PV loops derived with conductance catheters. The Vevo2100 ultrasound system does not reliably derive an accurate EF (unlike later models that dop, eg Vevo3100 onwards). FS data provides sufficient information that is required to confirm HFpEF.

Response: Thank you for your valuable suggestions. We have removed all EF data.

Comment 5. Blot for Extended data Fig 1C is poor for Cdh.

Response: The anti-CDH5 antibody from a new lot (GR3237262-1) was utilized to rerun the immunoblot analysis, and the results are as follows:

C

Extended Data Fig. 1C, Immunoblotting of SR-B1 and CDH5 in adult mouse cardiac endothelial cells (MCECs) and non-MCECs; Cdh5 abundance presented as quality control of isolated MCECs.

Comment 6. Extended data Fig 2D n=3 insufficient if you are trying to convince me that these groups are not different

Response: We sincerely appreciate your valuable suggestions. In *Extended Data Fig. 2E* (Previously as *Extended Data Fig. 2D*), we have expanded the sample size from 3 to 6. As demonstrated in the updated figure, the results align consistently with our earlier findings, confirming that endothelial SR-B1 deletion mediated by Cdh5-Cre does not significantly alter SR-B1 protein levels in macrophages following tamoxifen injection.

Extended Data Fig. 2E, Immunoblotting of SR-B1 in peritoneal macrophages isolated from $S^{fl/fl}$ and $S^{\Delta EC}$ mice 2 weeks after tamoxifen administration (left panel); densitometric analysis of SR-B1 protein level (n=6, right panel). Significant difference was evaluated *via* Student's t test.

Figure legend: The newly added Western blot results(n=3)

Comment 7. Extended Data Fig. 5 - I am convinced the TAC mice have HFREF but please increase n to more than 5

Response: In the current version, *Extended Data Fig. 5A–L* corresponds to the previous *Fig. 5*. In *Extended Data Figure 5A–L*, we have increased the sample size from n = 5 to n = 8–9. The results remain consistent with our earlier findings, confirming that endothelial SR-B1 deletion does not affect the progression of TAC-induced cardiac hypertrophic remodeling, systolic function, or pulmonary edema.

Extended Data Fig. 5 Endothelial-specific SR-B1 is dispensable for acute cardiac systolic dysfunction and pathological remodeling induced by TAC. All cardiac function index parameters obtained from echocardiography were in mice 2 weeks after TAC operation (A-L, n=8,8,9,9). A, Representative color Doppler images of aortic arch in sham and TAC mice (upper, the white arrow indicates the ligation site); pressure gradient across the constriction site was calculated according to the modified Bernoulli equation $4 \times V_{max}^2$, and V_{max} is the maximum peak aortic velocity obtained from Pulsed-wave Doppler imaging in mice 1 week after TAC operation (lower). B, Representative M-mode images in the parasternal short axis view. C, FS, left ventricular fractional shortening. D, LVPWd, left ventricular posterior wall diastolic thickness. E, Left ventricular volume at the end of diastole. F, LVIDd, left ventricular internal diastolic diameter. G, LV mass, left ventricular mass. H, LVRI, left ventricular remodeling index. I, Heart rate in parasternal short axis view. J, Body weight. K, Ratio between cardiac ventricle weight and tibia length (TL). L, Ratio between wet lung weight and TL in $S^{fl/fl}$ and $S^{\Delta EC}$ mice sacrificed 2 weeks after TAC operation (n=8,8,9,9).

To Reviewer #3

The authors show that loss of SRB1 in endothelial cells exacerbates HFpEF. This is associated with increased CXCL10 and IRF1. A CHIP experiment shows that IRF1 binds the CXCL10 promoter. Impressive human data (figure 8) are consistent with the proposed mechanism.

Response: We are deeply grateful for your invaluable suggestions, which have greatly enhanced the quality of our manuscript. With the inclusion of new data, the figure layout has been slightly adjusted. All textual revisions have been marked with a yellow background for clarity. Below are our point-by-point responses with the original comments in black and our responses in blue.

Comment 1. The authors interpret the CHIP data as causality for IRF1 driving CXCL10, but his is not correct. If the author's hypothesis is correct, an EC-specific KO of IRF1 should correct the problem and rescue the phenotype.

Response: We sincerely appreciate your valuable suggestions. We will address your concern from the following three aspects.

(1) To investigate the role of IRF1 in the pathological progression of HFpEF, we employed a targeted endothelial delivery system using RGD-peptide nanoparticles (RGD-Nano) to specifically knock down IRF1 in cardiac endothelial cells of mice, as previously described (*Liu et al., Circulation, 2019*)¹. Initially, we conducted *in vitro* screening using the mouse endothelial cell line H5V and identified an effective siRNA sequence targeting *Irf1* (siIrf1) (*Supplementary Table I*). This sequence significantly downregulated *Cxcl10* expression to 29% of baseline levels, as confirmed by RT-qPCR (*Extended Data Fig. 9A*). To ensure the specificity of the nanomaterial delivery to cardiac endothelial cells, we conjugated Cy5.5 to RGD-Nano and performed fluorescence imaging on cardiac slices. The results demonstrated that the nanomaterials specifically targeted CD31-positive cardiac endothelial cells (*Extended Data Fig. 9B*). Subsequently, we isolated cardiac endothelial cells from mice injected with RGD-Nano formulating either siIrf1 or its scramble control siRNA (siNC). RT-qPCR analysis confirmed the efficient knockdown of *Irf1 in vivo*, which was accompanied by a significant reduction in *Cxcl10* expression (approximately 50%) (*Extended Data Fig. 9C*), consistent with our *in vitro* findings.

To evaluate the therapeutic potential of IRF1 knockdown, we subjected 10-week-old mice to a 10-week HFD plus L-NAME treatment to induce HFpEF, followed by intravenous injection of siRNA-formulating RGD-Nano particles every three days (*Extended Data Fig. 9D*). Prior to treatment, there were no significant differences in body weight, blood glucose, or blood pressure between the two HFpEF groups (*Extended Data Fig. 9E-9H*). During the 10-week nanomedicine treatment period, the siNC HFpEF group exhibited severe mortality (5/7 mice), whereas the siIrf1 group showed a higher survival rate (1/12 mice). This mortality in the siNC group may be attributed to systolic dysfunction (evidenced by a significant decrease in FS) (*Extended Data Fig. 9I*), because only HFD plus L-NAME regimens did not contribute to a significant reduction in FS, suggesting that long-term RGD-Nano injections might have adversely affected cardiac function. Importantly, pathological examination revealed that IRF1 knockdown significantly attenuated

cardiac hypertrophic remodeling (*Extended Data Fig. 9K*), supporting our hypothesis that elevated IRF1 exacerbates cardiac dysfunction in SR-B1-deficient mice. These findings are presented in the revised manuscript on **page 12 (line 29)** and **page 13 (lines 1-29)**.

Extended Data Fig. 9 Knockdown of IRF1 specifically in cardiac endothelial cells significantly ameliorates cardiac pathological remodeling in HFpEF mice. A, Quantitative analysis of *Irf1* and *Cxcl10* mRNA expression levels in mouse endothelial cell line H5V following transfection with either scramble control siRNA (siNC) or IRF1-specific siRNA (siIRF1) ($n = 7$, Student's t-test). B, Representative immunofluorescence microscopy images demonstrating co-localization (indicated by white arrows) of CD31+ endothelial cells (green) and Cy5.5-labeled RGD peptide-conjugated magnetic nanoparticles (red) in cardiac tissue sections. Nuclei were counterstained with DAPI (blue). Scale bar, 100 μm . C, Targeted delivery of IRF1-siRNA using RGD-conjugated nanoparticles significantly reduces *Irf1* and *Cxcl10* mRNA expression in isolated cardiac endothelial cells from wild-type mice, as quantified by RT-qPCR ($n = 5$; Student's t-test). D, Workflow of the experimental design to investigate the role of endothelial-specific IRF1 in the HFpEF pathogenesis. E, Body weight ($n=7, 7, 10$). F, Blood glucose during intraperitoneal glucose tolerance test (ipGTT) ($n=7, 7, 10$), with corresponding area under the curve (G). H, Systolic blood pressure ($n=6, 6, 7$) of mice subjected to 10 weeks of HFD plus L-NAME regimens (One-way ANOVA with Bonferroni's multiple comparisons test). At 10 weeks post-injection of RGD nanoparticles, cardiac systolic function parameters were assessed by echocardiography, including (I) fractional shortening (FS) and (J) left ventricular remodeling index (LVRI) ($n= 7, 2, 9$, differences between sham group and HFpEF-siIRF1 group only analyzed using Student's

t-test). K, Ratio of cardiac ventricle weight to tibia length (TL) (n=7, 7, 10, One-way ANOVA with Bonferroni's multiple comparisons test).

(2) To clarify whether IRF1 rescue CXCL10 expression upregulation following SR-B1 knockdown, we knock-downed the expression of IRF1 in SR-B1 deficient HCMECs in the revised manuscript. As shown in **Fig. 6H**, we observed a significant upregulation of CXCL10 expression in SR-B1-deficient HCMECs, which was reversed upon IRF1 knockout. This result further supports the involvement of IRF1 in SR-B1-mediated CXCL10 upregulation.

Fig. 6H, Representative immunoblot images showing protein expression profiles of SR-B1, IRF1, CXCL10, and GAPDH in HCMECs following transfection with control siNC, siSR-B1, or combined siSR-B1 and siIRF1 (upper panel), and quantitative densitometric analysis of relative protein expression levels for SR-B1, CXCL10, and IRF1 (lower panel, n = 6, Brown-Forsythe test and Welch ANOVA with Dunnett's T3 multiple comparisons).

(3) Finally, we would like to clarify the rationale behind our selection of IRF1. In addition to predictions from PROMO (*Extended Data Fig. 9B*), ChIP-qPCR (*Fig. 6F*), and luciferase assays (*Fig. 6F*), our single-cell RNA sequencing data revealed that IRF1 is the top 1 high-expressed

gene in capillary subcluster 0 (**Fig. 5E**), a subpopulation of inflammatory microvascular endothelial cells that is a key focus of our study.

Fig. 5E, Heatmap depicts expression levels of differentially expressed marker genes in sub-clusters of cardiac microvascular ECs. Yellow represents high gene expression and purple represents low gene expression.

In summary, our *in vivo* and *in vitro* findings demonstrate that IRF1 plays a critical role in SR-B1-mediated CXCL10 upregulation and contributes to the pathological hypertrophic progression of HFpEF. Through targeted endothelial delivery of IRF1 siRNA using RGD-Nano nanoparticles, we observed significant improvements in cardiac hypertrophy and survival rates in HFpEF mice. However, the high mortality rate and severe systolic dysfunction in the control group precluded a definitive assessment of the effects of IRF1 knockdown on diastolic dysfunction. Therefore, we acknowledge the need for further experimental validation to fully elucidate the role of IRF1 in diastolic dysfunction in HFpEF. Additionally, we have explicitly addressed this limitation in the "Limitations" section of our manuscript on **page 18, lines 14-19**.

Comment 2. In figure 1A, B and all other immunofluorescence images, specificity controls are missing.

Response: We sincerely appreciate your valuable suggestions. All immunofluorescence images, specificity controls for all immunofluorescence experiments are provided in *Supplementary Fig. 4*.

Supplementary Fig. 4 All negative controls for immunofluorescence imaging are presented, with each panel clearly labeled to indicate the corresponding fluorescent image number. Tissue sections were incubated with antibody diluent in the absence of the primary antibody, while all other experimental conditions remained consistent.

Comment 3. The colocalization of SRB1 with CD31 (figure 1B) is not convincing

Response: Thank you very much for your suggestions. In **Fig. 1B**, we replaced the CD31 antibody (ab28364, Abcam) and performed co-localization imaging of SR-B1 and CD31 on frozen human heart sections using a new method based on the Treble-Fluorescence immunohistochemical mouse/rabbit kit (RS0035, Immunoway), which significantly enhanced the quality of the initial immunofluorescence data. Similarly, co-localization regions of SR-B1 and CD31 were annotated with white straight arrows. Overall, the immunofluorescence staining results in human hearts demonstrating that SR-B1 is predominantly expressed in microvascular endothelial cells rather than endothelial cells of large vessels.

Fig. 1B, Double-staining of SR-B1 and CD31 in cardiac left ventricle of a healthy human. Scale bar, 100 μ m.

Comment 4. Figure 2B shows that the knockdown is about 80% at the protein level. How many mice were tested? No qPCR data are shown.

Response: In *Figure 2B* ($n = 5$ vs 3), similar to *Figure 1F*, the $n = 1$ actually refers to a pooled sample from 2–3 mice, consistent with the method used for single-cell sequencing of pooled mixed samples. For this experiment, we physically triturated hearts from at least 2 mice, followed by enzymatic digestion. We then used anti-CD31 beads for sorting the endothelial cells, and performed Western blot analysis.

Moreover, we have now added qPCR ($n=11$) data to assess the SR-BI knockout efficiency in the heart, as shown in *Extended Data Fig. 2C*.

C

Extended Data Fig. 2C, *Scarb1* (SR-B1) mRNA abundance was assessed in cardiac ventricle of $S^{fl/fl}$ and $S^{\Delta EC}$ mice two weeks post-tamoxifen administration ($n=11$, Mann-Whitney test).

Comment 5. While the floxed mice without Cre are a good control, the other control (Cre+ and SRB1fl/+ or +/+) is also needed, because none of these mice are genetically pure and there could be contributions from the genome around the Cre site.

Response: We sincerely appreciate your valuable suggestions. Regarding the concern about the potential contribution of the genome around the *Cre* site, we consider this unlikely to significantly affect our study findings.

Our single-cell RNA sequencing results are based on a lineage tracing mouse model (as shown in *Fig. 4D* and *Fig. 5A*), where $SR-B1^{fl/+}$ and $RFP^{+/-}$ with *Cre* mice serve as controls of $SR-B1^{fl/fl}$ and $RFP^{+/-}$ with *Cre* mice. The key difference between the two groups lies in whether the FloxP sites are present on one or both alleles, rather than the presence or absence of *Cre*. This addresses your concern about the appropriateness of the controls.

Furthermore, the single-cell data, supported by this control model, revealed that the secretory factor CXCL10 and IRF1 are key molecules regulated by SR-B1 in inflammatory microvascular

endothelial cells. These findings have been robustly validated in subsequent studies using an alternative control ($S^{fl/fl}$ **without** *Cre*) and *in vitro* cell experiments.

Therefore, we conclude that the SR-B1 deficiency-mediated activation of the IRF1-CXCL10 axis and the resulting infiltration of CXCR3+ T cells into the heart are independent of the presence of *Cre* in the control mice.

Fig. 4D, Workflow of experiment to generate endothelial-specific SR-B1 deficient and RFP over-expression mice (a lineage tracing mouse model), investigating the *in vivo* transdifferentiating fate of endothelial cells in HFpEF in the presence or absence of endothelial SR-B1.

Fig.5 A, A schematic diagram of single-cell transcriptomic sequencing experiment.

Comment 6. Figure 3C: Tail cuff blood pressures are considered unreliable in mice. No telemetric measurements are shown.

Response: We sincerely appreciate your valuable suggestions. In the revised manuscript, we employed radiotelemetry monitoring to more accurately assess the 24-hour circadian blood pressure profiles in endothelial SR-B1-deficient mice and control mice, both in the HFpEF and healthy states. Consistent with our previous results obtained using the tail-cuff blood pressure measurement method, no significant differences were observed (*Extended Data Fig.3K and 3L*).

Extended Data 3K, The 24-hour blood pressure in 10-week-old $S^{fl/fl}$ and $S^{\Delta EC}$ healthy mice was monitored by radiotelemetry at 3 weeks following tamoxifen injection (n = 5-6). Longitudinal blood pressure data were analyzed using a mixed-effects model with repeated measures. **L**, The 24-hour blood pressure profiles of $S^{fl/fl}$ and $S^{\Delta EC}$ HFpEF mice, which underwent a 20-week regimen of high-fat diet (HFD) combined with L-NAME treatment, were monitored using radiotelemetry at 3 weeks post-tamoxifen injection (n = 4; a mixed-effects model with repeated measures).

Comment 7. The flow cytometry data in figure 4G are not compensated correctly.

Response: We sincerely appreciate your valuable suggestions. We have re-optimized the compensation settings and supplemented the gating strategy (*Fig. 4G and Supplementary Fig. 1A.*), as detailed below:

Fig. 4G. Single cells from lineage-tracing mice of different groups were FACS sorted according to RFP and CD31 expression. Top panel: negative controls from a heart of *RFP Δ EC* mice not injected by tamoxifen (left panel); CD31-positive controls from a heart of *RFP Δ EC* mice not injected by tamoxifen (middle panel); RFP-positive controls from a heart of *RFP Δ EC* mice 2 weeks after tamoxifen injection (right panel). Bottom panel, double-channel signals were presented from isolated left ventricular cells in mice from different four groups. RFP signal was detected with PE channel (y axis), while CD31 protein expression was measured through FITC channel (x axis).

A Gating strategy for the EnMT data

Supplementary Fig. 1A Gating strategy for flow cytometry of EnMT. Red fluorescence protein (RFP) from cardiac endothelial cell was detected with PE channel (y axis), and endothelial marker CD31 detected with FITC channel (x axis).

Comment 8. Figure 7: how EC-specific is AAV1? What other cells were tested?

Response: In our AAV1 delivery system, the CXCL10 or SR-B1 expression is driven by the endothelial cell-specific promoter *Cdh5*. This expression vector was provided by Professor Yu Huang from the School of Biomedical Sciences at the Chinese University of Hong Kong, and has been previously utilized for specific *in-vivo* overexpression in mouse aortic endothelial cells (*Huang et al., Circulation Research, 2021*)¹⁰. In the revised manuscript, we confirmed through immunofluorescence analysis of heart sections from mice injected with AAV1 expressing *zsGreen* that *zsGreen* was predominantly expressed in cardiac CD31+ endothelial cells (*Extended Data Fig. 4C*).

C

Extended Data Fig. 4C, Representative fluorescence microscopy images showing *ZsGreen* signals mainly localized in CD31-positive cells in cardiac ventricular sections of wild-type (WT) mice following AAV1-mediated gene delivery. Scale bar, 50 μ m.

Comment 9. The sources for the SRB1 floxed mice and the CDH5-ERT Cre mice should be referenced in the text, not only in the supplemental methods.

Response: We sincerely appreciate your valuable suggestions. The relevant content has been incorporated into the revised manuscript on page 19, lines 9–12, as follows:

“*SR-BI^{fl/fl}* mice were constructed using CRISPR-Cas9 gene-editing technology by Beijing Viewsolid Biotechnology Co. LTD (Beijing, China) from C57BL/6n background mice. *Cdh5-Cre^{ER}* mice were obtained from Model Animal Research Center of Nanjing University (Nanjing, China).”

Comment 10. Figure 1E: This SNP is known, so what is the purpose of this graph?

Response: Your concern is highly appreciated. Our intention was to present the data more visually through the figure. To avoid any potential misinterpretation, we have moved *Fig. 1E* to the Extended Data, where it is now labeled as *Extended Data Fig. 1D*.

References

1. Liu, J. *et al.* Endothelial Forkhead Box Transcription Factor P1 Regulates Pathological Cardiac Remodeling Through Transforming Growth Factor-beta1-Endothelin-1 Signal Pathway. *Circulation* **140**, 665-680 (2019).
2. Gold, J.I., Gao, E., Shang, X., Premont, R.T. & Koch, W.J. Determining the absolute requirement of G protein-coupled receptor kinase 5 for pathological cardiac hypertrophy: short communication. *Circulation research* **111**, 1048-1053 (2012).
3. Wang, M. *et al.* OTUD1 promotes pathological cardiac remodeling and heart failure by targeting STAT3 in cardiomyocytes. *Theranostics* **13**, 2263-2280 (2023).
4. Garcia-Menendez, L., Karamanlidis, G., Kolwicz, S. & Tian, R. Substrain specific response to cardiac pressure overload in C57BL/6 mice. *American journal of physiology. Heart and circulatory physiology* **305**, H397-402 (2013).
5. Ngwenyama, N. *et al.* CXCR3 regulates CD4+ T cell cardiotropism in pressure overload-induced cardiac dysfunction. *JCI insight* **4** (2019).
6. Xiong, J. *et al.* A Metabolic Basis for Endothelial-to-Mesenchymal Transition. *Molecular cell* **69**, 689-698 e687 (2018).
7. Evrard, S.M. *et al.* Endothelial to mesenchymal transition is common in atherosclerotic lesions and is associated with plaque instability. *Nature communications* **7**, 11853 (2016).
8. Huby, T. *et al.* Knockdown expression and hepatic deficiency reveal an atheroprotective role for SR-BI in liver and peripheral tissues. *The Journal of clinical investigation* **116**, 2767-2776 (2006).

-
9. Smolgovsky, S. *et al.* Impaired T cell IRE1alpha/XBP1 signaling directs inflammation in experimental heart failure with preserved ejection fraction. *The Journal of clinical investigation* **133** (2023).
 10. Huang, J. *et al.* KLF2 Mediates the Suppressive Effect of Laminar Flow on Vascular Calcification by Inhibiting Endothelial BMP/SMAD1/5 Signaling. *Circulation research* **129**, e87-e100 (2021).
 11. Yang, L. *et al.* MicroRNA-122-Mediated Liver Detargeting Enhances the Tissue Specificity of Cardiac Genome Editing. *Circulation* **149**, 1778-1781 (2024).
 12. Valacchi, G., Sticozzi, C., Lim, Y. & Pecorelli, A. Scavenger receptor class B type I: a multifunctional receptor. *Annals of the New York Academy of Sciences* **1229**, E1-7 (2011).
 13. Schiattarella, G.G. *et al.* Nitrosative stress drives heart failure with preserved ejection fraction. *Nature* (2019).
 14. Cochran, J.D. *et al.* Clonal Hematopoiesis in Clinical and Experimental Heart Failure With Preserved Ejection Fraction. *Circulation* **148**, 1165-1178 (2023).
 15. Tong, D. *et al.* Female Sex Is Protective in a Preclinical Model of Heart Failure With Preserved Ejection Fraction. *Circulation* **140**, 1769-1771 (2019).
 16. Schiattarella, G.G. *et al.* Nitrosative stress drives heart failure with preserved ejection fraction. *Nature* **568**, 351-356 (2019).

Response to the editorial comments

We sincerely appreciate the insightful feedback from the reviewers and editors, which has significantly improved the quality of our manuscript.

Here are the key editorial issues

1* Please increase the number of animals and biological replicates to ensure adequate statistical power, as highlighted in Referee #2's comments (points 2 and 7);
Sample size was determined based on a prior pilot experiment and power analysis

(1) The point 2 of Reviewer #2 "*N of 5-6 is still quite low (I would be more comfortable with n=10) - a power calculation should be provided to justify such a small sample size.*"

Response:

We are most grateful to the editors and Reviewer #2 for their valuable statistical guidance. We conducted a power analysis using PASS 15 software on the differential data from small sample sizes ($n < 7$), revealing statistical power of < 0.8 for certain findings (as shown in the *Response Letter Table 1 on pages 33-37*). Following the editorial recommendations, we have enhanced the statistical power of the essential data that support our conclusions by increasing both the number of animals and biological replicates. Data in *Response Letter Table 1* that is highlighted with a yellow background indicates increased biological replicates and animal numbers. Data with power below 0.8 are marked in red font.

We have primarily supplemented the following studies:

I. In Fig. 1, the statistical power for Fig. 1E was insufficient (Power = 0.63). Biological replicates were therefore increased to ensure statistical power for this result (Power = 1).

II. In Fig. 2, 3, and 5, statistical power below 0.8 for certain cardiac function and pathological indicators in SRB1 endothelial deleted HFpEF mice. The number of SR-B1 deleted mice with HFpEF was increased, yielding consistent results that enhanced and guaranteed statistical power.

III. Statistical power was below 0.8 in *Fig. 6A* and *6C*. We increased biological replicates that subsequently provided analysis with power greater than 0.8.

IV. Clarification is required regarding the statistical power limitations observed groups with a small number of data points (Table I). However, based on other data,

our core conclusions remain unchanged. Specifically:

---Fig. 3A and 3B: The two sections feature $n=10$. The power for other cardiac function and pathological indicators in mice with SR-B1 deletion was greater than 0.8.

---Fig. 5F: Although analysis of CXCR3 mRNA abundance was statistical power below 0.8, the protein abundance (*Fig.5G*) had adequate power, which is more critical to data interpretation.

---Figure 8C: The statistical power for the differential upregulation of CXCL10 in HFpEF hearts ($n=5-12$) was 0.72, likely attributable to a outlier in the sample. Considering its role as a correlative data point and the ELISA results from large-scale plasma samples confirming the statistically significant upregulation of CXCL10, which also serves as an independent factor, no adjustments were made.

V. The results of the additional biological replicates in Western blot analysis are as follows:

Response Letter Fig.1 The additional Western blot results used for *Fig.1E* statistical analysis. Protein samples were separated by electrophoresis using MOPS running buffer on a 4%-20% Bis-Tris gel.

Response Letter Fig.2 The additional Western blot results used for *Fig.5G* statistical analysis. Protein samples were separated by electrophoresis using MOPS running buffer on a 4%-20% Bis-Tris gel.

Response Letter Fig.3 The additional Western blot results used for Fig.6C statistical analysis. Protein samples were separated by electrophoresis using MOPS running buffer on a 4%-20% Bis-Tris gel.

(2) The point 7 of Reviewer #2 “Fig.1 E, Relative SR-B1 protein abundance in MCECs isolated from 20 weeks of two-hit HFpEF mice and control mice - n = 3 independent experiments using pooled samples - even if these are from pooled samples each comprising 2-3 mouse hearts, is still too low for statistical comparison. 5-6 independent experiments are required to be confident in the stats used. ”

Response:

In accordance with Reviewer #2 request, biological replicates were increased, followed by appropriate statistical analysis. Significant differences were observed (Fig. 1E), with the test yielding a result of Power=1.00. The additional Western blot results are as follows:

Response Letter Fig.1 The additional Western blot results (n=3 vs 2, each protein band representing 2-3 mouse mixture samples).

2* Please include a 4-week post-TAC timepoint (R#2). (Point 5: “Alignment of the time points for comparison in the HFrEF and HFpEF models: I appreciate that the authors were concerned about potential high mortality in the TAC mice on this strain (which an independent lab has reported at 7 weeks post TAC), but to not extend their own follow-up beyond 2 weeks post TAC is disappointing - a 4 week post TAC timepoint should be well within their grasp to address this point.”)

Response:

Endothelial SR-B1 deleted mice (S^{AEC}) and their controls ($S^{fl/fl}$) underwent transverse aortic constriction (TAC) surgery (n=8–10). One week post-surgery, blood velocity at aortic arch ligation site was measured using a high-resolution ultrasound system (Vevo F2). The results demonstrated similar pressure gradient at the ligation site in both groups (*Extended Data Fig. 5P*).

Serial assessments of cardiac function were performed at 2, 4, 6, and 8 weeks post-TAC using a Vevo F2 system, which revealed development of heart failure in mice, as evidenced by a left ventricular EF below 50% (*Extended Data Fig. 5Q*), demonstrating a progressive decline in both ejection fraction (EF) and fractional shortening (FS) that was accompanied by a gradual increase in the left ventricular remodeling index (LVRI). Despite these changes, no significant intergroup differences were detected in any of these measures (*Extended Data Fig. 5Q - S*).

Survival analysis showed that mice began to die from the second week post-surgery, with the mortality rate reaching up to 30% by the eighth week (*Extended Data Fig. 5T*). In light of this increased mortality risk, the mice were euthanized. Pathological examination revealed that SR-B1 deficiency did not significantly alter cardiac hypertrophy or pulmonary edema (*Extended Data Fig. 5U - V*).

These serial assessments of cardiac function in TAC-induced mice do not support a major role for endothelial SR-B1 in driving the progression of HFrEF.

Extended Data Fig.5P-Q, Cardiac functional dynamics after TAC surgery was assessed (n=8-10, Unpaired Student's t-test or Mann-Whitney test). **P**, Pressure gradient across aortic constriction 1 week post-TAC. **Q**, EF. **R**, FS. **S**, LVRI. **T**, Percent survival after TAC surgery. **U**, Ratio between cardiac ventricle weight and tibia length (TL) in $S^{fl/fl}$ and $S^{\Delta EC}$ mice euthanized 8 weeks after TAC surgery. **V**, Ratio between wet lung weight and TL.

3* please providing compelling evidence for the causal role of IRF1 in diastolic dysfunction in HFpEF, as requested by Referee #3;

Response:

Based on our current data and published literature, IRF1 is a key contributor to the pathophysiology of cardiac diastolic dysfunction.

Using single-cell transcriptomic analysis and molecular assays (*Extended Data Fig. 9A-9C*), we revealed increased cardiac endothelial IRF1 expression in heart of diastolic dysfunction. *In vivo* studies have confirmed that deletion of IRF1 in endothelial cells of HFpEF mice using nano-targeting materials reduced mortality and attenuated hypertrophic remodeling (*Extended Data Fig.9D-9N*).

Through *in vitro* and *in vivo* approaches, we demonstrated that IRF1 directly bound to the *Cxcl10* promoter, and significantly up-regulated *Cxcl10* mRNA abundance (*Fig.5E-5G*, and *Extended Data Fig.8*, *Extended Data Fig.9D-9F*). Furthermore, we demonstrated that overexpressing endothelial *Cxcl10* significantly exacerbated diastolic dysfunction and CXCR3+ T-cells cardiotropism *in vivo* (*Fig.7* and *Extended Data Fig.10A-H*). Importantly, the activation of IRF1-CXCL10-CXCR3 axis has recently been implicated in another inflammatory disease¹.

Several independent studies have established a causative link between elevated IRF1 levels and the development of cardiac fibrosis or diastolic dysfunction. Type 4 Cardiorenal Syndrome (CRS-4) is a condition characterized by cardiac diastolic dysfunction resulting from chronic kidney disease (CKD). *In vivo*, IRF1-deleted mice exhibit a marked attenuation of hypertrophic remodeling and associated pathological markers for CRS-4². Furthermore, cardiac fibrosis, a pathological hallmark of diastolic dysfunction in HFpEF, was also effectively suppressed by IRF1 deletion³, and IRF1 was regarded as a causal role of diastolic dysfunction mediated by IL-18 induction of osteopontin⁴.

Overall, these findings establish that IRF1 is a key pathological contributor to cardiac diastolic dysfunction.

Extended Data Fig. 9A, The bubble map of *Irf1* abundance in cardiac capillary endothelial cells from HFpEF and NC groups. **B**, Quantitative analysis of *Irf1* expression levels from the HFpEF and control groups. Mann-Whitney U test was used to evaluate the statistical difference. **C**, *Irf1* abundance was measured in mouse cardiac ECs from normal control and HFpEF mice at 20 weeks of HFD plus L-NAME regimens (n=6).

4* Please include a true rescue experiment involving a double EC-specific deleted of IRE1 and SR-B1 to determine whether IRE1 deleted rescues the EC-specific SR-B1 deleted phenotype (R#3);

Response:

Thank you for raising this important point. Reviewer #3 commented in Point #1: *“The authors interpret the CHIP data as causality for IRF1 driving CXCL10, but this is not correct. If the author’s hypothesis is correct, an EC-specific KO of IRF1 should correct the problem and rescue the phenotype.”*

The central concern of this comment relates to whether a causal role of IRF1 in driving CXCL10 expression under SR-B1 loss. The Reviewer #3 explicitly acknowledged that our in vitro experiments support this causality, stating: *“Knocking down (not out) IRF1 and SRB1 rescues the elevated CXCL10 levels in SRB1 knockdown HCMEC cells. This indeed supports causality.”*

As noted by Reviewer #2, our nano-based nucleic acid drug targeting IRF1 should be evaluated in SR-B1 deleted mice. We would like to clarify that our decision to administer this reagent in WT-HFpEF mice was driven by two factors: **firstly**, only a limited number of endothelial-specific SR-B1 deleted mice have been committed to other critical experiments, including gain-of-function experiments and TAC-induced model; **secondly and more importantly**, we reasoned that the intervention effect in WT-HFpEF mice would not differ from that in SR-B1-deficient mice. This is based on the observation that SR-B1 mRNA abundance decreased progressively during HFpEF progression—RNA abundance in cardiac endothelial cells were reduced approximately 40% at 12 weeks, 50% at 18 weeks (*Extended Figure 1G*), and protein abundance declined by about 60% at 20 weeks (*Figure 1E*). By comparison, in inducible endothelial-specific SR-B1 deleted mice, reduction in protein abundance was slightly more pronounced, at around 70%. Therefore, although IRF1 knockdown was performed in WT-HFpEF mice, we consider the therapeutic efficacy to be highly comparable to that observed in the endothelial cell-specific SR-B1 deleted model.

A recent study reported that IRF1 drives CXCL10 upregulation and promotes CXCR3+ T-cell infiltration in an inflammatory disease¹—a mechanistic axis that aligns well with our own preclinical and clinical data. Therefore, our in vivo cardioprotective intervention targeting endothelial IRF1 deletion can also be viewed as supporting evidence for the rescue phenotype observed in vitro.

Additionally, we have revised the inappropriate characterization in the manuscript that presented *“The ChIP data as direct causality for IRF1-driven transcription”* (see manuscript Page 12, Line 19 to Page 13, Line 7).

Extended Data Fig.1G, *Scarb1* mRNA abundance in mouse cardiac ECs at various time points following administration of a high-fat diet (HFD) combined with L-NAME (n=5-6 / group, Welch's or Student's t-test).

Fig.1E, Relative SR-B1 protein abundance in MCECs isolated from 20 weeks of two-hit HFpEF mice and control mice was measured by immunoblotting (upper panel) and was statistically analyzed via Student's t-test (lower panel, n=5-6 independent experiments using pooled samples, with each data point representing 2-3 mouse hearts).

Current HFpEF pathogenesis research mainly focuses on cardiomyocyte metabolic dysregulation, while largely overlooking microvascular endothelial inflammation and cardiac immune remodeling. Our integrated preclinical and clinical data demonstrate that microvascular endothelial SR-B1 deficiency induced CXCL10 secretion and recruited CXCR3⁺ T-cells, establishing a key immune remodeling mechanism in HFpEF. This discovery not only deepens our understanding of HFpEF pathogenesis but also identifies novel targets for immunotherapy. Generating, characterising, and analyzing the conditional double-deleted mouse model would

require at least 1-2 years. The present study is highly conceptually innovative and conceptually timely. Given the highly competitive nature of this research field, we hope the Reviewer agrees that we should not delay reporting our findings for such an extended period to develop the mouse model.

5* Please report whether female HFpEF mice exhibit microvascular abnormalities. (Reviewer 2# Point 5: “Lack of results confirmed in females, in whom HFpEF is the major HF phenotype - the authors do not appear to have investigated this themselves but instead rely on a letter (not a full article) of the work of others. Some labs (others, but also our own lab has unpublished data showing this) can demonstrate diastolic dysfunction and fibrosis in female HFpEF mice. Further, even if diastolic dysfunction is not evident, microvascular abnormalities (which is the focus of the study here) are likely still evident.”)

Response:

We agree with the Reviewer #2 that the phenotype of female mice should be investigated, particularly given the higher clinical prevalence of HFpEF in women.

First, we established a two-hit female HFpEF model by subjecting ovariectomized (OVX) mice to a high-fat diet (HFD) supplemented with L-NAME (two-hit model). This model was compared with our novel, currently unpublished HFpEF female mouse model induced by HFD plus mineralocorticoid DOCA administration (novel HFpEF). Echocardiographic and pathological analyses revealed that after 20 weeks on the specific diet, the two-hit female mice developed significant—though relatively attenuated—hypertrophic remodeling and diastolic dysfunction without evident lung edema (*Response Letter Fig.4A–F*), which suggests that the two-hit female mouse model may have limited applicability for studying clinical HFpEF in women.

Furthermore, key pathological indices such as lung edema and fibrosis were absent in female mice (*Supplementary Fig.2A*), but their prominence was observed in males (*Fig.3B-3F*). This apparent protective effect in female mice stands in stark contrast to the higher susceptibility observed in women with HFpEF ⁵. A prior study from Dr. Joseph Hill's group on sexual dimorphism in this model also indicated that females are protected in this preclinical HFpEF model ⁶.

Our present study focuses on microvascular abnormalities, with key representative molecules including *Irf1*, *Cxcl1*, *Cxcl2*, *Cxcl10*, and *Icam1* (*Fig.5E*)—we further examined the abundance of these representative molecules in the two-hit female HFpEF model. However, no significant changes were detected (*Supplementary Fig.2*). Similarly, a recent study ⁷ reported downregulation of XBP1s, a pivotal player in their findings, in splenic T cells from male HFpEF mice under the two-hit model, while its expression was unaltered in females. Therefore, we speculate that this lack of change may be attributable to a protective effect of female sex in this two-hit preclinical model.

Figure for reviewers removed.

The data supporting this point are from an ongoing study and are available from the corresponding author upon reasonable request.

Supplementary Fig. 2 A, *Bnp*, *Col3a1*, and *Col1a1* mRNA abundance in female mouse left ventricles (n=7-8). Data for *Bnp* were analyzed using the Mann-Whitney test. **B**, *Irf1*, *Cxcl1*, *Cxcl2*, *Cxcl10* and *Icam 1* mRNA abundance (n=7-8).

Point-by-point response to the two reviewers:

Reviewer #2 (Remarks to the Author):

1. The single-cell RNA sequencing (scRNA-seq) used does not appear to include any cell types other than endothelial cells, given that sequencing was conducted exclusively on RFP-positive cells (in which only ECs are stained red). Referring to this as scRNAseq (which usually includes ALL cardiac cell types) is hence misleading. Not only is there an absence of cardiomyocytes (which can be included in traditional scRNASeq by isolating cardiomyocyte nuclei, or by undertaking snRNASeq), but fibroblasts, immune cells, other vascular cell types and nerve cells etc are ALL excluded. This specific detail was clearly articulated by the reviewers in the original review of the work.

Response:

We thank the Reviewer for this valuable suggestion. To enhance methodological clarity, we have revised the terminology from "single-cell RNA sequencing" to the more specific "single-cardiac-endothelial-cell RNA sequencing" (scecRNA-seq) in the revised manuscript.

2. N of 5-6 is still quite low (I would be more comfortable with n=10) - a power calculation should be provided to justify such a small sample size.

Response:

We conducted a power analysis using PASS 15 software on the differential data from small sample sizes ($n < 7$), revealing statistical power of < 0.8 for certain findings (as shown in the *Response Letter Table 1 on pages 33-37*). Following the editorial recommendations, we have enhanced the statistical power of the essential data that support our conclusions by increasing both the number of animals and biological replicates. Data in *Response Letter Table 1* that is highlighted with a yellow background indicates increased biological replicates and animal numbers. Data with power below 0.8 are marked in red font.

We have primarily supplemented the following studies:

I. In Fig. 1, the statistical power for Fig. 1E was insufficient (Power = 0.63). Biological replicates were therefore increased to ensure statistical power for this result (Power = 1).

II. In Fig. 2, 3, and 5, statistical power below 0.8 for certain cardiac function and pathological indicators in SRB1 endothelial deleted HFpEF mice. The number of SR-B1 deleted mice with HFpEF was increased, yielding consistent results that enhanced and guaranteed statistical power.

III. Statistical power was below 0.8 in *Fig. 6A* and *6C*. We increased biological

replicates that subsequently provided analysis with power greater than 0.8.

IV. Clarification is required regarding the statistical power limitations observed groups with a small number of data points (Table I). However, based on other data, our core conclusions remain unchanged. Specifically:

---Fig. 3A and 3B: The two sections feature $n=10$. The power for other cardiac function and pathological indicators in mice with SR-B1 deletion was greater than 0.8.

---Fig. 5F: Although analysis of CXCR3 mRNA abundance was statistical power below 0.8, the protein abundance (*Fig.5G*) had adequate power, which is more critical to data interpretation.

---Figure 8C: The statistical power for the differential upregulation of CXCL10 in HFpEF hearts ($n=5-12$) was 0.72, likely attributable to a outlier in the sample. Considering its role as a correlative data point and the ELISA results from large-scale plasma samples confirming the statistically significant upregulation of CXCL10, which also serves as an independent factor, no adjustments were made.

V. The results of the additional biological replicates in Western blot analysis are as follows:

Response Letter Fig.1 The additional Western blot results used for *Fig.1E* statistical analysis. Protein samples were separated by electrophoresis using MOPS running buffer on a 4%-20% Bis-Tris gel.

Response Letter Fig.2 The additional Western blot results used for *Fig.5G* statistical analysis. Protein samples were separated by electrophoresis using MOPS running buffer on a 4%-20% Bis-Tris gel.

Response Letter Fig.3 The additional Western blot results used for *Fig.6C* statistical analysis. Protein samples were separated by electrophoresis using MOPS running buffer on a 4%-20% Bis-Tris gel.

3. qPCR (n=11) data showing SR-BI deleted efficiency in the heart: I am unable to readily locate Extended Data Fig. 2C in the revised manuscript (this should be in the main figures of the revised manuscript) but the figure included in the rebuttal is convincing - but suggests a 60% (not 80% knockdown).

Response:

In accordance with the reviewer’s advice, we have moved the results from *Extended Data Fig. 2C* into the main Figures, now displayed as *Fig. 2B*. Regarding instances where qPCR-detected deletion efficiency appears lower than Western blot results, this may stem from qPCR samples being derived from whole ventricular tissue rather than specifically isolated from cardiac endothelial cells. Therefore, we consider the Western blot analysis of SR-B1 in mouse cardiac endothelial cells to better reflect the actual deletion status of SR-B1.

Moreover, the functional role of macrophage SR-B1 in disease pathogenesis is well-established^{8,9}. Our data confirmed that SR-B1 mRNA abundance is detectable in macrophages, and its expression level remains unaltered in macrophages from endothelial-specific SR-B1 deleted mice (*Extended Data Fig. 2D*). Therefore, the lower *Scarb1* mRNA deletion efficiency observed in ventricular samples may be related to the presence of SR-B1 mRNA in non-endothelial cells in the heart.

4. Request for rescue of cardiac function with EC-specific knock out of IRF1, as suggested by Reviewer #3 - the authors have undertaken this, but apparently in wildtypes rather than concomitant EC-specific knockdown of SR-B1 (so they cannot show whether they can rescue the phenotype with this approach). Further, they show that EC-specific knock out of IRF1 mimics EC-specific knockdown of SR-B1 rather than rescuing it. How do they interpret the mechanism of EC-specific knockdown of IRF1 cardioprotection in light of this data?

Response:

We sincerely appreciate this Reviewer’s inquiry regarding this point. Our decision to perform IRF1 knockdown in wild-type HFpEF mice—rather than in healthy wild-type controls—was based on the following rationale: endothelial SR-B1 expression is significantly reduced in HFpEF heart (*Extended Data Fig. 1G* and *Fig. 1E*), and the extent of SR-B1 protein reduction at 20 weeks mirrors the deleted efficiency observed in SR-B1 deleted mice (*Fig. 2B-2C*). Therefore, although IRF1 deletion was performed in WT-HFpEF mice, we consider the therapeutic efficacy to be highly comparable to that observed in the endothelial cell-specific SR-B1 deleted mice.

Both the deletion of cardioprotective SR-B1 and the upregulation of CXCL10 can lead to increased cardiac infiltration of CXCR3+ T cells, elevated fibrosis, and diastolic dysfunction. In vitro experiments confirmed that IRF1 deletion reversed the SR-B1 deletion-induced increase in CXCL10 protein abundance (*Fig. 6I*). Correspondingly, in vivo knocking down endothelial IRF1 significantly reduced CXCL10 mRNA abundance in cardiac endothelial cells, attenuated mortality, and markedly decreased heart weight in mice fed a diet of HFD and L-NAME.

These findings suggest that reduced CXCL10 secretion upon IRF1 deletion diminishes recruitment of CXCR3+ T cells into the heart, representing an important mechanism through which IRF1 inhibition confers cardiac protection, further supporting the recently described broader pathogenic role of the IRF1-CXCL10/CXCR3 axis in other inflammatory conditions¹.

Extended Data Fig.1G, *Scarb1* mRNA abundance in mouse cardiac ECs at various time points following administration of a high-fat diet (HFD) combined with L-NAME (n=5-6 / group, Welch’s or Student’s t-test).

Fig.1E, Relative SR-B1 protein abundance in MCECs isolated from 20 weeks of two-hit HFpEF mice and control mice was measured by immunoblotting (upper panel) and was statistically analyzed via Student's t-test (lower panel, n=5-6 independent experiments using pooled samples, with each data point representing 2-3 mouse hearts).

5. Alignment of the time points for comparison in the HFrEF and HFpEF models: I appreciate that the authors were concerned about potential high mortality in the TAC mice on this strain (which an independent lab has reported at 7 weeks post TAC), but to not extend their own follow-up beyond 2 weeks post TAC is disappointing - a 4 week post TAC timepoint should be well within their grasp to address this point.

Response:

Following the Reviewer's recommendation, we performed transverse aortic constriction (TAC) surgery on endothelial cell-specific SR-B1 deleted (S^{AEC}) mice and their controls ($S^{fl/fl}$) to monitor changes in cardiac function over time. One week post-surgery, blood velocity at aortic arch ligation site was measured using a high-resolution ultrasound system (Vevo F2). The results demonstrated similar pressure gradient at the ligation site in both groups (*Extended Data Fig. 5P*).

At the 2nd, 4th, 6th, and 8th week after TAC, cardiac function was assessed using the Vevo F2 system.

Cardiac ultrasound revealed heart failure with reduced ejection fraction (HFrEF, $EF < 50\%$; *Extended Data Fig. 5Q*), characterized by a progressive decline in both EF and fractional shortening (FS), accompanied by a gradual increase in the left ventricular remodeling index (LVRI). Despite these changes, no significant intergroup differences were detected in any of these measures (*Extended Data Fig. 5Q-S*).

Survival analysis showed that mice began to die from the 2nd week post-surgery, with the mortality rate reaching up to 30% by the 8th week (*Extended Data Fig. 5T*). In light of this increased mortality risk, the mice were euthanized. Pathological examination revealed that SR-B1 deficiency did not significantly alter cardiac hypertrophy or pulmonary edema (*Extended Data Fig. 5U–V*).

These time-course assessments of cardiac function in TAC-induced mice further suggest that endothelial SR-B1 has a limited impact on the progression of heart failure with HFrEF.

Extended Data Fig. 5P-Q, Cardiac functional dynamics after TAC surgery was assessed (n=8-10, Unpaired Student's t-test or Mann-Whitney test). **P**, Pressure gradient across aortic constriction 1 week post-TAC. **Q**, EF. **R**, FS. **S**, LVRI. **T**, Percent survival after TAC surgery. **U**, Ratio between cardiac ventricle weight and tibia length (TL) in $S^{fl/fl}$ and $S^{\Delta EC}$ mice euthanized 8 weeks after TAC surgery. **V**, Ratio between wet lung weight and TL.

6. Thank you for providing the full blots with actual Mw markers in the rebuttal - these should also be included in the revised manuscript, not just cropped Western blots (not full blots), so the reader can verify this is the actual size of the proteins shown.

Response:

As recommended, we have updated all Western blot images to display actual MW markers, enabling readers to verify the actual size of the proteins.

7. Fig.1 E, Relative SR-B1 protein abundance in MCECs isolated from 20 weeks of two-hit HFpEF mice and control mice - n = 3 independent experiments using pooled samples - even if these are from pooled samples each comprising 2-3 mouse hearts, is still too low for statistical comparison. 5-6 independent experiments are required to be confident in the stats used.

Response:

We increased the number of biological replicates. Subsequent analysis revealed statistically significant differences (*Fig. 1E*), confirmed by a statistical power of 1.00, and the corresponding new Western blot data are presented below.

Response Letter Fig.1 The additional Western blot results (n=3 vs 2, each protein band representing 2-3 mouse mixture samples).

Fig.1E, Relative SR-B1 protein abundance in MCECs isolated from 20 weeks of

two-hit HFpEF mice and control mice was measured by immunoblotting (upper panel) and was statistically analyzed via Student's t-test (lower panel, n = 5-6 independent experiments using pooled samples, with each data point representing 2-3 mouse hearts).

8. Arrows on the immunofluorescence figures are actually V symbols and are not particularly helpful at pointing out what the authors wish to show - an alternate means of highlighting (e.g. including an inset or encircling the relevant area) should be considered.

Response:

Thanks for the Reviewer's suggestion. We have optimized the symbols on the immunofluorescence figures and provided explanations in the legends.

Fig. 1A, Immunofluorescence staining of SR-B1 in left ventricle of mouse hearts co-stained with fluorescence-conjugated isolectin B4 (IsoB4) to label endothelial cells and DAPI to label nuclei. Scale bar, 25 μm . Arrow, SR-B1/IsoB4 co-localization. Dashed ellipse: vascular lumen. **B**, Double-staining of SR-B1 and CD31 in left ventricle of a healthy human heart. Scale bar, 100 μm . Arrow, SR-B1/CD31 co-localization. Dashed ellipse: vascular lumen.

9. Lack of results confirmed in females, in whom HFpEF is the major HF phenotype - the authors do not appear to have investigated this themselves but instead rely on a letter (not a full article) of the work of others. Some labs (others, but also our own lab has unpublished data showing this) can demonstrate diastolic dysfunction and fibrosis in female HFpEF mice. Further, even if diastolic dysfunction is not evident, microvascular abnormalities (which is the focus of the study here) are likely still evident. It is hence inappropriate to state "these limitations are unlikely to significantly impact the conclusions of our current study"

Response:

We would like to thank this Reviewer for the constructive comments.

We agree that the phenotype of female mice should be investigated, particularly given the higher clinical prevalence of HFpEF in women.

First, we established a two-hit female HFpEF model by subjecting ovariectomized mice to a high-fat diet (HFD) supplemented with L-NAME. This model was compared with our novel, currently unpublished HFpEF female model induced by HFD plus mineralocorticoid DOCA administration (novel HFpEF).

Echocardiographic and pathological analyses revealed that after 20 weeks on the specific diet, the two-hit female mice developed significant—though relatively attenuated—hypertrophic remodeling and diastolic dysfunction (**Response Letter Fig. 4A–4F**).

Furthermore, key pathological indices such as lung edema and fibrosis were absent in female mice (*Supplementary Fig.2A*), but their prominence was observed in males (*Fig.3B-3F*). This apparent protective effect in female mice stands in stark contrast to the higher susceptibility observed in women with HFpEF⁵. A prior study from Dr. Joseph Hill's group on sexual dimorphism in this model also indicated that females are protected in this preclinical HFpEF model⁶.

Our present study focuses on microvascular abnormalities, with key representative molecules including *Irf1*, *Cxcl1*, *Cxcl2*, *Cxcl10*, and *Icam1* (*Fig.5E*)—we further examined the abundance of these representative molecules in the two-hit female HFpEF model. However, no significant changes were detected (*Supplementary Fig.2*). Similarly, a recent study⁷ reported downregulation of XBP1s, a pivotal player in their findings, in splenic T cells from male HFpEF mice under the two-hit model, while its expression was unaltered in females. Therefore, we speculate that this lack of change may be attributable to a protective effect of female sex in this two-hit preclinical model.

Figure for reviewers removed.

The data supporting this point are from an ongoing study and are available from the corresponding author upon reasonable request.

Supplementary Fig.2A, *Bnp*, *Col3a1*, and *Col1a1* mRNA abundance in female mouse left ventricles (n=7-8). Data for *Bnp* were analyzed using the Mann-Whitney test. **B**, *Irf1*, *Cxcl1*, *Cxcl2*, *Cxcl10* and *Icam 1* mRNA abundance (n=7-8).

10. eNOS, NO bioavailability and nitrosative stress: the authors do not measure any of these (BP relies on many factors and is not a surrogate measure of NO bioavailability or NOS functionality). It is not appropriate for the authors to state that "the current research findings do not support the hypothesis that nitrosative stress serves as a core potential risk factor for the deterioration of cardiac function in *SR-B1* deleted mice" as they have not measured it.

Response:

We appreciate the Reviewer for this critical point. We fully agree that our statement regarding nitrosative stress was overreaching. It was inappropriate to draw a definitive negative conclusion without these direct measurements. We sincerely apologize for this oversight.

Initial experiments in primary mouse cardiac endothelial cells and H5V cells demonstrated that *SR-B1* loss downregulated total eNOS protein and consequently reduced NO levels (*Response Letter Fig.5*). Given the administration of the eNOS inhibitor L-NAME in the two-hit HFpEF model, we prioritized other mechanistic avenues for in-depth investigation.

Response Letter Fig.5 A, Citrulline levels in the left ventricle of healthy mice, a co-product of NO synthesis by eNOS (n=3). **B**, NO levels detected by an NO-sensitive fluorescent probe in mouse cardiac endothelial cells (MCECs). **C, D**, Western blot and quantitative analysis of eNOS and phosphorylated eNOS (p-eNOS) protein levels in cardiac endothelial cells isolated from Homozygous SR-B1 deleted mice ($S^{-/-}$) and Heterozygous SR-B1 ($S^{+/-}$) control mice (n=3-5). **E**, NO levels detected by an NO-sensitive fluorescent probe in mSR-B1-GFP-overexpressing H5V cells. **F, G**, Western blot analysis and quantitative analysis of eNOS protein levels in H5V cells transduced with either control (lenti) or mouse SR-B1 (mSR-B1) lentivirus (n=3). **H**, Quantitative real-time PCR analysis of SR-B1 and eNOS mRNA levels in H5V cells transduced with control (Lenti) or mSR-B1 lentivirus (n=6).

11. All my previous minor comments have now been satisfactorily addressed. However, three additional authors appear to be included: Hui Li, Han Xiao, Lingyun Zu - justification for this should be provided.

Response:

We appreciate this comment. Three additional authors—Hui Li, Han Xiao, and Lingyun Zu—have been added in the first revision. Their inclusion is based on their contributions to radiotelemetric blood pressure monitoring and the collection and analysis of clinical data.

Reviewer #3:

The authors addressed most comments. However, there is still concern with the evidence for causality, the efficacy of the knockdown and the endothelial specificity of AAV1.

Comment 1. The authors interpret the CHIP data as causality for IRF1 driving CXCL10, but this is not correct. If the author's hypothesis is correct, an EC-specific KO of IRF1 should correct the problem and rescue the phenotype.

The authors respond with 3 aspects:

First, they show that knockdown of IRF1 ameliorates pathological remodeling. This experiment does not address the potential causality.

Second, they show that knocking down (not out) IRF1 and SRB1 rescues the elevated CXCL10 levels in SRB1 knockdown HCMEC cells. This indeed supports causality.

Third, they justify their focus on IRF1, which does not address causality. They acknowledge that "further experimental validation to fully elucidate the role of IRF1 in diastolic dysfunction in HFpEF" is needed. This remains a critical limitation.

Response:

Thank you for your valuable comments. Regarding the issues, we would like to provide the following responses:

(1) Evidence of Causality

First, we sincerely appreciate the recognition of our in vitro evidence supporting causality.

Concerning in vivo causality evidence, as Reviewer #2 noted, our nano-based nucleic acid drug targeting IRF1 should ideally be tested in SR-B1 deleted mice rather than in a wild-type background. In this regard, we would like to clarify our experimental rationale: We administered siIRF1 therapy in WT-HFpEF mice, not in wild-type healthy mice. The interventional effects observed in WT-HFpEF mice would not substantially differ from those in SR-B1-deficient mice. This is based on the observation that SR-B1 mRNA abundance decreased progressively during HFpEF progression (*Extended Data Fig.1G*), and protein abundance declined by about 60% at 20 weeks (*Fig.1E*). By comparison, in inducible endothelial-specific SR-B1 deleted mice, protein reduction is slightly more pronounced, at around 70%. Therefore, although IRF1 deletion was performed in WT-HFpEF mice, we expect the therapeutic outcomes to be highly comparable to those in endothelial-specific SR-B1 deleted mice.

A recent study reported that IRF1 drives CXCL10 upregulation and promotes CXCR3+ T-cell infiltration in an inflammatory disease—a mechanistic axis that aligns well with our own preclinical and clinical data¹. Therefore, our in vivo cardioprotective intervention targeting endothelial IRF1 deletion can also be viewed as supporting evidence for the rescue phenotype observed in vitro.

Additionally, we have revised the inappropriate characterization in the manuscript that presented "the ChIP data as direct causality for IRF1-driven transcription" (see Page 12, Line 19 to Page 13, Line 7).

Extended Data Fig.1G, *Scarb1* mRNA abundance in mouse cardiac ECs at various time points following administration of a high-fat diet (HFD) combined with L-NAME (n=5-6 / group, Welch's or Student's t-test).

Fig.1E, Relative SR-B1 protein abundance in MCECs isolated from 20 weeks of two-hit HFpEF mice and control mice was measured by immunoblotting (upper panel) and was statistically analyzed via Student's t-test (lower panel, n=5-6 independent experiments using pooled samples, with each data point representing 2-3 mouse hearts).

(2) Role of IRF1 in diastolic dysfunction

Based on our current data and the existing literature, we believe that IRF1 is a

key contributor to the pathophysiology of cardiac diastolic dysfunction.

Using single-cell transcriptomic analysis and molecular assays (Extended Data Fig. 9A–9C), we revealed increased cardiac endothelial IRF1 expression in heart of diastolic dysfunction. *In vivo* studies have confirmed that knocking down IRF1 in endothelial cells of HFpEF mice using nano-targeting materials reduces mortality and attenuates hypertrophic remodeling (Manuscript Extended Data Fig.9D-9N).

Through *in vitro* and *in vivo* approaches, we demonstrated that IRF1 directly binds to the *Cxcl10* promoter, and significantly up-regulates *Cxcl10* expression (Fig. 5E-5G, and Extended Data Fig.8, Extended Data Fig.9D-9F). Furthermore, we demonstrated that overexpressing endothelial *Cxcl10* significantly exacerbates diastolic dysfunction and CXCR3+ T-cells cardiotropism *in vivo* (Fig.7 and Extended Data Fig. 10A-M). Importantly, the activation of IRF1-CXCL10-CXCR3 axis has recently been implicated in another inflammatory disease¹.

Several independent studies have established a causative link between elevated IRF1 levels and the development of cardiac fibrosis or diastolic dysfunction. Type 4 Cardiorenal Syndrome (CRS-4) is a condition characterized by cardiac diastolic dysfunction resulting from chronic kidney disease (CKD). *In vivo*, IRF1-deleted mice exhibit a marked attenuation of hypertrophic remodeling and associated pathological markers for CRS-4². Furthermore, cardiac fibrosis, a pathological hallmark of diastolic dysfunction in HFpEF, was also effectively suppressed by IRF1 deletion³, and IRF1 was regarded as a causal role of diastolic dysfunction mediated by IL-18 induction of osteopontin⁴.

Overall, these findings establish that IRF1 is a key pathological contributor to cardiac diastolic dysfunction.

Current HFpEF pathogenesis research mainly focuses on cardiomyocyte metabolic dysregulation, while largely overlooking microvascular endothelial inflammation and cardiac immune remodeling. Our integrated preclinical and clinical data demonstrate that microvascular endothelial SR-B1 deficiency induces CXCL10 secretion and recruits CXCR3+ T-cells, establishing a key immune remodeling mechanism in HFpEF. This discovery not only deepens our understanding of HFpEF pathogenesis but also identifies novel targets for immunotherapy.

Generating, characterising, and analyzing the conditional double-deleted model would require approximately 1.5 years. The present study is highly conceptually innovative and conceptually timely. Given the highly competitive nature of this research field, we hope the Reviewer agrees that we should not delay reporting our findings for such an extended period to develop the mouse model.

Extended Data Fig. 9A, The bubble map of *Irf1* expression in cardiac capillary endothelial cells from HFpEF and NC groups. **B**, Quantitative analysis of *Irf1* expression levels from the HFpEF and control groups. Mann-Whitney U test was used to evaluate the statistical difference. **C**, *Irf1* abundance was measured in mouse cardiac ECs from normal control and HFpEF mice at 20 weeks of HFD plus L-NAME regimens (n=6).

Comment 4. Figure 2B shows that the knockdown is about 80% at the protein level. How many mice were tested? No qPCR data are shown.
PCR data show that the knockdown is only ~50%, LESS IN SOME MICE.

Response:

Thank the Reviewer for the comments. Regarding instances where qPCR-detected deletion efficiency appears lower than Western blot results, this may stem from qPCR samples being derived from whole ventricular tissue rather than specifically isolated from cardiac endothelial cells. Therefore, we consider the Western blot analysis of SR-B1 in mouse cardiac endothelial cells to better reflect the actual deletion status of SR-B1.

Moreover, the functional role of macrophage SR-B1 in disease pathogenesis is well-established^{8,9}. Our data confirmed that SR-B1 mRNA abundance is detectable in macrophages, and its expression level remains unaltered in macrophages from endothelial-specific SR-B1 deleted mice (*Extended Data Fig. 2D*). Therefore, the lower *Scarb1* mRNA deletion efficiency observed in ventricular samples may be related to the presence of SR-B1 mRNA in non-endothelial cells in the heart.

Comment 8. Figure 7: how EC-specific is AAV1?

New figure Extended Data 4C shows that AAV1 is NOT endothelial specific. Most CD31+ cells do NOT express ZsGreen. In fact, the large vessel on the right edge of the figure strongly suggests mosaicism.

What other cells were tested? Apparently none

Response:

A magnified view of *Extended Data Fig. 4C* is provided for clarity. This image clearly shows that AAV1-Cdh5-mediated ZsGreen expression localizes to microvascular endothelial cells (see arrows).

Furthermore, co-staining with the cardiomyocyte marker cTnT and the fibroblast marker PDGFR α revealed that ZsGreen signals were primarily localized within the interstitium between cardiomyocytes, with negligible expression in fibroblasts (*Extended Data Fig. 4D*). The corresponding negative control is provided in *Supplementary Fig. 3M*.

Extended Data Fig. 4C, Representative fluorescence images showing ZsGreen signals mainly localized in CD31-positive cells in cardiac ventricular sections of wild-type (WT) mice following AAV1-mediated gene delivery. Scale bar, 100 μm. Arrow, CD31+ endothelial cells expressing ZsGreen. **D**, Representative fluorescence images, showing signals for ZsGreen, cardiac troponin T (cTnT), and platelet-derived growth factor receptor alpha (PDGFα) after AAV1-mediated gene delivery. Scale bar, 50 μm.

References

1. Cui, W. *et al.* Interferon regulatory factor-1-expressing astrocytes are epigenetically controlled and exacerbate TBI-associated pathology in mice. *Science translational medicine* **17**, eadr5300 (2025).
2. Huang, Y. *et al.* IRF1-mediated downregulation of PGC1alpha contributes to cardiorenal syndrome type 4. *Nature communications* **11**, 4664 (2020).
3. Jiang, D.S. *et al.* Interferon regulatory factor 1 is required for cardiac remodeling in response to pressure overload. *Hypertension* **64**, 77-86 (2014).
4. Yu, Q. *et al.* IL-18 induction of osteopontin mediates cardiac fibrosis and diastolic dysfunction in mice. *American journal of physiology. Heart and circulatory physiology* **297**, H76-85 (2009).
5. Mishra, S. & Kass, D.A. Cellular and molecular pathobiology of heart failure with preserved ejection fraction. *Nature reviews. Cardiology* **18**, 400-423 (2021).

6. Tong, D. *et al.* Female Sex Is Protective in a Preclinical Model of Heart Failure With Preserved Ejection Fraction. *Circulation* **140**, 1769-1771 (2019).
7. Smolgovsky, S. *et al.* Impaired T cell IRE1alpha/XBP1 signaling directs inflammation in experimental heart failure with preserved ejection fraction. *The Journal of clinical investigation* **133** (2023).
8. Galle-Treger, L. *et al.* Targeted invalidation of SR-B1 in macrophages reduces macrophage apoptosis and accelerates atherosclerosis. *Cardiovascular research* **116**, 554-565 (2020).
9. Tao, H. *et al.* Macrophage SR-B1 modulates autophagy via VPS34 complex and PPARalpha transcription of Tfeb in atherosclerosis. *The Journal of clinical investigation* **131** (2021).

Response Letter Table 1. Statistical power values are presented.

Statistical Graph	Significant indicators	Sample Size in 1st Version	Power value in 1st Version	Sample Size in 2nd Version	Power value in 2nd Version
Fig.1C	Pecam1 mRNA level	n=3	1.00	n=3	1.00
	Scarb1 mRNA level	n=3	1.00	n=3	1.00
Fig.1D	SR-B1 protein level	n=5	0.98	n=5	0.98
Fig.1E	SR-B1 protein level	n=3	0.63	n=6,5	1.00
Fig.2C	SR-B1 protein level	n=5,4	1.00	n=5,4	1.00
Fig.2G	LVPWd (HFpEF: $S^{fl/fl}$ vs $S^{\Delta EC}$)	n=6	0.90	n=10	0.91
Fig.2H	LV mass (HFpEF: $S^{fl/fl}$ vs $S^{\Delta EC}$)	n=6	0.50	n=10	0.85
Fig.2I	LVRI (HFpEF: $S^{fl/fl}$ vs $S^{\Delta EC}$)	n=6	0.89	n=10	1.00
Fig.2K	MPI (HFpEF: $S^{fl/fl}$ vs $S^{\Delta EC}$)	n=6	0.63	n=10	0.91
Fig.2L	E/e' (HFpEF: $S^{fl/fl}$ vs $S^{\Delta EC}$)	n=6	0.66	n=10	1.00
Fig.3A	Ventricle/TL (HFpEF: $S^{fl/fl}$ vs $S^{\Delta EC}$)	n=6	0.68	n=10	0.78
Fig.3B	LW/TL (HFpEF: $S^{fl/fl}$ vs $S^{\Delta EC}$)	n=6	0.53	n=10	0.56

Fig.3D	Fibrosis (HFpEF: $S^{fl/fl}$ vs $S^{\Delta EC}$)	n=6	0.52	n=10	0.81
Fig.3E	Col3a1 (HFpEF: $S^{fl/fl}$ vs $S^{\Delta EC}$)	n=5	0.53	n=9	0.94
	BNP (HFpEF: $S^{fl/fl}$ vs $S^{\Delta EC}$)	n=5	0.58	n=9	0.90
Fig.3F	aSMA+ cells (HFpEF: $S^{fl/fl}$ vs $S^{\Delta EC}$)	n=6	0.99	n=6	0.99
Fig.3I	LV mass (HFpEF: $S^{\Delta EC/zGreen}$ vs $S^{\Delta EC/SRB1}$)	n=6	0.90	n=6	0.90
Fig.3J	E/e' (HFpEF: $S^{\Delta EC/zGreen}$ vs $S^{\Delta EC/SRB1}$)	n=6	0.96	n=6	0.96
Fig.3K	Running distance (HFpEF: $S^{\Delta EC/zGreen}$ vs $S^{\Delta EC/SRB1}$)	n=6	0.97	n=6	0.97
Fig.3L	Ventricle/TL (HFpEF: $S^{\Delta EC/zGreen}$ vs $S^{\Delta EC/SRB1}$)	n=6	0.90	n=6	0.90
Fig.5F	Scarb1 mRNA level (HFpEF: $S^{fl/fl}$ vs $S^{\Delta EC}$)	n=5	0.97	n=9, 8	1.00
	CXCL10 mRNA level (HFpEF: $S^{fl/fl}$ vs $S^{\Delta EC}$)	n=5	0.92	n=9, 8	1.00
	CXCR3 mRNA level (HFpEF: $S^{fl/fl}$ vs $S^{\Delta EC}$)	n=5	0.11	n=9, 8	0.41

Fig.5G	CXCL10 protein level (HFpEF: S ^{fl/fl} vs S ^{ΔEC})	n=6	0.81	n=10	1.00
	CXCR3 protein level (HFpEF: S ^{fl/fl} vs S ^{ΔEC})	n=6	0.77	n=10	1.00
Fig.5H	CXCR3+ cells/mm ² (HFpEF: S ^{fl/fl} vs S ^{ΔEC})	n=6	0.57	n=10	0.82
Fig.5K	CXCR3+ cells/mg LV (HFpEF: S ^{fl/fl} vs S ^{ΔEC})	n=6	0.83	n=6	0.83
Fig.6A	SR-B1 mRNA level (siNC vs siSR-B1)	n=3	1.00	n=7	1.00
	CXCL1 mRNA level (siNC vs siSR-B1)	n=3	0.55	n=7	0.80
	CXCL2 mRNA level (siNC vs siSR-B1)	n=3	0.83	n=7	1.00
	CXCL10 mRNA level (siNC vs siSR-B1)	n=3	0.77	n=7	1.00
Fig.6B	CXCL10 mRNA level (Vehicle: siNC vs siSR-B1)	n=5	1.00	n=5	1.00
	CXCL10 mRNA level (LPS: siNC vs siSR-B1)	n=5	1.00	n=5	1.00
Fig.6C	CXCL10 protein level (Vehicle: siNC vs siSR-B1)	n=5	0.74	n=9	1.00

	CXCL10 protein level (siNC: Vehicle vs LPS)	n=5	0.88	n=9	1.00
	CXCL10 protein level (LPS: siNC vs siSR-B1)	n=5	0.57	n=9	0.92
Fig.6D	CXCL10 mRNA level siNC vs siSR-B1)	n=5	0.95	n=5	0.95
	CXCL10 mRNA level (siSR-B1 vs si/oe SR-B1)	n=5	0.94	n=5	0.94
Fig.6F	CXCL10 promoter abundance	n=5	0.95	n=5	0.95
Fig.6G	Relative luciferase activities	n=7	1.00	n=7	1.00
Fig.6H	SR-B1 protein level (siNC vs siSR-B1)	n=6-13	1.00	n=6-13	1.00
	SR-B1 protein level (siNC vs siSR-B1+siIRF1)	n=6-13	1.00	n=6-13	1.00
	IRF1 protein level (siNC vs siSR-B1)	n=6-13	0.99	n=6-13	0.99
	IRF1 protein level (siNC vs siSR-B1+siIRF1)	n=6-13	1.00	n=6-13	1.00
	IRF1 protein level (siSR-B1 vs siSR-B1+siIRF1)	n=6-13	1.00	n=6-13	1.00

	CXCL10 protein level (siNC vs siSR-B1)	n=6-13	0.96	n=6-13	0.96
	CXCL10 protein level (siSR-B1 vs siSR-B1+siIRF1)	n=6-13	0.99	n=6-13	0.99
Fig.7Q	CXCR3/CD3 in heart	n=6	0.80	n=6	0.80
Fig.7R	CXCR3+ cells / mg LV	n=6	0.81	n=6	0.81
Fig.8A	BNP mRNA level (Healthy vs HFpEF)	n=5-12	0.94	n=5-12	0.94
	COL3A1 mRNA level (Healthy vs HFpEF)	n=5-12	0.89	n=5-12	0.89
Fig.8B	SCARB1 mRNA level (Healthy vs HFpEF)	n=5-12	0.98	n=5-12	0.98
Fig.8C	IRF1 mRNA level (Healthy vs HFpEF)	n=5-12	1.00	n=5-12	1.00
	CXCL10 mRNA level (Healthy vs HFpEF)	n=5-12	0.72	n=5-12	0.72
	CXCR3 mRNA level (Healthy vs HFpEF)	n=5-12	0.98	n=5-12	0.98

9th Feb 2026

Dear Prof. Zheng,

Thank you for submitting your revised manuscript to EMBO Molecular Medicine. Your manuscript will be ready for acceptance pending the following final amendments:

- 1) Authors: Please update the order in which the author names are listed in our system; it should match what is in the manuscript text.
- 2) In the main manuscript file, please do the following:
 - Please address all comments suggested by our data editors listed below:
 - o Data availability statement:
 1. Please note that reviewer access code for GSE317186 dataset is provided in the manuscript.
 - o Figure legends:
 1. Please note that the exact p values are not provided in the legends of figures 1E; 2B, C, G, I, K, L, N; 3A, B, D, E, F; 4B, 5F, G; 6A, B, C, G, I; 8A, B, E, F.
 2. Please note that the box plots need to be defined in terms of minima, maxima, centre, bounds of box and whiskers, and percentile in the legend of figure 8D.
 - o Data citation:
 1. Please note that the data callout in the text for GEO database GSE236585 data citation does not include "Data ref:" as a prefix.
 - Please correct the order of the sections in the manuscript text to: Abstract / The Paper Explained / Introduction / Results / Discussion / Methods / Data Availability / Acknowledgements / Disclosure and Competing Interests Statement / References / Figure Legends / Expanded View Figure Legends
 - Remove the reagent table from manuscript text and upload as a separate .docx file.
 - In Methods, provide the antibody dilutions that were used for each antibody.
 - In Methods, provide the statement that in addition to the WMA Declaration of Helsinki the experiments also conformed to the principles set out in the Department of Health and Human Services Belmont Report.
 - Indicate in legends number and nature of replicates and exact p= values, not a range, along with the statistical test used. To keep the figures "clear" some authors found providing an Appendix table Sx with all exact p-values preferable. You are welcome to do this if you want to.
 - Please rename "Disclosure of interest" to "Disclosure and competing interests statement". We updated our journal's competing interests policy in January 2022 and request authors to consider both actual and perceived competing interests. Please review the policy <https://www.embopress.org/competing-interests> and update your competing interests if necessary.
 - Author contributions: Please remove it from the manuscript and specify author contributions in our submission system. CRediT has replaced the traditional author contributions section because it offers a systematic machine-readable author contributions format that allows for more effective research assessment. You are encouraged to use the free text boxes beneath each contributing author's name to add specific details on the author's contribution. More information is available in our guide to authors: <https://www.embopress.org/page/journal/17574684/authorguide#authorshipguidelines>
 - Correct the reference citation in the text and reference list. In the text, a reference should be cited by author and year of publication. Include a space between a word and the opening parenthesis of the reference that follows. In the reference list, citations should be listed in alphabetical order. Where there are more than 10 authors on a paper, 10 will be listed, followed by "et al.". Please check "Author Guidelines" for more information. <https://www.embopress.org/page/journal/17574684/authorguide#referencesformat>
- 3) Funding: Make sure that information about all sources of funding are complete in both our submission system and in the manuscript. Currently U22A20286 and NO2021001814 are missing in our submission system. Please correct.
- 4) Appendix: Please remove line numbers.
- 5) The Paper Explained: Please add it to the main manuscript text.
- 6) Synopsis: Please check your synopsis text and image before submission with your revised manuscript. Please be aware that in the proof stage minor corrections only are allowed (e.g., typos).
- 7) As part of the EMBO Publications transparent editorial process (see our Editorial at <http://embomolmed.embopress.org/content/2/9/329>), EMBO Molecular Medicine will publish online a Review Process File (RPF) to accompany accepted manuscripts. This file will be published in conjunction with your paper and will include the anonymous referee reports, your point-by-point response and all pertinent correspondence relating to the manuscript. Let us know if you want to remove or not any figures from it prior to publication. Please note that the Authors checklist will be published at the end of the RPF.
- 8) Please provide a point-by-point letter INCLUDING my comments as well as the reviewer's reports and your detailed responses (as Word file).

I look forward to reading a new revised version of your manuscript as soon as possible.

Yours sincerely,

Zeljko Durdevic

Zeljko Durdevic
Senior Editor
EMBO Molecular Medicine

The authors addressed the remaining editorial issues.

19th Feb 2026

Dear Prof. Zheng,

We are pleased to inform you that your manuscript is accepted for publication and is now being sent to our publisher to be included in the next available issue of EMBO Molecular Medicine.

You may qualify for financial assistance for your publication charges - either via a Springer Nature fully open access agreement or an EMBO initiative. Check your eligibility: <https://link.springer.com/journal/44321/how-to-publish-with-us>

Zeljko Durdevic
Senior Editor
EMBO Molecular Medicine

>>> Please note that it is EMBO Molecular Medicine policy for the transcript of the editorial process (containing referee reports and your response letter) to be published as an online supplement to each paper. If you do NOT want this, you will need to inform the Editorial Office via email immediately. More information is available here: <https://link.springer.com/partners/embo-press/editorial-policies#Peer%20review>